



# A process-based evaluation of the Intermediate Complexity Atmospheric Research Model (ICAR) 1.0.1

Johannes Horak[1], Marlis Hofer[1], Ethan Gutmann[2], Alexander Gohm[1], and Mathias W. Rotach[1]

[1]Universität Innsbruck, Department of Atmospheric and Cryospheric Sciences, Innsbruck, Austria
[2]Research Applications Laboratory, National Center for Atmospheric Research, Boulder, Colorado, USA

**Correspondence:** Johannes Horak (johannes.horak@uibk.ac.at)

**Abstract.** The verification of models in general is a non-trivial task and can, due to epistemological and practical reasons, never be considered as complete. As a consequence, a model may yield correct results for the wrong reasons, i.e. by a different chain of processes than found in observations. While in the atmospheric sciences guidelines and strategies exist to maximize the chances that models are correct for the right reasons, these are mostly applicable to full-physics models, such as numerical weather prediction models. The Intermediate Complexity Atmospheric Research (ICAR) model is an atmospheric model employing linear mountain wave theory to represent the wind field. In this wind field atmospheric quantities, such as temperature and moisture are advected and a microphysics scheme is applied to represent the formation of clouds and precipitation. This study conducts an in-depth process-based evaluation of ICAR, employing idealized simulations to increase the understanding of the model and develop recommendations to maximize the probability that its results are correct for the right reasons. To contrast the obtained results from the linear-theory-based ICAR model to a full-physics model, idealized simulations with the Weather Research and Forecasting (WRF) model are conducted. The impact of the developed recommendations is then demonstrated with a case study for the South Island of New Zealand. The results of this investigation suggest three modifications to improve different aspects of ICAR simulations. The representation of the wind field within the domain improves when the dry and the moist Brunt-Väisälä frequencies are calculated in accordance to linear mountain wave theory from the unperturbed base state rather than from the time-dependent perturbed atmosphere. Imposing boundary conditions at the upper boundary different to the standard zero gradient boundary condition is shown to reduce errors in the potential temperature and water vapor fields. Furthermore, the results show that there is a lowest possible model top elevation that should not be undercut to avoid influences of the model top on cloud and precipitation processes within the domain. The method to determine the lowest model top elevation is applied to both the idealized simulations as well as the real terrain case study. Notable differences between the ICAR and WRF simulations are observed across all investigated quantities such as the wind field, water vapor and hydrometeor distributions, and the distribution of precipitation. The case study indicates a large shift in the precipitation maximum for the ICAR simulation employing the developed recommendations in contrast to an unmodified version of ICAR. The cause for the shift is found in influences of the model top on cloud formation and precipitation processes in the ICAR simulations. Furthermore, the results show that when model skill is evaluated from statistical metrics based on comparisons to surface observations only, such analysis may not reflect the skill of the model in capturing atmospheric processes such as gravity waves and cloud formation.





# 1 Introduction

All numerical models of natural systems are approximations to reality. They generate predictions that may further the understanding of natural processes and allow the model to be tested against measurements. However, the complete verification of a
model is impossible for epistemological reasons (Popper, 1935; Oreskes et al., 1994).

This proposition includes models employed in the earth sciences, such as coupled atmosphere-ocean general circulation models, numerical weather prediction models and regional climate models. These models approximate and simplify the world and processes in it by discretizing the governing equations in time and space and by modeling subgrid-scale processes with
adequate parametrizations. The applied simplifications are often the result of a trade-off between physical fidelity of the modeled processes and the associated computational demand. However, even with a firm basis in natural laws, such models may generate results that match measured data but arrive at them through a causal chain differing from that inferred from observations ("right, but for the wrong reason"; e.g. Zhang et al., 2013). Additionally, the reason for a matching result may even be found in unphysical artifacts introduced by the numerical methods of these models (e.g. Goswami and O'Connor, 2010). In
acknowledgment of the fundamental limitation of verification, best practices and strategies have been outlined to maximize the probability that the results obtained from a model are correct for the right reasons (e.g. Schlünzen, 1997; Warner, 2011). Most of these criteria, however, apply to full physics-based models such as regional climate models or numerical weather prediction models that are expected to model atmospheric processes comprehensively.

The Intermediate Complexity Atmospheric Research model (ICAR; Gutmann et al., 2016) employed in this study is intended to be a simplified representation of atmospheric dynamics and physics over mountainous terrain. With a basis in linear mountain wave theory, it is a computationally efficient alternative to full physics regional climate models such as the Weather Researching and Forecasting (WRF; Skamarock et al., 2019) model. Compared to simpler linear-theory-based models of orographic precipitation (e.g. Smith and Barstad, 2004), ICAR allows for a spatially and temporally variable background flow, a detailed
vertical structure of the atmosphere and employs a complex microphysics scheme. However, for instance, precipitation induced by convection or enhanced by non-linearities in the wind field is not considered by ICAR but may be accounted for with other methods (e.g. Jarosch et al., 2012; Horak et al., 2019). For such cases Schlünzen (1997) advises that a model has to be assessed with respect to its limit of application. Therefore, a direct comparison to a full physics-based model is generally not sufficient for an evaluation of ICAR. Note that ICAR is not intended to provide a full representation of atmospheric physics. Furthermore,
whether the results obtained from ICAR simulations are correct for the right reasons cannot be inferred from comparisons to measurements alone (Schlünzen, 1997).

However, in the literature the evaluation efforts for ICAR so far focused mainly on comparisons to measurements or WRF



output. Gutmann et al. (2016) compared monthly precipitation fields for Colorado, USA, obtained from ICAR to WRF output and an observation-based gridded data set. While Gutmann et al. (2016) additionally performed idealized hill experiments, these focused on the qualitative comparison of the vertical wind field and the distribution of precipitation between ICAR and WRF. Bernhardt et al. (2018) applied ICAR to study changes in precipitation patterns in the European Alps in dependence of

the chosen microphysics scheme. Horak et al. (2019) evaluated ICAR for the South Island of New Zealand based on multi-year precipitation time series from weather station data and diagnosed the model performance with respect to season, atmospheric background state, synoptic weather patterns and the location of the model top. By comparing to measurements, Horak et al. (2019) observed a strong dependence of the performance of ICAR on the location of the model top, finding an optimal setting of $4.0\,\mathrm{km}$ above topography that minimized the mean squared errors calculated at all weather stations. However, the analysis of

cross sections revealed numerical artifacts in the topmost vertical levels, suggesting these to be responsible for the high model skill, thus rendering the model right for the wrong reason.

This study aims to improve the understanding of the ICAR model and develop recommendations that maximize the probability that the results of ICAR simulations, such as the distribution of precipitation, are correct for the right reasons. For a given

initial state, a correct representation of the fields of wind, temperature and moisture as well as of the microphysical processes are a necessity to obtain the correct distribution of precipitation for the right reasons. Therefore, simulations of an idealized mountain ridge are employed to investigate and verify the respective fields and processes in ICAR. This study first analyses quantitatively and qualitatively how closely the ICAR wind and potential temperature fields match the analytical solution for the ideal ridge and contrasts them to a WRF simulation to infer the aspects not captured by linear theory (Sect. 4.1). In a sec-

ond step the influence of the height of the model top and the upper boundary conditions on the microphysical cloud formation processes are quantified with a sensitivity study (Sect. 4.2 – 4.4). Thirdly, the differences in the hydrometeor and precipitation distribution due to non-linearities and other processes not represented by linear theory are investigated in a comparison of ICAR to WRF (Sect. 4.5). Finally, the impact of recommendations derived from the preceding steps on a real case are demonstrated (Sect. 4.6). The case study is conducted for the South Island of New Zealand and contrasted to the results of Horak et al.

(2019). All findings are discussed in Sect. 5 and the conclusions, including the recommendations, are summarized in Sect. 6.

## 2   ICAR Model

### 2.1   Overview

ICAR is an atmospheric model based on linear mountain wave theory (Gutmann et al., 2016). The input datasets required by ICAR are a digital elevation model supplying the high-resolution topography and forcing data, i.e., a set of 3-D atmospheric

variables as supplied by atmospheric reanalysis such as ERA5 or coupled atmosphere-ocean general circulation models. The forcing data set represents the background state of the atmosphere and must comprise the horizontal wind components, pressure, temperature and water vapor mixing ratio. ICAR stores all dependent variables on a 3-D staggered Arakawa C-grid (Arakawa and Lamb, 1977, pp.180-181) and employs a terrain-following coordinate system with constant grid cell height.



In contrast to dynamical downscaling models, ICAR avoids solving the Navier-Stokes equations of motion explicitly. Instead, ICAR calculates the perturbations to the horizontal background winds analytically for a given time step by employing linearized Boussinesq-approximated governing equations that are solved in frequency space with the Fourier transformation

(Barstad and Grønås, 2006). Besides the horizontal background winds, these equations depend on the topography and the Brunt-Väisälä frequency $N$ for which, depending on whether a grid cell is saturated or not, either the moist, $N_m$, or dry Brunt-Väisälä frequency $N_d$ is used. The vertical wind speed perturbation is eventually calculated from the density-weighted horizontal winds. The atmospheric quantities (e.g. temperature and moisture), supplied at the domain boundaries by the forcing data set, are advected with the calculated wind field.

In linear mountain wave theory, the wind field is entirely determined by the topography and the background state of the atmosphere (Sawyer, 1962; Smith, 1979) and, for a horizontally and vertically homogeneous background state, given by a set of analytical equations (e.g. Barstad and Grønås, 2006). This formal simplicity is achieved by a number of simplifications such as, for instance, neglecting the interaction of waves with waves, waves with turbulence or non-linear effects such as gravity

wave breaking, time-varying wave amplitudes or low-level blocking and flow splitting. Discussions of the limitations of linear theory resulting from this reduction of complexity can be found in the literature (e.g. Dörnbrack and Nappo, 1997; Nappo, 2012).

ICAR is based on the equations derived in Barstad and Grønås (2006). Therefore, ICAR currently neglects the reflection

of waves at the interface of atmospheric layers with different Brunt-Väisälä frequencies and neglects the vertical increase of the amplitude of the wind field perturbations with drecreasing density.

ICAR allows for the selection of different microphysics (MP) schemes. In this study an updated version of the Thompson MP scheme is employed (Thompson et al., 2008). It predicts mixing ratios for water vapor $q_v$, cloud water $q_c$, cloud ice $q_i$, rain

$q_r$, snow $q_s$ and graupel $q_g$, from here on referred to as microphysics species, as well as the number concentrations for cloud ice and rain. The Thompson MP scheme is a double moment scheme in cloud ice and rain and a single moment scheme for the remaining quantities.

The forcing data set in ICAR represents the atmospheric background state, ideally without the effect of the topography, yielding

a sequence of steady-state wind fields for each forcing time step, between which ICAR interpolates linearly. Statically unstable atmospheric conditions (i.e., $N^2 < 0$) in the forcing data are avoided by enforcing a minimum Brunt-Väisälä frequency of $N_{\min} = 3.2 \times 10^{-4}\,\mathrm{s}^{-1}$ throughout the domain. A full description of ICAR is given by Gutmann et al. (2016).





## 2.2 Modifications to ICAR

The investigations described in this study were conducted with a modified version of ICAR 1.0.1. All modifications are publicly available as download (Gutmann et al., 2020).

### 2.2.1 Calculation of the Brunt-Väisäla frequency

The 3-D fields of potential temperature and microphysics species are initialized in ICAR by linearly interpolating the corresponding fields of the forcing dataset to the high-resolution ICAR domain. From this initial state of the fields at $t_{f_0}$ ICAR calculates the (moist or dry) Brunt-Väisälä frequency $N$ for all model times $t_m$ smaller or equal to the first forcing time $t_{f_1}$. During each model time step the potential temperature and microphysics species fields in the ICAR domain are modified by advection and microphysical processes. For model times $t_m$ between forcing time $t_{f_n}$ and $t_{f_{n+1}}$, $N$ is based on the perturbed

state of the potential temperature and $q_v + q_c + q_i$ at $t_{f_n}$. However, in linear mountain wave theory $N$ is a property of the unperturbed background state (e.g. Durran, 2015), an assumption that is not satisfied by the calculation method employed by the standard version of ICAR. This study therefore employs a modified version of ICAR that, in accordance with linear mountain wave theory, calculates $N$ from the state of the atmosphere given by the forcing data set if the corresponding option is activated. In the following, this modification of ICAR is referred to as ICAR-N, while the unmodified version is referred to as

the original version (ICAR-O). If properties applying to both versions are discussed, the term ICAR is chosen.

### 2.2.2 Treatment of the upper boundary in the advection numerics

ICAR imposes a zero gradient boundary condition (ZG BC) at the upper boundary on all quantities subject to numerical advection. This section details how, particularly for the microphysics species, a ZG BC has the potential to cause problems by e.g., triggering influx of additional water vapor into the domain. Due to its conceptual simplicity, the issue is illustrated for the

upwind advection scheme, which is the standard advection scheme employed by ICAR.

In the following the mass levels are indexed from 1 to $N_z$ and the half levels bounding the $k$-th mass level, i.e. the levels where the vertical wind components is defined as $k - 1/2$ and $k + 1/2$.

The advection equation for a quantity $\psi$ employed by ICAR (Gutmann et al., 2016) is:

$$\frac{\partial \psi}{\partial t} = -\left( \frac{\partial(u\psi)}{\partial x} + \frac{\partial(v\psi)}{\partial y} + \frac{\partial(w\psi)}{\partial z} \right). \tag{1}$$

To arrive at the discrete equations of the upwind advection, the flux divergences $\partial(u\psi)/\partial x$, $\partial(v\psi)/\partial y$ and $\partial(w\psi)/\partial z$ on the right hand side of equation 1 are discretized as, e.g., in Patankar (1980). The vertical flux gradient $\phi_z$ across mass level $k$ at time step $t$ due to downdrafts ($w^t_{k+1/2} < 0$ and $w^t_{k-1/2} < 0$) is then approximated by

$$\phi_z = \frac{\partial(w\psi)}{\partial z} \approx \frac{1}{\Delta z} \left( \psi^t_{k+1} w^t_{k+1/2} - \psi^t_k w^t_{k-1/2} \right), \tag{2}$$





with $\Delta z$ as the vertical grid spacing. The resulting value of $\psi$ at mass level $k$ at time step $t+1$ is calculated with an explicit first-order Euler forward scheme as

$$\psi_k^{t+1} = \psi_k^t - \frac{\Delta t}{\Delta z}\left(\psi_{k+1}^t w_{k+1/2}^t - \psi_k^t w_{k-1/2}^t\right), \tag{3}$$

where $\Delta t$ denotes the length of the time step. At the upper boundary, where $k = N_z$ with $N_z$ being the number of vertical

levels, by default ICAR applies a zero gradient boundary condition to $\psi$ by setting $\psi_{N_z+1} = \psi_{N_z}$. In case of downdrafts (see above) and vertical convergence in the wind field across the topmost vertical mass level ($w_{N_z+1/2} < w_{N_z-1/2}$), this results in a negative vertical flux-gradient and an associated increase in $\psi$ (see equation 3). If $w_{N_z+1/2} < w_{N_z-1/2}$ persists for more than one time step, the concentration of the quantity in the topmost vertical level will continue to increase until it is redistributed within the domain via advection or conversion into other microphysics species. As observed by Horak et al. (2019), this influx

of additional water therefore may cause numerical artifacts such as the formation of spurious clouds.

In contrast to ICAR, full physics models such as the Integrated Forecasting System (IFS) of the European Center for Medium-Range Weather Forecasts (ECMWF, 2018), the COSMO model (Doms and Baldauf, 2018) or the Weather Research and Forecasting (WRF) model (Skamarock et al., 2019) place the location of the upper boundary at elevations high enough where

moisture fluxes across the boundary are negligible. While applying the same treatment to ICAR is, in general, an option, it is undesirable since high model tops would severely increase the computational cost of ICAR simulations. Therefore, this study investigates whether the application of alternative boundary conditions is able to reduce errors caused by, e.g., the unphysical mass influx described above. To this end additional boundary conditions are added to the ICAR code with the option to apply different boundary conditions to different quantities $\psi$. Furthermore this study assesses whether the lowest possible model top

elevation necessary to avoid the model top's impact on the results can be chosen substantially below that of full-physics models without sacrificing the physical fidelity of the results.

## 3 Methods

To investigate ICAR with respect to the influence of the elevation of the upper boundary and the boundary conditions applied to it, idealized numerical simulations and a real case study are conducted. Simulations are run with ICAR-O, ICAR-N and

WRF in order to assess to what degree ICAR simulations approximate the results of the analytical solution and a full-physics model. In addition, WRF is employed to infer differences due to non-linearities.

### 3.1 Simulation setup

Simulations in this study are conducted with version 1.0.1 of ICAR (ICAR-O) and version 4.1.1 of WRF. Additionally, a modification of ICAR-O, referred to as ICAR-N, where the Brunt-Väisälä frequency $N$ is calculated from the background state given by the forcing data set is employed. Note that ICAR-O, on the other hand, calculates $N$ from the perturbed state of the

atmosphere predicted by the ICAR-O. In the idealized simulations the forcing data set is represented by an idealized sounding





while for the real case it is the ERA-Interim reanalysis. For idealized simulations a period of 18 hours is used for spinup and the model output from $t = 19\,\mathrm{h}$ to $t = 30\,\mathrm{h}$ with an interval of $1\,\mathrm{h}$, is evaluated. The ICAR setup for the real case is described in Horak et al. (2019).

The ideal case consists of an infinite ridge extending along the south-north direction in the domain and westerly flow. The horizontal grid spacings of ICAR and WRF are chosen as $\Delta x = \Delta y = 2\,\mathrm{km}$ with $404$ grid points along the west-east axis and open boundary conditions at the western and eastern boundaries. Since ICAR currently does not support periodic boundary conditions, $104$ grid points are employed along the south-north axis to minimize the influence of the boundaries on the domain center. For ICAR, open boundary conditions are imposed at the southern and northern boundaries. WRF, on the other hand, just

uses three grid points along the south-north axis and periodic boundary conditions. The vertical spacing in ICAR simulations is set to $\Delta z = 200\,\mathrm{m}$, while the $26\,\mathrm{km}$ high WRF domain is subdivided in $130$ grid cells, resulting in an average vertical spacing of approximately $200\,\mathrm{m}$. At the lower boundary ICAR and WRF employ a free-slip boundary condition. An implicit Rayleigh dampening layer (Klemp et al., 2008) is applied to the uppermost $16\,\mathrm{km}$ of the WRF domain, with a dampening coefficient of $0.3\,\mathrm{s}^{-1}$.

Idealized ICAR simulations are run for different model top elevations. The elevation of the upper boundary of the domain, referred to as model top elevation $z_{\mathrm{top}}$, is increased by adding additional vertical levels while keeping the vertical spacing constant. The lowest model top is set at $4.4\,\mathrm{km}$ while the highest is located at $14.4\,\mathrm{km}$ with steps of $1\,\mathrm{km}$ in between. The lower end of the model top range reflects the lowest settings employed in preceding studies, such as Horak et al. (2019) where

the optimal setting was determined at $4.0\,\mathrm{km}$ or Gutmann et al. (2016) who set the top of the ICAR domain to $5.64\,\mathrm{km}$. An additional simulation with $z_{\mathrm{top}} = 20.4\,\mathrm{km}$ is conducted to serve as a reference simulation where the cloud processes within the troposphere are not affected by the model top. The Thompson microphysics scheme as described in Sect. 2 is employed in all models. The code of the Thompson MP implementation in ICAR and WRF was reviewed and tested to ensure that both implementations produce the same results for the same input. All input files and model configurations are available for download

(Horak, 2020).

### 3.2 Topographies and initial soundings

The topography is given by a Witch of Agnesi ridge defined by $h(x) = h_m \left( a^2/(x^2 + a^2) \right)$ with a height of $h_m = 1\,\mathrm{km}$ at the domain center at $x = 0\,\mathrm{km}$ and a half width at half maximum of $a = 20\,\mathrm{km}$. Along the $y$-axis the ridge extends through the entire domain. To investigate the influence of the topography, additional ICAR simulations for ridge configurations with

$a = 20\,\mathrm{km}$ and heights of $0.5\,\mathrm{km}$, $2\,\mathrm{km}$ and $3\,\mathrm{km}$ are conducted, as well as $1\,\mathrm{km}$ high ridges with $a = 10\,\mathrm{km}$, $a = 15\,\mathrm{km}$, $a = 30\,\mathrm{km}$ and $a = 40\,\mathrm{km}$, respectively.

The vertical potential temperature profile of the base state is characterized by a potential temperature at the surface, $\Theta_0 = 270\,\mathrm{K}$ and a constant Brunt-Väisälä frequency, $N = 0.01\,\mathrm{s}^{-1}$. The horizontal wind components of the base state are chosen as



$U = 20\,\mathrm{m\,s^{-1}}$ and $V = 0\,\mathrm{m\,s^{-1}}$, and the surface pressure as $p_0 = 1013\,\mathrm{hPa}$. For the comparison of the ICAR and WRF wind fields to an analytical solution, dry conditions with $\mathrm{RH} = 0\,\%$ are employed while otherwise saturated conditions with $\mathrm{RH} = 100\,\%$ are prescribed at all heights. The sensitivity to the base state is investigated by either varying $U$ between $5\,\mathrm{m\,s^{-1}}$ and $40\,\mathrm{m\,s^{-1}}$ in steps of $5\,\mathrm{m\,s^{-1}}$ or varying $N$ between $0.005\,\mathrm{s^{-1}}$ and $0.015\,\mathrm{s^{-1}}$ with a step size of $0.0025\,\mathrm{s^{-1}}$ for the $1\,\mathrm{km}$

high and $20\,\mathrm{km}$ wide ridge. An overview of the parameter space covered by the simulations is given in Table 1. A particular combination of topography and sounding is referred to as scenario.

**Table 1.** Overview of the combinations of topographies and soundings (scenarios) used to initialize the idealized ICAR simulations. Here $h_m$ denotes the ridge height, $a$ the half width at half maximum of the ridge, $U$ the west-east wind component of the base state, RH the relative humidity, $N_d$ the dry Brunt-Väisälä frequency of the base state, $\lambda_z$ the vertical wavelength of the hydrostatic mountain waves for dry conditions and $\epsilon$ the non-dimensional mountain height for dry conditions. The default scenario used, for instance, for the comparison of ICAR to WRF is highlighted in bold.

| $h_m$ (km) | 0.5 | 1.0 | 1.0 | 1.0 | 1.0 | 1.0 | 1.0 | 1.0 | 1.0 | 1.0 | **1.0** | 1.0 | 1.0 | 1.0 | 1.0 | 1.0 | 1.0 | 1.0 | 1.0 | 2.0 | 3.0 |
|---|---|---|---|---|---|---|---|---|---|---|---|---|---|---|---|---|---|---|---|---|---|
| $a$ (km) | 20 | 10 | 15 | 20 | 20 | 20 | 20 | 20 | 20 | 20 | **20** | 20 | 20 | 20 | 20 | 20 | 20 | 30 | 40 | 20 | 20 |
| $U$ (ms$^{-1}$) | 20 | 20 | 20 | 20 | 5 | 10 | 15 | 20 | 20 | 20 | **20** | 20 | 20 | 25 | 30 | 35 | 40 | 20 | 20 | 20 | 20 |
| RH (%) | 100 | 100 | 100 | 100 | 100 | 100 | 100 | 100 | 100 | 0 | **100** | 100 | 100 | 100 | 100 | 100 | 100 | 100 | 100 | 100 | 100 |
| $N_d$ (s$^{-1}$) | 0.01 | 0.01 | 0.01 | 0.01 | 0.01 | 0.01 | 0.01 | 0.005 | 0.0075 | 0.01 | **0.01** | 0.0125 | 0.015 | 0.01 | 0.01 | 0.01 | 0.01 | 0.01 | 0.01 | 0.01 | 0.01 |
| $\lambda_z$ (km) | 12.6 | 12.6 | 12.6 | 12.6 | 3.1 | 6.3 | 9.4 | 25.1 | 16.8 | 12.6 | **12.6** | 10.1 | 8.4 | 15.7 | 18.8 | 22 | 25.1 | 12.6 | 12.6 | 12.6 | 12.6 |
| $\epsilon$ (1) | 0.25 | 0.5 | 0.5 | 0.5 | 2 | 1 | 0.67 | 0.25 | 0.38 | 0.5 | **0.5** | 0.63 | 0.75 | 0.4 | 0.33 | 0.29 | 0.25 | 0.5 | 0.5 | 1 | 1.5 |

For the default scenario with the $1\,\mathrm{km}$ high and $20\,\mathrm{km}$ wide ridge and a background state with $U = 20\,\mathrm{m\,s^{-1}}$, $N = 0.01\,\mathrm{s^{-1}}$ and $\mathrm{RH} = 100\,\%$, the vertical wavelength of hydrostatic mountain waves is $\lambda_z = 2\pi U/N_d = 12.6\,\mathrm{km}$ and the non-dimensional mountain height is $\epsilon = h_m N_d/U = 0.5$. While the listed values for $\lambda_z$ and $\epsilon$ are valid only for dry conditions, they are em-

ployed to summarize the basic characteristics of the background state. For the Witch of Agnesi ridge, the critical value for the onset of wave breaking in a dry (unsaturated) atmosphere is $\epsilon_c = 0.85$ (Miles and Huppert, 1969). Note that while a saturated atmosphere has been shown to increase the values of $\epsilon$ and $\epsilon_c$ (Jiang, 2003), wave breaking does not occur due to $\epsilon < \epsilon_c$. Nonetheless other non-linear effects, such as wave amplification, cannot be completely neglected. The combination of this sounding and topography is therefore suitable as an indicator of how well the ICAR solution approximates scenarios in which

non-linearities occur, a situation ICAR is very likely to encounter in real-world applications. To this end an ICAR-N simulation is compared to a WRF simulation employing the same topography and sounding.

### 3.3   Analytical solution

ICAR calculates the perturbations to the horizontal background wind with analytical equations based on linear theory while the

vertical wind speed is calculated to balance the density-weighted horizontal winds (see Eq. 9; Gutmann et al., 2016). Perturbations to the potential temperature and microphysics species fields, on the other hand, result from advection and microphysical processes calculated with numerical methods. In ICAR-O this introduces a time dependency for $N$ and, in turn, for the wind field perturbations that depend on $N$ as input variable. Furthermore, ICAR assembles the wind field with an algorithm that





allows for a spatially variable background state (Gutmann et al., 2016). It is therefore necessary to ascertain how well the exact analytical perturbations are reproduced by ICAR. This cannot be inferred from a direct comparison to WRF since the wind field of the latter is influenced by non-linear processes not modeled by ICAR. For the topography given in Sect. 3.2 linear-theory-based analytical expressions for the resulting perturbations to a horizontally and vertically uniform background

state have been derived as (e.g. Smith, 1979):

$$u'(x,z) = A(z)N\frac{a\sin(lz) + x\cos(lz)}{a^2 + x^2}, \tag{4}$$

$$w'(x,z) = A(z)U\frac{(x^2 - a^2)\sin(lz) - 2ax\cos(lz)}{(a^2 + x^2)^2}, \tag{5}$$

$$\theta'(x,z) = -A(z)\frac{N^2}{g}\frac{a\cos(lz) - x\sin(lz)}{a^2 + x^2}\Theta, \tag{6}$$

with $u'$ as the perturbation to the horizontal background wind $U$, $w'$ the perturbation to the vertical wind speed, $\theta'$ the per-

turbation to the background potential temperature $\Theta$, $g = 9.81\,\mathrm{m\,s^{-2}}$ as the gravitational acceleration, $l$ the Scorer parameter defined as $l = N/U$ and $A(z)$ as the elevation dependent amplitude of the perturbations. $A(z)$ is given by

$$A(z) = h_m a\sqrt{\rho(0)/\rho(z)}, \tag{7}$$

where $\rho$ is the height-dependent air density of the background state. However, since the underlying equations employed by ICAR neglect the effect of wave amplification due to decreasing density with height, the term $\sqrt{\rho(0)/\rho(z)}$ in equation (7) is

set to unity in the following.

### 3.4 Boundary conditions at the model top

In this study the effect of the boundary conditions (BCs) imposed by ICAR at the upper boundary of the simulation domain is investigated. To this end several alternative BCs to the existing zero gradient boundary condition are added to the ICAR code, their abbreviations and their numerical implementation are summarized in Table 2. Per default ICAR imposes a zero gradient

BC at the model top to all quantities. For this study, options to the ICAR code are added which allow the application of different BCs to water vapor, potential temperature and the hydrometeors (cloud water, ice, rain, snow and graupel) respectively, herein after referred to as set of boundary conditions. To indicate which BCs were applied to what group in a specific model run, the runs are labeled with a three digit code, see Table 3. The first digit indicates the BC imposed on $\theta$, the second digit the BC imposed on $q_v$ and the third digit the BC imposed on the hydrometeors $q_{\mathrm{hyd}}$, which encompass all remaining MP species ($q_c$, $q_i$,

$q_r$, $q_s$ and $q_g$). The number ID associated with each BC is listed in Table 2. In this notation, for instance, $014$ denotes a simulation imposing a zero gradient BC to $\theta$, a constant gradient BC to $q_v$, and a constant flux gradient BC to the hydrometeors $q_{\mathrm{hyd}}$.

The ten combinations of BCs tested in the sensitivity study are listed in Table 3. While a much larger set of combinations of BCs exists, physically not meaningful BC combinations, such as a zero value BC imposed on potential temperature, were

ruled out beforehand. Additionally, to reduce the parameter space further, a preliminary study was conducted to exclude sets of BCs that yielded results with distinctly higher errors than the standard zero gradient BC.





**Table 2.** Overview of all types of boundary conditions that were imposed at the model top of ICAR in the sensitivity study. The table lists the ID number, the abbreviation used in this study, the full name of the BC, and the equation for $\psi_{N_z+1}$ required to calculate the flux at the top boundary of the domain in equation (3). Note that the zero gradient BC is a special case of the constant gradient BC. Due to the upwind advection scheme each BC is only applied if $w_{N_z} < 0$.

| ID | abbreviation | boundary condition | $\psi_{N_z+1}$ |
|----|--------------|--------------------|----------------|
| 0 | ZG | zero gradient | $\psi_{N_z}$ |
| 1 | CG | constant gradient | $\max(0, 2\psi_{N_z} - \psi_{N_z-1})$ |
| 2 | ZV | zero value | $0$ |
| 3 | CF | constant flux | $\frac{w_{N_z-1}}{w_{N_z}}\psi_{N_z}$ |
| 4 | CFG | constant flux gradient | $\frac{1}{w_{N_z}}\left(2\psi_{N_z-1}w_{N_z-1} - \psi_{N_z-2}w_{N_z-2}\right)$ |

**Table 3.** Combinations of BCs tested in the sensitivity study with idealized simulations. Each column represents a combination of three BCs used in a specific simulation. Each digit of the three digit code refers to the ID number of a specific BC listed in Table 2 that was applied to one of the three quantities listed in the rows below. For all combinations of BCs, simulations for all of the topographic settings and background conditions listed in Table 1 were performed.

| quantity | BC combination | | | | | | | | | |
|----------|------|-----|-----|-----|-----|-----|-----|-----|-----|-----|
| code | 000 | 011 | 111 | 114 | 113 | 014 | 044 | 141 | 142 | 133 |
| $\theta$ | ZG | ZG | CG | CG | CG | ZG | ZG | CG | CG | CG |
| $q_v$ | ZG | CG | CG | CG | CG | CG | CFG | CFG | CFG | CF |
| $q_{\text{hyd}}$ | ZG | CG | CG | CFG | CF | CFG | CFG | CG | ZV | CF |

## 3.5 Evaluation

All evaluations conducted in this study focus on cross-sections along the west-east axis of the domain, oriented parallel to the background flow. Since ICAR does currently not support periodic boundary conditions, the ICAR domain is extended along the south-north axis to minimize influences from the boundaries (see Sect. 3.1). Additionally, for ICAR the four centermost west-east cross sections from the south-north axis in the domain are averaged and the average is found as representative of the domain center in preliminary tests (not shown). In WRF the central west-east cross section from the south-north axis is used.

The effect of the Brunt-Väisälä frequency calculation method is investigated with a comparison of the $u'$ and $w'$ fields obtained from ICAR-N and ICAR-O simulations to the fields given by the analytical expressions in equations (4) and (5). Non-linear effects on the wind field are investigated by a comparison of ICAR to WRF. Differences between the models' and the analytical solution are quantified with the bias B and the mean absolute error MAE (MAE, Wilks, 2011b, chap. 8). Since WRF uses a





different model grid than ICAR, WRF fields are linearly interpolated to the ICAR grid for this comparison.

For the evaluation in this study the mixing ratios of the microphysics species are assigned to three groups. Water vapor $q_v$, suspended hydrometeors $q_{\mathrm{sus}} = q_c + q_i$ and precipitating hydrometeors $q_{\mathrm{prc}} = q_r + q_s + q_g$. The total mass of water vapor $Q_v$,

suspended hydrometeors $Q_{\mathrm{sus}}$ and precipitating hydrometeors $Q_{\mathrm{prc}}$ is calculated as

$$Q(t) = V \sum_{i=0}^{N_x} \sum_{j=0}^{N_z} \rho_{ij}(t)\, q_{ij}(t), \tag{8}$$

where $N_x$ and $N_z$ are the horizontal and vertical number of grid cells respectively, $V$ the grid cell volume, $q_{ij}(t)$ the mixing ratio of the respective hydrometeor group and $\rho_{ij}(t)$ the density of dry air within the grid cell. Note that in contrast to WRF the grid cell volume in ICAR is constant and all vertical levels have the same height $\Delta z$.

The sensitivity of the physical processes simulated by ICAR-N to the elevation of the upper boundary and the imposed boundary conditions (BCs) is inferred from the total mass of the MP species in the cross-section and the spatial distribution of potential temperature, the MP species and the 12-h accumulated precipitation $P_{12\mathrm{h}}$. Except for $P_{12\mathrm{h}}$ all quantities are averaged over the 12 hour period after a spinup of $18\,\mathrm{h}$ when an approximately steady state is reached. $P_{12\mathrm{h}}$ is the precipitation accumu-

lated over the same period.

Differences in the spatial distribution of time-averaged quantities $\bar{\psi}$, $P_{12\mathrm{h}}$ and time-averaged total mass of the MP species $\bar{Q}$ with respect to the reference simulation are quantified with the sum of squared errors (SSE). The SSE is calculated between ICAR simulations with different values of $z_{\mathrm{top}}$ and the reference simulation employing the default zero gradient BCs at the

upper boundary where $z_{\mathrm{top}}$ is $z_{\mathrm{max}} = 20.4\,\mathrm{km}$. This model top is high enough so that cloud processes within the troposphere are not affected by the model top. The SSE is calculated over all vertical levels defined in both simulations as

$$\mathrm{SSE}(\psi, z_{\mathrm{top}}, \mathrm{BCs}) = \sum_{i=0}^{N_x} \sum_{j=0}^{N_z} \left( \bar{\psi}_{ij}(z_{\mathrm{top}}, \mathrm{BCs}) - \bar{\psi}_{ij}(z_{\mathrm{max}}) \right)^2. \tag{9}$$

Here $\bar{\psi}_{ij}(z_{\mathrm{top}}, \mathrm{BCs})$ is the time averaged value of a quantity $\psi$ in an ICAR simulation at grid point $(i,j)$ with the model top at $z_{\mathrm{top}}$ and the set of upper BCs, and $\bar{\psi}_{ij}(z_{\mathrm{max}})$ is the value of a quantity at the same location in the reference sim-

ulation with $z_{\mathrm{top}} = z_{\mathrm{max}}$. For 12-hour accumulated precipitation a one-dimensional version of equation (9) with the summation only along the $x$-axis is employed while for total mass no summation is necessary and only the squared difference $(\bar{Q}(z_{\mathrm{top}}, \mathrm{BCs}) - \bar{Q}(z_{\mathrm{max}}))^2$ is calculated. The SSE is preferred over the mean squared error (MSE) since different model top settings result in different domain sizes, potentially favoring simulations with higher model tops due to the larger area that the errors are averaged over. While, conversely, the SSE tends to favor smaller domains, lower SSEs obtained for simulations with

higher model tops are then a stronger indicator that increasing the model top effectively reduces errors.

To quantify the improvement of one simulation (with a set of boundary conditions BCs and model top $z_{\mathrm{top}}$) over another by





choosing a different set of boundary conditions, BCs$'$, at the upper boundary or another model top elevation $z'_{top}$, the reduction of error (RE) measure is employed (Wilks, 2011a, chap. 8). It is given by

$$\mathrm{RE}(\psi) = 1 - \frac{\mathrm{SSE}(\psi, z'_{top}, \mathrm{BCs}')}{\mathrm{SSE}(\psi, z_{top}, \mathrm{BCs})}. \tag{10}$$

This way, RE can be interpreted as a percentage improvement due to the alternative choice of $z'_{top}$ or BCs$'$ over the original

settings $z_{top}$ and BCs, with $\mathrm{RE} = 0$ corresponding to no improvement and $\mathrm{RE} = 1$ corresponding to a complete removal of errors.

To characterize the effect of increasing the model top elevation on the SSE while keeping the set of boundary conditions unchanged, RE is evaluated for increasing values of $z'_{top}$ between $4.4\,\mathrm{km}$ and $14.4\,\mathrm{km}$ with $z_{top} = 4.4\,\mathrm{km}$ and $\mathrm{BCs} = \mathrm{BCs}'$ in

Eq. (10). The resulting RE values then are equivalent to the percentage change of the SSEs achieved by increasing $z_{top}$ in comparison to the lowest tested model top setting. Similarly, to investigate the effect of an alternative set of boundary conditions, RE is evaluated for $z_{top} = z'_{top}$ and $\mathrm{BCs} \neq \mathrm{BCs}'$. Here the resulting RE values quantify the percentage improvement of the SSEs achieved by changing the imposed boundary conditions at the upper boundary while leaving the model top elevation unchanged.

The quantity $z_{min}(\psi, \mathrm{BCs})$ is introduced which defines the model top elevation for a given set of boundary conditions BCs and parameter $\psi$ for which RE exceeds $95\,\%$ for the first time and remains above that threshold for $z_{top} \geq z_{min}$. In preliminary studies the $95\,\%$ threshold value was found as a suitable indicator for reaching a saturation in error reduction (not shown). The lowest possible model top elevation $Z_{min}$ is then calculated as the maximum of $z_{min}(\psi, \mathrm{BCs})$ for all quantities $\psi$ and a particular combination of boundary conditions BCs. However, $\theta$ is excluded since this study focuses mainly on hydrometeors. Nonethe-

less any relevant error in $\theta$ influences the MP fields and the distribution of precipitation, thereby directly affecting $Z_{min}$. In this context $Z_{min}$ can then be interpreted as the lowest possible model top elevation such that the cloud and precipitation processes in the domain are sufficiently independent from influences of the model top.

### 3.6 Case study

To investigate the effects of the suggested modifications to ICAR on the distribution of precipitation for a real world applica-

tion, a case study is conducted for the Southern Alps on the South Island of New Zealand located in the southwestern Pacific Ocean. Furthermore, the procedure to identify the lowest possible model top elevation $Z_{min}$, as described in Sect. 3.5, is applied to this real case scenario and the result compared to the optimal model top elevation of $4\,\mathrm{km}$ found by Horak et al. (2019) for this region. In their study the model top elevation was chosen as the elevation that led to the lowest mean squared errors between simulated and measured 24-h accumulated precipitation for eleven sites in the Southern Alps. This section addition-

ally investigates whether this seemingly optimal result, as suggested by the lowest mean squared errors, was achieved for the wrong reasons. To this end the hydrometeor and precipitation distribution along cross sections through the Southern Alps are compared.





To maintain comparability to Horak et al. (2019), the ICAR simulations for ICAR-O and ICAR-N are forced with the ERA-Interim reanalysis (ERAI, Dee et al., 2011) instead of the more recent ERA5 reanalysis. For the ICAR-O simulation the model top is set to $4\,\mathrm{km}$, the elevation that was identified as seemingly optimal in Horak et al. (2019) and ZG BCs are applied to $\theta$ and all microphysics species (BC code 000). For the ICAR-N simulation $Z_{\min}$ is determined for the day of the case study

as described in Sect. 3.5 by conducting multiple simulations with model tops between $5$–$20\,\mathrm{km}$. A ZG BC is imposed on the potential temperature field to avoid numerical instabilities arising for a CG BC due to strongly stratified atmospheric layers and a CG BC is imposed on the microphysics species (BC code 011). The remaining setup for ICAR-O and ICAR-N, such as the forcing data set and the model domain have been described in detail in Horak et al. (2019).

The case study focuses on the 6 May 2015 LT, a day with stably stratified large-scale northwesterly flow throughout the troposphere impinging on the Southern Alps over a 24-h period. Upstream of the South Island, ERAI exhibits a 24-h averaged relative humidity of more than $80\,\%$ in the lowest $2\,\mathrm{km}$ of the atmosphere, an averaged moist Brunt-Väisäla frequency of $0.012\,\mathrm{s}^{-1}$, a mean near-surface temperature of $16.5\,^{\circ}\mathrm{C}$ and a mean specific humidity at the surface of $11\,\mathrm{g\,kg}^{-1}$.

## 4    Results

### 4.1    Comparison to the analytical solution

Figure 1 shows the horizontal and vertical perturbations to the background state, as well as the isentropes of the perturbed potential temperature field as calculated with the analytical solution based on linear theory and simulated with ICAR-N, ICAR-O and WRF up to an elevation of $15\,\mathrm{km}$. ICAR-N and ICAR-O simulations were run with $z_{\mathrm{top}} = 20.4\,\mathrm{km}$ and zero gradient boundary conditions (BC code 000). The simulations are conducted for a 2-D ridge and the default scenario with the modifi-

cation that $\mathrm{RH} = 0\,\%$ (see Sect. 3.2).

Generally, the horizontal west-east and the vertical perturbations to the background state calculated by ICAR-N reproduce those obtained from the analytical expressions well (cf. Fig. 1a-b and Fig. 1e-f). The range of values of $u'$ in ICAR-N is $-8.4\,\mathrm{m\,s}^{-1}$ to $8.2\,\mathrm{m\,s}^{-1}$ compared to the $-10.0\,\mathrm{m\,s}^{-1}$ to $10.0\,\mathrm{m\,s}^{-1}$ derived from the analytical expression. While, for the

south-north perturbations, the analytical solution yields $v' = 0\,\mathrm{m\,s}^{-1}$, ICAR-N calculates an average magnitude of $0.02\,\mathrm{m\,s}^{-1}$. The minimum and maximum of $v'$ are $-1.6\,\mathrm{m\,s}^{-1}$ and $1.5\,\mathrm{m\,s}^{-1}$ respectively, localized in close proximity to the western and eastern domain boundaries. Along the domain center $v'$ lies between $-0.5\,\mathrm{m\,s}^{-1}$ and $0.5\,\mathrm{m\,s}^{-1}$. For $w'$, values obtained with ICAR-N lie between $\pm 1.1\,\mathrm{m\,s}^{-1}$ as opposed to $\pm 1.0\,\mathrm{m\,s}^{-1}$ for the analytical solution. The mean absolute error (MAE) in relation to the analytical solution of $u'$ is $0.9\,\mathrm{m\,s}^{-1}$, which corresponds to $11\,\%$ of the absolute perturbation maximum. For $w'$

the MAE is $0.027\,\mathrm{m\,s}^{-1}$ or $2\,\%$ of the absolute perturbation maximum. This indicates a smaller error in the $w'$ field in ICAR-N in contrast to the $u'$ field. In comparison to the analytical fields (Fig. 1a) the $u'$ field in ICAR-N exhibits slight distortions, particularly visible in the region where $u' < 0\,\mathrm{m\,s}^{-1}$ from approximately $8\,\mathrm{km}$ upward (Fig. 1b). The isentropes in ICAR-N are overall very similar to those calculated analytically (see Fig. 1a-b), yielding an MAE of $0.26\,\mathrm{K}$.



The wind and potential temperature fields simulated by ICAR-O (Fig. 1c, g) exhibit clear differences to the analytical solution, especially above an elevation of about $6\,\mathrm{km}$. The deterioration increases with elevation and is clearly visible from approximately $z = 8\,\mathrm{km}$ upward, particularly for $w'$ (Fig. 1g) but still well pronounced for $u'$ and the isentropes (Fig. 1c). This

is reflected in slightly elevated MAEs in comparison to ICAR-N with $1.0\,\mathrm{m\,s^{-1}}$ in $u'$, $0.034\,\mathrm{m\,s^{-1}}$ in $w'$ and $0.32\,\mathrm{K}$ in $\theta$. The reason for the relatively small difference to the MAEs of ICAR-N is that the MAE is calculated across the entire cross section while the largest deviations are localized in a comparatively small region around the topographical ridge at the center.

WRF is not expected to perfectly reproduce the analytical solution due to the occurrence of non-linearities for the chosen

non-dimensional mountain height of $\epsilon = 0.5$ and the amplification of perturbations due to the decrease in density with height. Furthermore, the occurrence of partial wave reflections from the model top is not entirely mitigated despite the careful selection of a damping layer (see Sect. 3.1). However, the WRF simulation serves as an indicator to what degree ICAR is able to capture the results obtained with a full-physics model. As expected, the WRF simulation shows a larger deviation from the analytical wind field (cf Fig. 1a, e with Fig. 1d, h). The amplitudes in the perturbation fields in WRF are larger and exhibit an elevation

dependence. For $u'$ the range of observed values is $-14.8\,\mathrm{m\,s^{-1}}$ to $14.6\,\mathrm{m\,s^{-1}}$ and values of $w'$ lie between $-1.7\,\mathrm{m\,s^{-1}}$ and $2.4\,\mathrm{m\,s^{-1}}$. These larger maximum values in comparison to the analytical solution can mainly be attributed to the amplification of the perturbations due to the exponential decrease in density with height. For instance, at the elevation of the $w'$ maximum (Fig. 1h), the pressure has dropped to about one third of the surface pressure. According to the pressure amplification term in Eq. (7) this increases the amplitude by a factor of $1.7$. The remaining difference of $0.7\,\mathrm{m\,s^{-1}}$ is most likely caused by wave

amplification due to non-linearities and wave reflections at the damping layer. However, the general characteristics of the perturbation fields, such as the periodicity of the perturbations with elevation and the approximate location of the positive and negative perturbations, are similar to that of their corresponding analytical counterparts. The increase in the amplitude of the perturbations due to the exponential decrease in density with height continues up until approximately $15\,\mathrm{km}$ (not shown) above which the dampening effects of the damping layer become increasingly noticeable.



**Figure 1.** Perturbations of the horizontal perturbation wind component $u'$ (top row) and vertical perturbation wind component $w'$ (bottom row) calculated analytically (left column) and calculated by ICAR-N (second column), ICAR-O (third column) and WRF (right column). The vertical wavelength of a two-dimensional hydrostatic mountain wave $\lambda_z$ is indicated by the dash-dotted horizontal line, the dotted curve shows the $0\,\mathrm{m\,s^{-1}}$ countour line and the solid black contour lines show the isentropes. For panel (a) and (e), where the perturbation field is evaluated on constant height levels starting at $z = 0\,\mathrm{m}$, the topography is indicated by the dashed curve as to not obscure the perturbation field. All simulations are conducted for a 2-D ridge with $h_m = 1\,\mathrm{km}$ and $a = 20\,\mathrm{km}$ and a background state with $U = 20\,\mathrm{m\,s^{-1}}$, $N_d = 0.01\,\mathrm{s^{-1}}$ and RH = 0 %.





## 4.2 Sensitivity to the set of upper boundary conditions

Figures 2a-e show the reduction of error (RE) achieved for ICAR-N simulations for a given model top elevation $z_{\text{top}}$ by applying different upper boundary conditions than the ICAR default (BC code 000). RE values are largest when a CG BC is chosen for $\theta$ (Fig. 2a), more dependent on $z_{\text{top}}$ for $q_v$ (Fig. 2b) and smallest for the remaining quantities (Fig. 2c-e) with similar

results for all tested topographies and the respective time averaged total masses $\overline{Q}_v$, $\overline{Q}_{\text{sus}}$ and $\overline{Q}_{\text{prc}}$ (not shown). Most tested BC combinations reduce the error in at least one of the investigated quantities, but generally not for all, with the exception of the combinations 141 and 142. However, in case of $q_{\text{sus}}$, $q_{\text{prc}}$ and $P_{12\text{h}}$ no improvements for any BC combination are observed once $z_{\text{top}} > 4.4\,\text{km}$ (Fig. 2c-e). Potential temperature fields are improved the most when a CG BC is imposed on $\theta$ (Fig. 2a). The water vapor field shows improvements for all BCs except for a CF BC, with the largest REs found for a CG BC imposed

on $q_v$. For the hydrometeors and $P_{12\text{h}}$ the improvement at the lowest model top setting of $4.4\,\text{km}$ is only found if a CFG BC is applied to water vapor and either a CG, ZV or CFG to $q_{\text{hyd}}$, otherwise the RE is approximately zero.

The choice of an alternative BC over the standard ZG BC has the largest potential for a reduction of error when the grid cells of the uppermost vertical level coincide with (i) regions of vertical convergence where $w < 0$ and $dw/dz < 0$ and (ii)

when the vertical flux gradients $\phi_z$ in these regions are negative (see Sect. 2.2.2). For potential temperature, in case of the specified sounding, both conditions are always satisfied in some region no matter at what elevation the model top is chosen, see Figure 3a where the vertical flux gradient of the potential temperature divided by the local potential temperature, given by $\tilde{\phi}_z(\theta) = \phi_z(\theta)/\theta$, is shown. Consequently $\theta$ exhibits the largest reductions of error across all values of $z_{\text{top}}$ with only a small dependence on $z_{\text{top}}$ (see Fig. 2a). For water vapor, as shown in Fig. 2b, RE as a function of $z_{\text{top}}$ exhibits two peaks, the

first at $z_{\text{top}} = 4.4\,\text{km}$, and a second peak at $z_{\text{top}} = 11.4\,\text{km}$ with a minimum in between. Here the exponential decay of $q_v$ with height results in comparatively small values for $\phi_z(q_v)$ above an elevation of $4\,\text{km}$ (not shown). However, $\tilde{\phi}_z(qv)$ still exhibits minima and maxima at higher elevations due to the periodicity of the vertical velocity field (see Fig. 3b). At the locations of these minima und maxima of $\tilde{\phi}_z(q_v)$ the relative error introduced by a boundary condition can therefore be large as well. In case of $q_v$, as shown in Fig 3b, the model top of a simulation with $z_{\text{top}} = 11.4\,\text{km}$ would coincide with a downdraft region

of strong vertical convergence and negative $\tilde{\phi}_z(q_v)$ close to the domain center, implying strong water vapor flux convergence. The same situation occurs for $z_{\text{top}} = 4.4\,\text{km}$ albeit in a region with a lower value of $\tilde{\phi}_z(q_v)$ and weaker vertical convergence. Therefore, the local change in $q_v$ due to a mass influx caused by the boundary condition is comparatively small, resulting in a lower relative error. Note that for simulations with $4.4\,\text{km} < z_{\text{top}} < 11.4\,\text{km}$ the vertical convergence in downdraft regions at the model top is weaker and $\tilde{\phi}_z(q_v)$ is lower. Therefore, as shown in Fig. 2b, the RE achieved for $q_v$ exhibits two peaks where

the RE is high for the lowest model top setting at $4.4\,\text{km}$, exhibits a maximum at $z_{\text{top}} = 11.4\,\text{km}$ and is low otherwise.



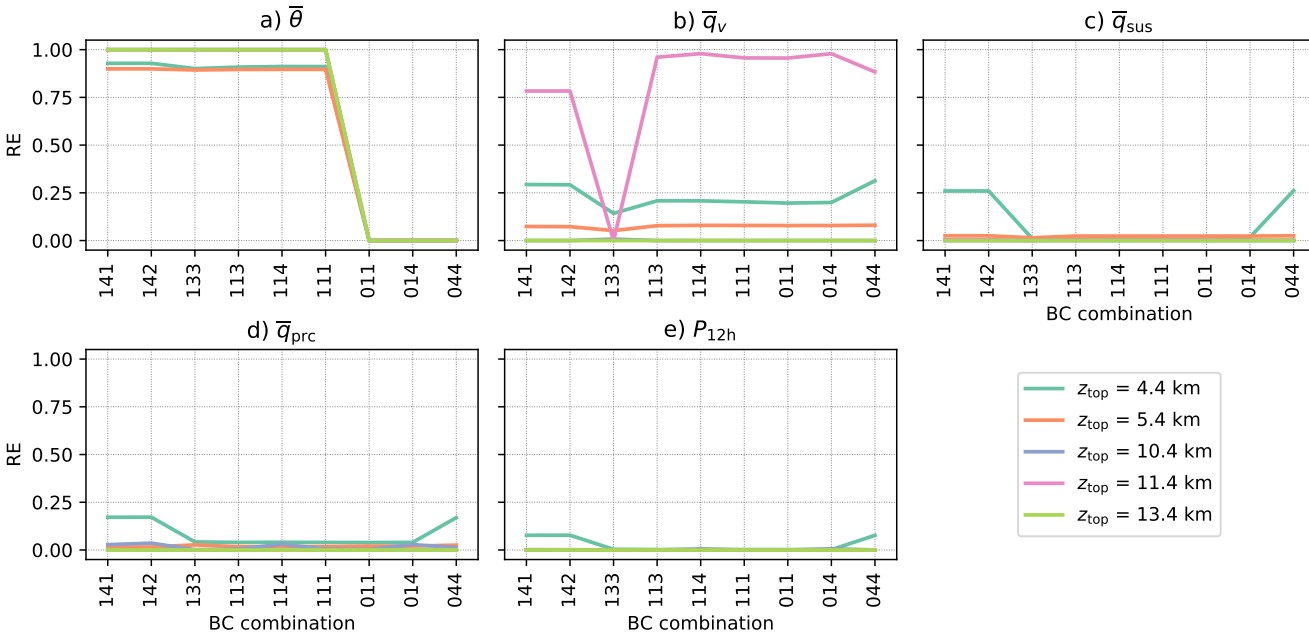

**Figure 2.** The reduction of error (RE) in dependence of the chosen combination of boundary conditions ($x$-axis, see Table 3 for the key to the BC combination code) for (a) potential temperature $\overline{\theta}$, (b) water vapor $\overline{q}_v$, (c) suspended hydrometeors $\overline{q}_{sus}$, (d) precipitating hydrometeors $\overline{q}_{prc}$ and (e) the 12-h precipitation sum $P_{12h}$. Note that overbars denote the temporal average of the respective quantity over 12 hours following 18 hours of model spinup. REs were calculated between an ICAR-N simulation with an alternative set of boundary conditions imposed at the upper boundary and an ICAR-N simulation employing the standard zero gradient boundary condition (BC code 000), both run with the same model top elevation $z_{top}$ (indicated by line color). All simulations are conducted for the default scenario.



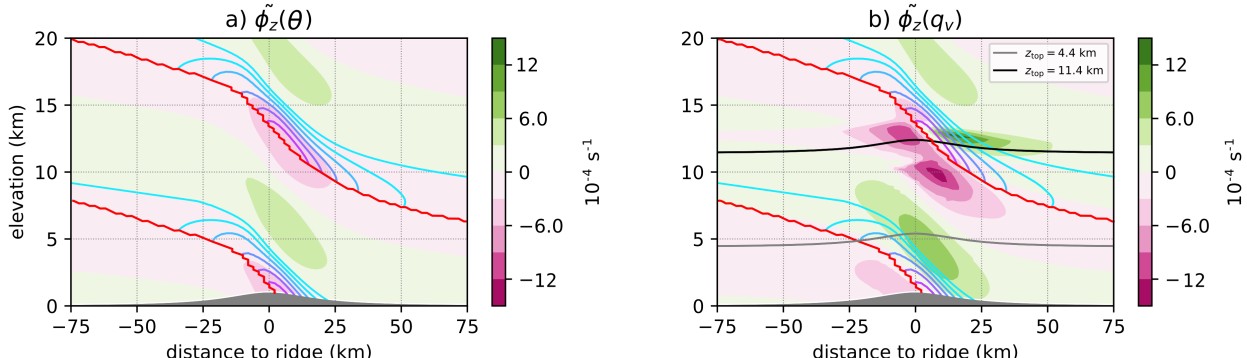

**Figure 3.** The normalized vertical flux gradient of (a) potential temperature and (b) water vapor (see text for further description). The values are calculated from an ICAR-N simulation at $t = 30\,\mathrm{h}$ with $z_{\mathrm{top}} = 20.4\,\mathrm{km}$ and ZG BCs (000) for the default scenario. The contour lines indicate the vertical convergence ($dw/dz < 0\,\mathrm{s}^{-1}$) in regions were $w < 0\,\mathrm{m\,s}^{-1}$. Here the violet contour lines represent stronger and the teal contour lines weaker vertical convergence in the range of $\pm 1.5 \cdot 10^{-3}\,\mathrm{s}^{-1}$ spaced in increments of $0.3 \cdot 10^{-3}\,\mathrm{s}^{-1}$. The red contour line indicates where $w = 0\,\mathrm{m\,s}^{-1}$. In panel (b) grey and black lines additionally indicate the location of the model top for $z_{\mathrm{top}} = 4.4\,\mathrm{km}$ and $z_{\mathrm{top}} = 11.4\,\mathrm{km}$, respectively.

For the investigated scenarios, altering the boundary condition applied to $\theta$ has only a negligible effect on the microphysics species fields and $P_{12\mathrm{h}}$. This is observed, for instance, for simulations 011 and 111 where the BC applied to $\theta$ was changed from a ZG to CG while the BCs imposed on the MP species remained the same: Both BC settings lead to very similar RE values for the MP species (Fig. 2b-d) and $P_{12\mathrm{h}}$ (Fig. 2e) despite the RE drop observed for $\overline{\theta}$ (Fig. 2a). This is due to the location of the

5    errors that are introduced with the standard ZG BC on $\theta$. As shown in Fig. 4, for simulations with higher model tops these are mainly confined to the topmost kilometer of the model domain. If $z_{\mathrm{top}}$ is set high enough these deviations therefore do not affect the cloud processes below. While the results indicate that a CG BC effectively reduces errors in $\theta$, it is found to be problematic for atmospheres with stronger stratifications. For the 1-km high and 20-km wide Witch of Agnesi ridge and a background state of RH $= 100\,\%$, $U = 20\,\mathrm{m\,s}^{-1}$ and $N \geq 0.0175\,\mathrm{s}^{-1}$, ICAR-N simulations began to exhibit numerical instabilities. These were

10    triggered by the CG BC causing the upper levels of the model domain to heat up, an issue not observed for the ZG BC (not shown).





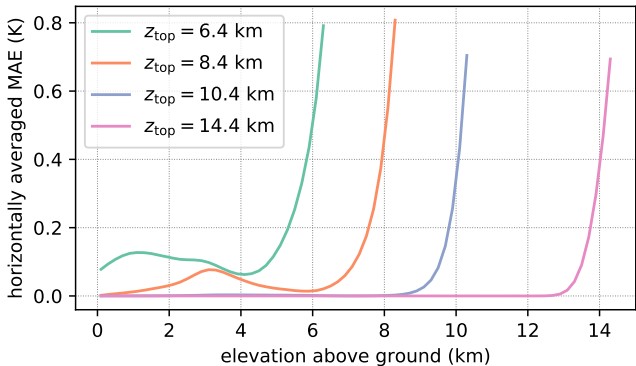

**Figure 4.** The mean absolute error (MAE) of potential temperature in ICAR-N simulations employing ZG BCs (000) with different model top settings $z_{\text{top}}$ in dependence of the elevation above ground ($x$-axis). The MAE is calculated with respect to a reference simulation with $z_{\text{top}} = 20.4\,\text{km}$ and ZG BCs (000). All simulations are conducted for the default scenario.

Figure 5a-b shows that the model top elevation necessary for a RE of $95\,\%$, $z_{\text{min}}(\psi, \text{BCs})$, is essentially constant and therefore independent of the imposed BCs for all investigated quantities except for potential temperature. Imposing a CG BC on $\theta$ at the upper boundary lowers $z_{\text{min}}(\Theta, \text{BCs})$ from $12.4\,\text{km}$ to $9.4\,\text{km}$. Similar results are found for ICAR-N simulations conducted for the other tested topographies (not shown). To reduce the parameter space in the following analysis, and since the results for

5    each BC combination are very similar, the idealized simulations from here on focus on CG BCs imposed at the model top (BC code 111).

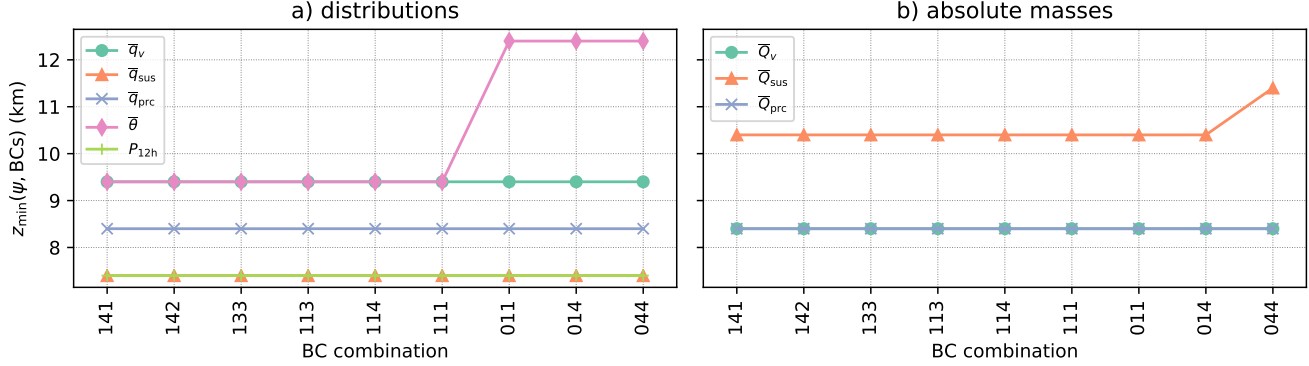

**Figure 5.** The panels show the minimum model top elevation $z_{\text{min}}(\psi, \text{BCs})$ necessary to reduce the error by $95\,\%$ for (a) water vapor $\overline{q}_v$, suspended hydrometeors $\overline{q}_{\text{sus}}$, precipitating hydrometeors $\overline{q}_{\text{prc}}$, potential temperature $\overline{\theta}$ and the 12-hour precipitation sum $P_{12h}$ and (b) the total mass of water vapor $\overline{Q}_v$, suspended hydrometeors $\overline{Q}_{\text{sus}}$ and precipitating hydrometeors $\overline{Q}_{\text{prc}}$, respectively in dependence of the set of upper boundary conditions. The ICAR-N simulations are run for the default scenario.





### 4.3  Sensitivity to the model top elevation

As shown in Fig. 6a-h, for most investigated quantities the reduction of error (RE) increases monotonously with the model top elevation $z_{\text{top}}$ for all tested topographies. Once the threshold of $95\%$ is exceeded, further increases in $z_{\text{top}}$ correspond to distinctly lower increases in RE. However, non-monotonic exceptions exist as, for instance, the total mass of water vapor $\overline{Q}_v$ shown in Fig. 6e. Here $\overline{Q}_v$ exhibits a local maximum at $z_{\text{top}} = 5.4\,\text{km}$, before dropping to lower values that eventually converge towards $\text{RE} = 1$. This is a direct consequence of the influence of the model top on the cloud processes within the domain, which for the investigated scenarios is particularly pronounced for suspended hydrometeors $q_{\text{sus}}$. For ICAR-N simulations conducted for the default scenario (BC code 111) with increasing values of $z_{\text{top}}$, Fig. 7a shows the cloud boundary of suspended hydrometeors. Here it is defined as the contour line where $q_{\text{sus}} = 10\,\text{mg}\,\text{kg}^{-1}$. While the upwind cloud adjacent to the ridge occupies a large region in the simulations with the lowest model tops, it initially shrinks with increasing $z_{\text{top}}$ until a minimum extension is reached at $z_{\text{top}} = 7.4\,\text{km}$. After this minimum the cloud increases in size with higher $z_{\text{top}}$. The extension of a smaller secondary cloud upwind of the ridge decreases in size similarly before it vanishes completely for $z_{\text{top}} \geq 8.4\,\text{km}$. Conversely, downwind of the ridge at an elevation of approximately $6\,\text{km}$ to $9\,\text{km}$ a larger cloud forms only for $z_{\text{top}} \geq 6.4$. Altogether, the total mass of suspended hydrometeors, shown in Fig. 7b, initially decreases with increasing $z_{\text{top}}$ until a local minimum at $6.4\,\text{km}$ is reached. In the simulation with this model top elevation, less water vapor is converted into suspended hydrometeors $q_{\text{sus}}$, leading to a local maximum of $\overline{Q}_v$ at $z_{\text{top}} = 6.4\,\text{km}$ (Fig. 7b). This particular behavior is found independently of the imposed boundary conditions and results in the same cloud boundaries as shown in Fig. 7a. If a different Witch of Agnesi ridge configuration is employed, the same shrinking of the $q_{\text{sus}}$ cloud occurs with increasing $z_{\text{top}}$, however, in these simulations the cloud boundaries differ from those in Fig. 7a (not shown).

The results show that the total masses of the microphysics species alone are not sufficient to determine whether the processes within the domain are influenced by the model top. In other words, the distribution of these quantities needs to be taken into account as well. Conversely, even though the error in the distribution of $\overline{q}_{\text{sus}}$ is reduced by at least $95\%$ once a model top elevation of $7.4\,\text{km}$ is employed, the same occurs for the total mass $\overline{Q}_{\text{sus}}$ only at $z_{\text{top}} = 10.4\,\text{km}$ (cf Fig. 6b, f). Therefore, both measures, the distribution of a quantity and its total mass, are necessary to reliably determine whether the cloud formation processes within the domain is independent from influences of the model top. Overall the results show that for the default scenario a lowest possible model top elevation of $Z_{\text{min}} = 10\,\text{km}$ is required for ICAR-N to represent cloud processes undisturbed from the influence of the upper boundary of the domain. Furthermore, the value of $Z_{\text{min}}$ is found to depend strongly on the particular scenario simulated, with values ranging from $8\,\text{km}$–$14\,\text{km}$.



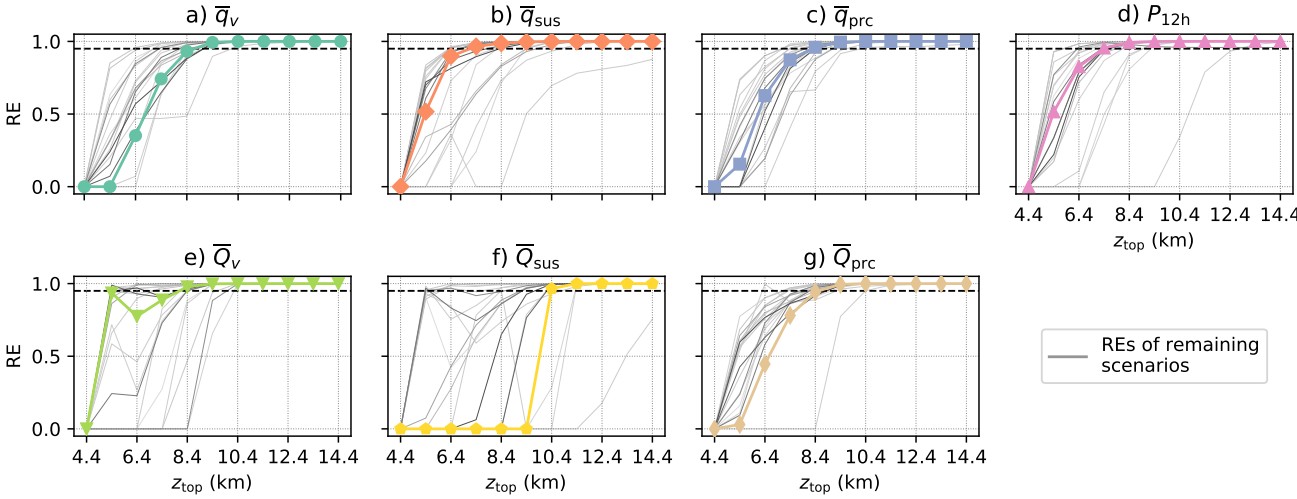

**Figure 6.** The reduction of error RE in dependence of $z_{top}$ evaluated for the time averaged distribution of (a) water vapor $\overline{q}_v$, (b) suspended hydrometeors $\overline{q}_{sus}$, (c) precipitating hydrometeors $\overline{q}_{prc}$, (d) 12-h precipitation sum $P_{12h}$ and the time averaged total masses of (e) water vapor $\overline{Q}_v$, (f) suspended hydrometeors $\overline{Q}_{sus}$ and (g) precipitating hydrometeors $\overline{Q}_{prc}$. The colored curves show RE($z_{top}$) of the respective quantity in the ICAR-N simulations conducted for the default scenario, while the gray curves indicate the RE of simulations for the other scenarios. The ICAR-N simulations imposed CG BCs on all quantities at the upper boundary (BC code 111). The black dashed line shows the 95 % RE threshold.

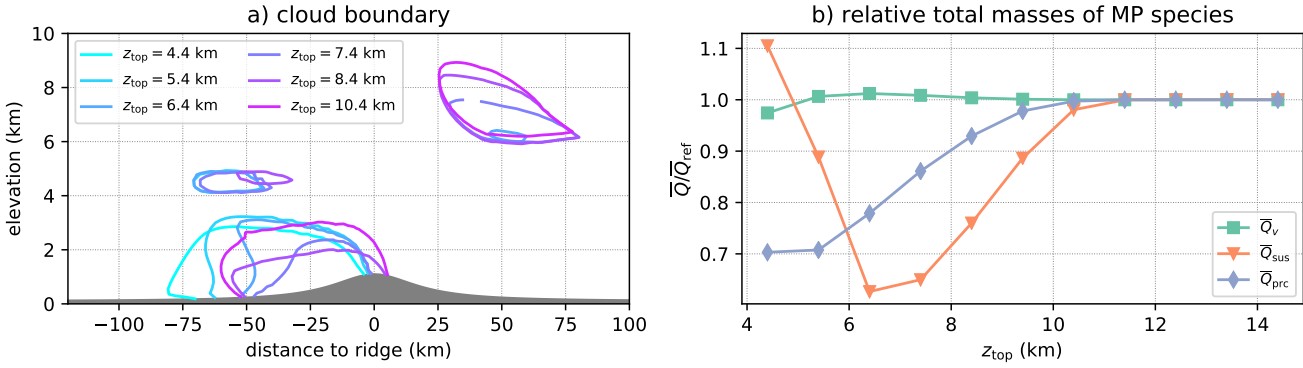

**Figure 7.** Panel (a) shows the boundary of a suspended hydrometeor cloud defined by the $q_{sus} = 10\,\mathrm{mg\,kg}^{-1}$ contour line for ICAR-N simulations with different model top elevations after 30 hours of simulation. Panel (b) shows the mean total mass of the microphysics species in ICAR-N simulations in dependence of $z_{top}$ normalized with their respective mass in a reference simulation with $z_{top} = 20.4\,\mathrm{km}$. The ICAR-N simulations are run for the default scenario with CG BCs imposed on all quantities at the upper boundary (BC code 111).





### 4.4 The lowest possible model top elevation

This section investigates how the lowest possible model top elevation $Z_{\min}$ depends on ridge height $h_m$ and width $a$, as well as the background state employed in the ICAR-N simulations. Note that $Z_{\min}$ is defined as the maximum of $z_{\min}(\psi, \mathrm{BCs})$ and thereby represents the model top elevation required for a $95\%$ reduction of error in all quantities (except $\theta$) for a given set

of boundary conditions (BC code 111 in the following). For a background state with $U = 20\,\mathrm{m\,s^{-1}}$ and $N = 0.01\,\mathrm{m\,s^{-1}}$ the results indicate a weak dependence of $Z_{\min}$ on the ridge height, with higher $Z_{\min}$ for higher ridges (Fig. 8a). The dependency of $Z_{\min}$ on the width of the ridge, on the other hand, exhibits no distinct pattern (Fig. 8b).

For a Witch of Agnesi ridge with $h_m = 1\,\mathrm{km}$ and $a = 20\,\mathrm{km}$, $Z_{\min}$ exhibits a clear dependence on the background state as

shown in Fig. 8c. In the following, the background state is characterized by the vertical wavelength of the resulting mountain wave in dry conditions, given by $\lambda_z = 2\pi U/N_d$. Note that the characteristics of the results remained unchanged (not shown) even if instead of $N_d$ the mean moist Brunt-Väisälä frequency $N_m$ in the lowest kilometer of the atmosphere (e.g., Jiang, 2003) is employed to calculate $\lambda_z$. In Fig. 8c $\lambda_z$ is varied either by keeping $N_d = 0.01\,\mathrm{s^{-1}}$ constant and varying $U$ or by fixing $U = 20\,\mathrm{m\,s^{-1}}$ and varying $N_d$. Figure 8c shows that $Z_{\min}$ decreases with increasing vertical wavelength. A potential reason

for this behavior is that lower $\lambda_z$ correspond to a higher number of periods of up- and downdrafts within the troposphere. This increases the likelihood that the model top passes through a region with convergent downdrafts and a negative vertical flux gradient $\phi_z$, thereby triggering the mass-influx mechanism outlined in Sect. 2.2.2. At high enough model top elevations all quantities (except for $\theta$) and in turn $\phi_z(\psi)$ eventually tend towards zero and any influence of the model top on the cloud and precipitation processes in the model domain becomes negligible. For longer vertical wavelengths another effect could come

into play. Here model top elevations at approximately $\lambda_z/2$ may become feasible due to the minimum of the vertical wind speeds at this height. For wavelengths larger than approximately $10\,\mathrm{km}$ the results are similar and do not depend on whether the longer wavelength is obtained by an increase in $U$ or by decreasing $N_d$ while keeping the other variable constant. However, they exhibit clear differences at shorter wavelengths. While, at shorter wavelengths, $Z_{\min}$ decreases gradually as $\lambda_z$ increases due to increasing $U$, the decrease in $Z_{\min}$ is distinctly steeper if the longer wavelength is obtained by lowering $N_d$. The majority

of the steeper decrease is explicable with the CG boundary condition chosen for $\theta$, which causes numerical instabilities for $N_d \geq 0.0175\,\mathrm{s^{-1}}$.

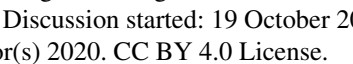





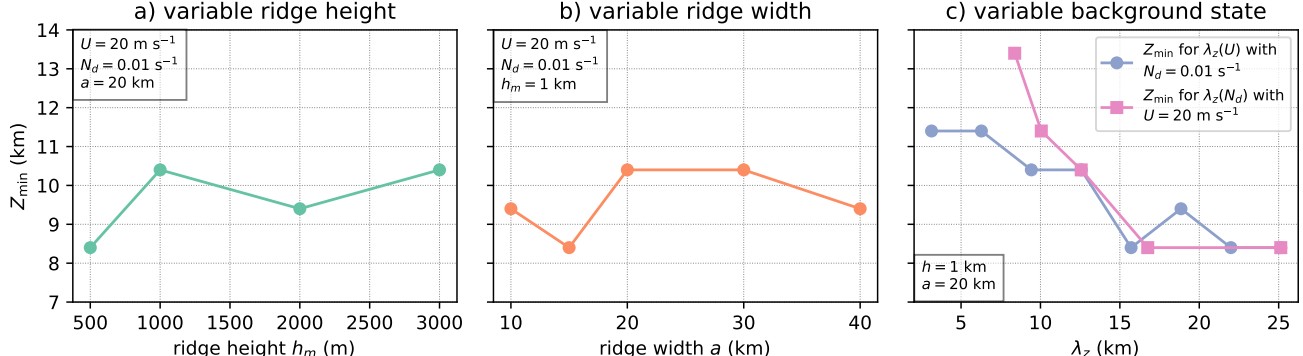

**Figure 8.** The dependence of the lowest possible model top elevation $Z_{\min}$ on (a) ridge height with constant ridge width of $20\,\mathrm{km}$, (b) ridge width with constant ridge height of $1\,\mathrm{km}$ and (c) vertical wavelength $\lambda_z$ of hydrostatic mountain waves where $\lambda_z$ is adjusted either by changing $U$ or $N_d$ for a ridge with $h_m = 1\,\mathrm{km}$ and $a = 20\,\mathrm{km}$. The ICAR-N simulations are conducted with CG BCs imposed on all quantities (BC code 111).

## 4.5 Comparison to WRF

This section compares the spatial distribution of water vapor $q_v$, suspended hydrometeors $q_{\mathrm{sus}}$, precipitating hydrometeors $q_{\mathrm{prc}}$ and 12-h sum of precipitation $P_{12h}$ calculated by ICAR-N to the corresponding fields in WRF. ICAR-N imposes CG BCs (111) and employs a model top elevation of $z_{\mathrm{top}} = 10.4\,\mathrm{km}$. This is the lowest possible model top elevation $Z_{\min}$ required for a $95\,\%$

reduction of error in all quantities for the chosen set of BCs determined for the default scenario. The distributions of $q_v$, $q_{\mathrm{sus}}$ and $q_{\mathrm{prc}}$ are investigated after 30 hours of simulation time, while $P_{12h}$ is investigated between 19 and 30 hours of simulation time. The comparison aims to highlight the differences that may be expected between an ICAR-N and WRF simulation due to the tradeoff between physical fidelity and model performance. The scenario is chosen such that the wind field is expected to exhibit non-linearities.

### 4.5.1 Water vapor and hydrometeors

With respect to water vapor ICAR-N is drier upwind of the topographical ridge and wetter downwind in comparison to WRF (see Fig. 9a-c). The regions with this dry and wet bias extend up to an elevation of approximately $6\,\mathrm{km}$ in which, farther upwind of the ridge, WRF exhibits slightly stronger updrafts than ICAR-N (Fig. 10c and d). Similarly, above the ridge the downdrafts calculated by WRF are of a higher magnitude than those predicted by ICAR-N, see Fig. 10c and d. Therefore, upwind of the

ridge WRF transports more moist air from close to the surface to higher elevations. Above the ridge, on the other hand, WRF advects drier air from higher elevations to lower levels. Hence, the two large regions in ICAR-N exhibiting a dry and wet bias in $q_v$ respectively are likely caused by the differences in the wind field. However, a wet bias close to the mountain slope on the windward side is presumably caused by microphysical conversion processes (Fig. 10c). Here the stronger orographic lifting





in WRF leads to a higher microphysical conversion rate of $q_v$ to hydrometeors, thereby resulting in the observed wet bias of ICAR-N in terms of $q_v$. Above the downwind slope of the ridge and up to approximately $100\,\mathrm{km}$ downwind, the downdrafts in WRF are still stronger than in ICAR-N. This potentially causes an increased conversion of hydrometeors to $q_v$ by evaporation, resulting in the dry bias of ICAR-N in this region.

Clear differences between the ICAR-N and WRF simulations are observed for suspended hydrometeors. While the approximate shape of the windward cap cloud (Fig. 9d and e) shows similarities, the mixing ratios calculated by ICAR-N are approximately one tenth of those in WRF (see Fig. 9f). Furthermore, the main constituent of the cap cloud in ICAR-N is ice $q_i$, while it is liquid water $q_c$ in WRF (not shown).

The majority of precipitating hydrometeors in ICAR-N are observed windward of the topographical ridge, extending over most of the upwind slope (Fig. 9g). In WRF, on the other hand, the distribution of $q_{\mathrm{prc}}$ is centered above the ridge and extends farther downwind than upwind (Fig. 9h). In both models the majority of $q_{\mathrm{prc}}$ consists of snow $q_s$ (not shown). However, WRF additionally predicts non-negligible amounts of graupel $q_g$ up to $20\,\mathrm{km}$ upwind of the ridge (not shown). Altogether, for precip-

15  itating hydrometeors (Fig. 9i) ICAR-N is wetter on the windward slope but drier above the ridge and the downwind slope. This is caused by a combination of two factors: (i) The higher vertical wind speeds above the windward slope of the topographical ridge predicted by WRF, lead to lower effective falls speeds of the hydrometeors (see Fig.10b). (ii) Higher horizontal wind speeds additionally contribute to a larger horizontal drift of $q_{\mathrm{prc}}$ and precipitation spill-over in WRF (see Fig.10c and, for a basic estimation of the drift distances, Sect. 4.5.2).



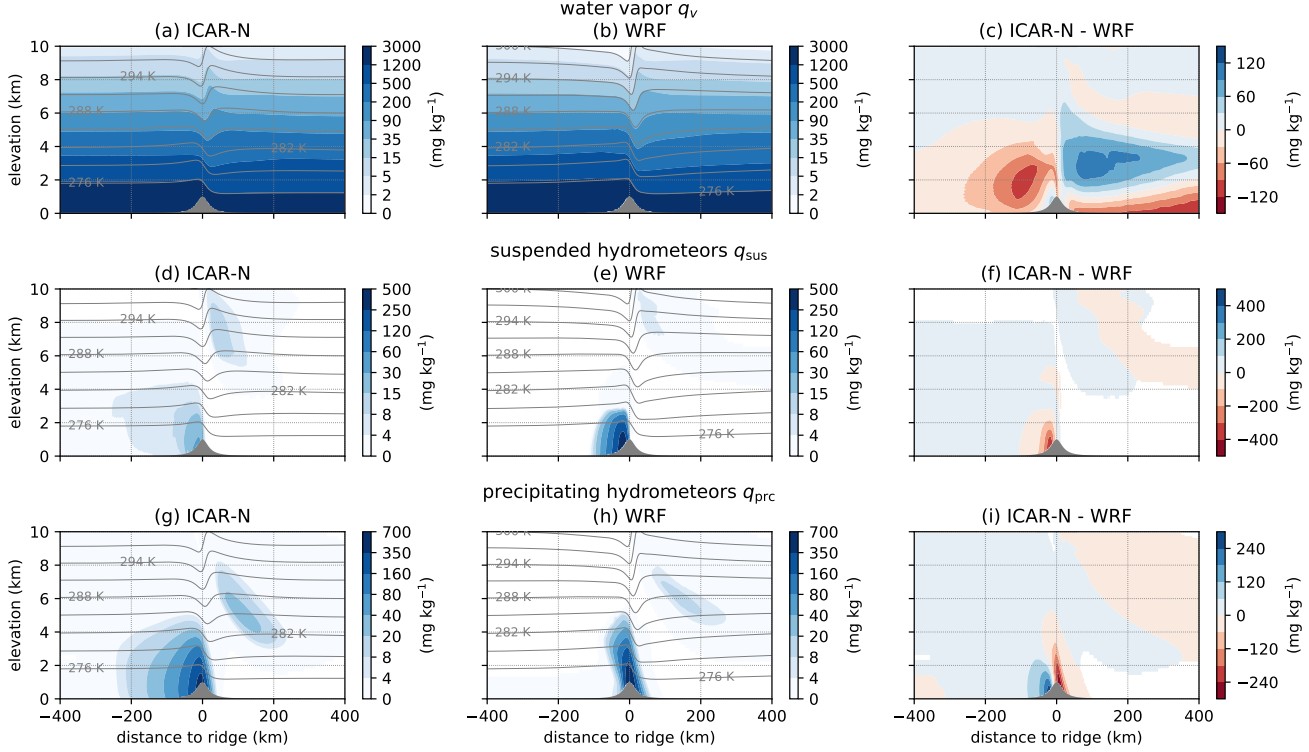

**Figure 9.** Mixing ratios (color contours) of water vapor (top row), suspended hydrometeors (middle row) and precipitating hydrometeors (bottom row) calculated with ICAR-N (left column), WRF (center column) and the difference between ICAR-N and WRF (right column) after 30 hours of simulation. The isentropes of ICAR-N and WRF are shown as gray contour lines with $3\,\mathrm{K}$ increments. The direction of the background flow is from left to right. Note that the scaling of the contours for all quantities is non-linear to reveal details in the respective distributions. ICAR-N and WRF simulations are conducted for the default scenario with ICAR-N imposing CG BCs on all quantities at the upper boundary (BC code 111).

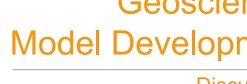
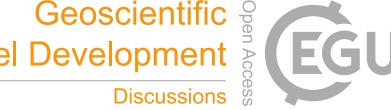

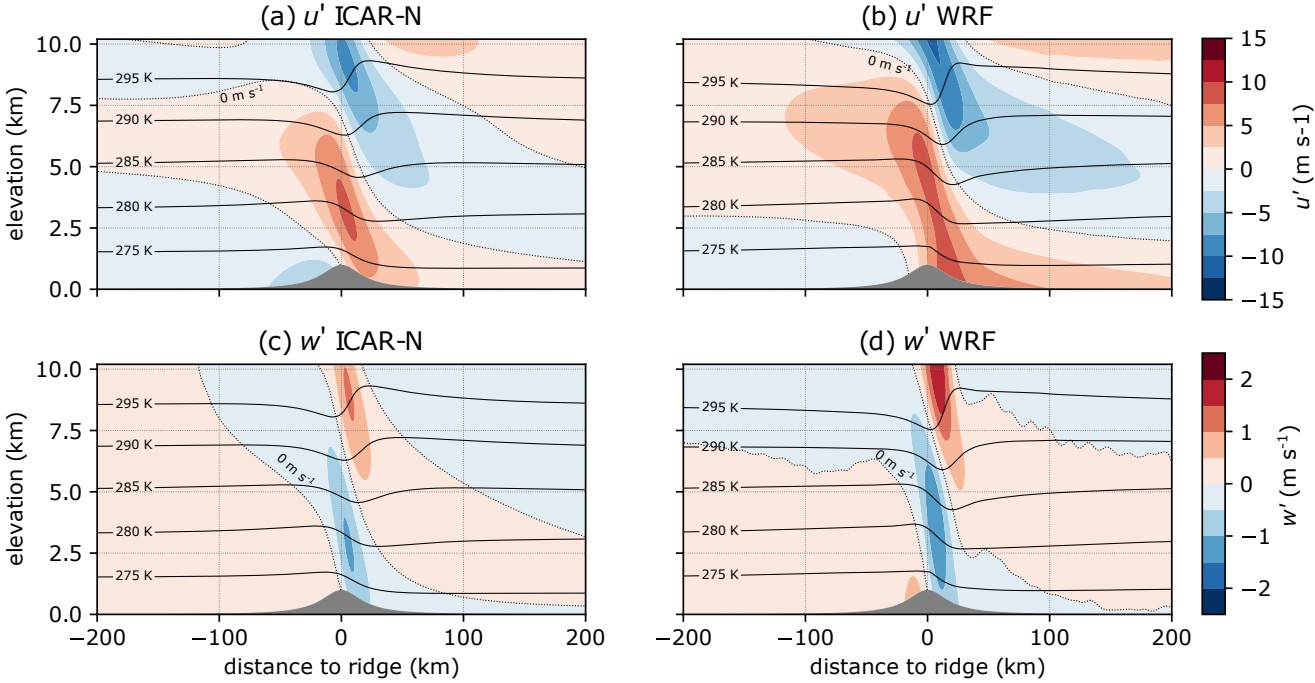

**Figure 10.** Perturbations of the horizontal wind component $u'$ (top row) and vertical wind component $w'$ (bottom row) calculated by ICAR-N with $z_{\text{top}} = 10.4\,\text{km}$ (left column) and WRF (right column). The dotted curve shows the $0\,\text{m}\,\text{s}^{-1}$ countour line and the black lines indicate the isentropes. Both simulations are run for the default scenario with ICAR-N imposing CG BCs on all quantities at the upper boundary (BC code 111).

#### 4.5.2 Precipitation

Figure 11a illustrates that $P_{12h}$ on the windward slope is substantially higher in ICAR-N than in WRF. Conversely, ICAR-N is drier along the leeward slope. Both observations correspond well to the distribution and shape of the precipitating hydrometeors close to the surface (see Fig. 9g and h) and the differences of $q_{\text{prc}}$ between ICAR-N and WRF (see Fig. 9i). The precipitation

5  maximum predicted by ICAR-N is approximately $25\,\text{mm}$ and lies $6\,\text{km}$ upwind of the ridge peak in comparison to the $32\,\text{mm}$ maximum in WRF, which lies $4\,\text{km}$ upwind of the ridge. The median of $P_{12h}$, however, is located upwind of the ridge peak in ICAR-N and downwind in WRF, separated by a distance of $20\,\text{km}$ (see Fig. 11b). Integration along the cross section shows that $63\,\%$ of ICAR-N precipitation falls out upwind of the domain center while for WRF, on the other hand, it is only $43\,\%$.

10  The distribution of precipitation in ICAR-N is asymmetric with a gradual increase until the maximum is reached and a steeper decrease after that. While in WRF $P_{12h}$ is asymmetric as well, the distribution exhibits a very steep increasing slope ending in a





distinct peak that is followed by a decreasing slope comparable to the decrease of $P_{12h}$ in ICAR-N. In WRF snow and graupel contribute to $P_{12h}$, while the precipitation in ICAR-N is solely composed of snow. The graupel shower predicted by WRF is localized within a $30\,\text{km}$ region centered approximately $10\,\text{km}$ upwind of the ridge and causes the distinct peak observed in the distribution of precipitation in WRF (Fig. 11a).

The maximum of accumulated snow in WRF is $48\,\text{mm}$ and the median of the distribution is shifted downstream by $22\,\text{km}$ in relation to the median of the precipitation distribution in ICAR-N, which is solely snow. The difference is mainly due to the different wind fields of ICAR-N and WRF. In the following a fall speed for snow in stagnant air of $-1\,\text{m}\,\text{s}^{-1}$ is assumed for the ICAR-N and WRF simulations alike. Starting $1\,\text{km}$ above the orography, the effective fall speeds in ICAR-N and WRF

10 are $-0.75\,\text{m}\,\text{s}^{-1}$ and $-0.25\,\text{m}\,\text{s}^{-1}$ respectively, based on an average $w'$ above the upwind slope of the ridge of $0.25\,\text{m}\,\text{s}^{-1}$ in ICAR-N and $0.75\,\text{m}\,\text{s}^{-1}$ in WRF (see Fig. 10c-d). In combination with an approximate average horizontal wind speed of $17.5\,\text{m}\,\text{s}^{-1}$ in ICAR-N and $21\,\text{m}\,\text{s}^{-1}$ in WRF (Fig. 10a-b) this results in a difference in the resulting horizontal drift of $19\,\text{km}$, which fits the observed difference in the medians of the accumulated snow precipitation distribution well. Hence, the discrepancy in the precipitation distribution appears to be mainly caused by an underestimation of the perturbation velocities in ICAR.

The absence of graupel in ICAR-N compared to WRF can be traced to the MP scheme and is a result of the atmospheric conditions it encounters. The Thompson MP predicts graupel formation if riming growth exceeds the depositional growth of snow (Thompson et al., 2004). While the necessary atmospheric conditions are easily satisfied in WRF, the cloud water mixing ratio in ICAR-N is too low to initiate sufficient riming growth (see Fig. 9d). However, no clear indication for the underlying

20 cause of the large difference in the cloud water mixing ratios between ICAR-N and WRF is found.





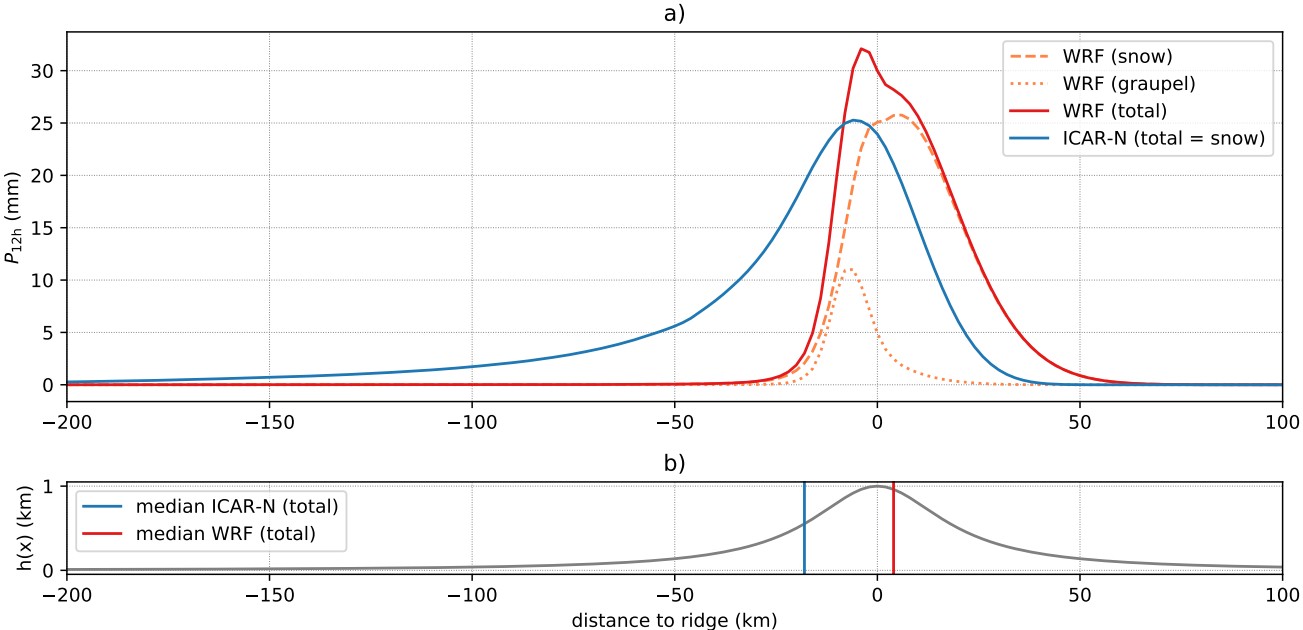

**Figure 11.** (a) 12-h accumulated total precipitation $P_{12h}$ along the cross-section for ICAR-N (solid blue curve) and WRF (solid red curve). Additional curves indicate the contribution of graupel (dotted orange curve) and snow (dashed orange curve) to the total precipitation of WRF. ICAR-N total precipitation consists solely of snow, i.e. rain and graupel are zero in this specific simulation. (b) topography along the cross-section with vertical blue and red lines indicating the locations of the medians of the total precipitation distribution of ICAR-N and WRF respectively. Both models are run for the default scenario while ICAR-N imposes CG BCs on all quantities at the upper boundary (BC code 111).

## 4.6   Case study

The previous sections have demonstrated that (i) the Brunt-Vaisälä frequency needs to be diagnosed from the background stratification in order to model a realistic perturbation flow field with ICAR, that (ii) it further requires a minimum model top elevation (which is dependent on the orography and the atmospheric background state) and that (iii) a combination of ZG/CG
5   BCs (BC codes 011 and 111) are optimal to be used at the top of the ICAR model domain. The effects of these suggested modifications to ICAR on a real world application are investigated with a case study conducted for the Southern Alps on the South Island of New Zealand located in the southwestern Pacific Ocean (Fig. 12a).

The Southern Alps are a mountain range approximately $800\,\mathrm{km}$ long and $60\,\mathrm{km}$ wide. They are oriented southwest-northeast
10   and extend from approximately $41^\circ\,\mathrm{S}$ to $46^\circ\,\mathrm{S}$, with approximately $97\,\%$ of the crest line lying above an elevation of $1500\,\mathrm{m}$ m.s.l. (meters above mean sea level) and the highest peaks rising above $3000\,\mathrm{m}$ m.s.l.. The mean precipitation regime in the humid





and maritime climate on the South Island of New Zealand is strongly influenced by the orography of the Southern Alps. The prevailing westerly and north-westerly winds advect moist air against the topographic barrier, leading to a precipitation maximum of approximately $14\,\mathrm{m\,yr^{-1}}$ along its western flanks in close proximity to the alpine ridge. While the western coast on average receives $5\,\mathrm{m\,yr^{-1}}$, the plains east of the alpine ridge receive at most $1\,\mathrm{m\,yr^{-1}}$ due to the precipitation shadow of the Southern Alps (Griffiths and McSaveney, 1983; Henderson and Thompson, 1999).

For this region ICAR-O and ICAR-N simulations are conducted. ICAR-O calculates the Brunt-Väisälä frequency $N$ based on the perturbed state of the atmosphere and imposes ZG BCs to all quantities (BC code 000). The model top is set to $4.4\,\mathrm{km}$, the elevation determined as optimal in Horak et al. (2019) by comparing 24-h accumulated precipitation to observations. ICAR-N, on the other hand, calculates $N$ from the forcing data set and imposes a zero gradient BC on the potential temperature field and constant gradient BCs on the microphysics species (BC code 011). The lowest possible model top elevation $Z_{\mathrm{min}}$ with an acceptably low error is determined by applying the method outlined in Sect. 3.5 based on multiple ICAR-N simulations with model top elevations between $5\,\mathrm{km}$–$20\,\mathrm{km}$ (Fig. 13). The resulting value of $Z_{\mathrm{min}}$ is found at $15.2\,\mathrm{km}$, which is in stark contrast to the value of $4.4\,\mathrm{km}$ in Horak et al. (2019). This indicates that determining the optimal model top elevation solely by comparing simulation output to measurements may lead to an incorrect result. The cloud formation processes in the ICAR-O simulation with the low model top elevation are likely unphysical and strongly disturbed by the model top.



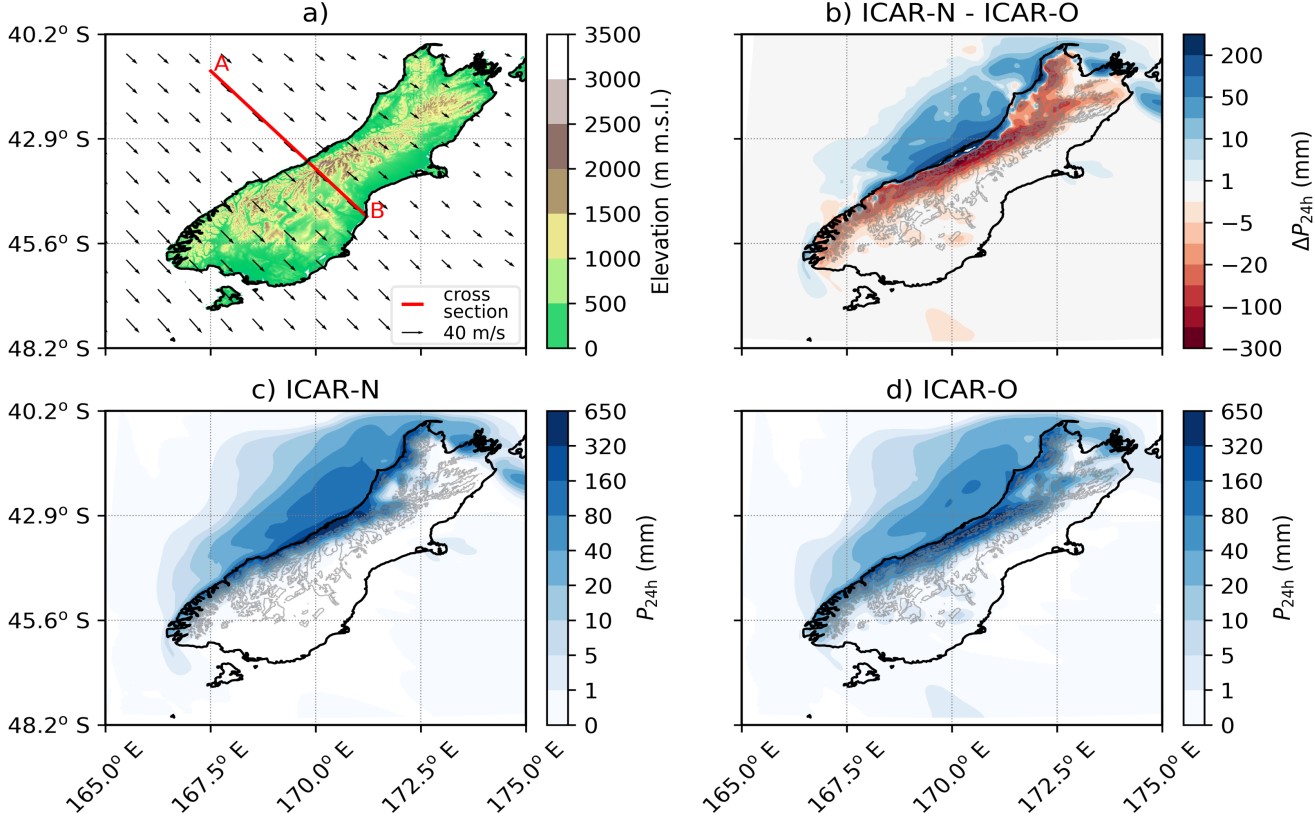

**Figure 12.** (a) The South Island of New Zealand study domain with the horizontal wind field at the $500\,\mathrm{hPa}$ level and the location of the vertical cross section (red line), (b) difference in 24-h accumulated precipitation $P_{24h}$ between ICAR-N and ICAR-O, (c) $P_{24h}$ pattern for ICAR-O with $z_{top} = 4.4\,\mathrm{km}$ imposing ZG BCs (BC code 000) and (b) $P_{24h}$ pattern for ICAR-N with $z_{top} = 15.2\,\mathrm{km}$ and a ZG BC imposed on $\theta$ and CG BCs imposed on the MP species (BC code 011) on the 6 May 2015 LT. Panels (b)-(d) additionally show the $1000\,\mathrm{m}$ m.s.l. contour line of the topography.

The resulting patterns of $P_{24h}$ for ICAR-O and ICAR-N on the South Island of New Zealand are shown in Fig. 12c and Fig. 12d, respectively while their difference is shown in Figure 12b. Overall the maximum amount of precipitation and the approximate distribution are similar for ICAR-N and ICAR-O. However, ICAR-N is clearly dryer in regions above $1000\,\mathrm{m}$ m.s.l. and downwind of the alpine range. Conversely, ICAR-N generates the majority of its precipitation in close proximity to the coast and is wetter in the regions upwind of the western slopes of the Southern Alps. The reason for ICAR-O to producing precipitation further downwind than ICAR-N can be found in the cross-sections of hydrometeor distributions shown in Fig. 14.





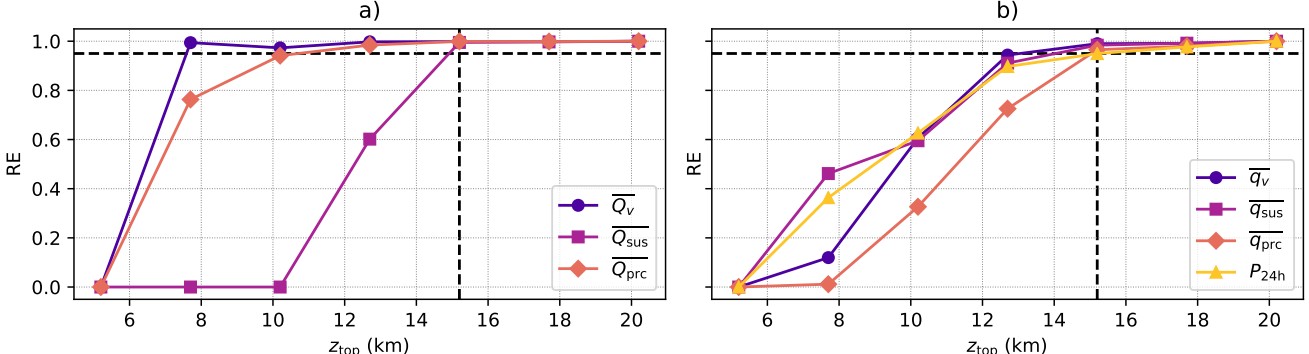

**Figure 13.** The reduction of error RE of the simulations for the South Island of New Zealand for (a) the total mass of the MP species in the domain and (b) the distribution of the MP species and precipitation in dependence of the model top elevation $z_{top}$. ICAR-N imposes a ZG BC on the potential temperature field and constant gradient BCs on the microphysics species (011). The dashed horizontal line indicates the $95\%$ RE threshold used to determine $Z_{min}$ and the dashed vertical line shows at which model top this threshold is exceeded for all quantities.

Clear differences can be observed in the distributions of $\overline{q}_{sus}$ (Fig. 14a and b) - note, e.g., the distinct maximum of $\overline{q}_{sus}$ above the initial topography peak in ICAR-O which is almost entirely absent in ICAR-N. These $\overline{q}_{sus}$ maxima occur in the topmost levels of the ICAR-O domain and suggest that the ZG BC overestimates the moisture content of the atmospheric column and artificially introduces additional water in the domain (as outlined in Sect. 2.2.2). This leads to the formation of artificial clouds

downwind of approximately $169.8°$ E. Note that in ICAR-N (Fig. 14b) the cloud formation is confined to a region upwind of $169.8°$ E.

Furthermore, this artificial cloud in ICAR-O near the model top generates precipitating hydrometeors that extend farther to the lee of the alpine crest compared to ICAR-N (Fig. 14c and d). While ICAR-N produces more precipitation overall and is

10 wetter than ICAR-O on the initial ramp of the western slope of the alpine range (up to approximately $169.8°$ E in Fig. 14f), ICAR-O is wetter downwind, yielding higher amounts of precipitation at the peak and the first leeward slope (Fig. 14e). Note that ICAR-O produces clouds in the topmost model levels even farther downstream as well (Fig. 14a), however, they do not generate precipitating hydrometeors during the investigated period. These results strongly indicate that the low model top setting of $4.4\,\mathrm{km}$ employed in Horak et al. (2019) is inadequate to allow for a correct representation of the cloud and precipitation

processes within the domain despite the relatively high skill found for ICAR-O in their study. Therefore, the results additionally demonstrate that when model skill is evaluated with statistical metrics based on surface observations alone (Horak et al., 2019), it does not necessarily reflect the skill of the model in correctly representing atmospheric processes such as gravity waves and associated cloud formation. Hence, it seems that the underestimation in precipitation near the crest and to its lee of an ICAR simulation with reasonably high model top compared to WRF (Fig. 9) is partly compensated in an ICAR simulation with a





too low model top (ICAR-O in Fig. 14) by spurious effects introduced by the upper boundary conditions. It follows that the seeming improvement in the latter case is right but for the wrong reason.

**Figure 14.** Cross-sections along the South Island of New Zealand (line A-B in Fig. 12a) for an ICAR-O simulation ($z_{top} = 4.0$km, BCs 000, left column) and an ICAR-N simulation ($z_{top} = 15.2$km, BCs 011, right column). The panels show the 24-h averaged mixing ratio of suspended hydrometeors $\overline{q}_{sus}$ (top row), precipitating hydrometeors $\overline{q}_{prc}$ (middle row) and the 24-h accumulated precipitation as well as the difference in precipitation between ICAR-N and ICAR-O (bottom row).





## 5   Discussion

The results highlight that a more accurate representation of the wind fields is obtained only when the Brunt-Väisälä frequency, in accordance with linear mountain wave theory, is calculated from the unperturbed background state of the atmosphere (ICAR-N) rather than from the perturbed state (ICAR-O). The remaining differences of the wind fields in ICAR-N to the analytical

solution may be attributable to two causes: Firstly, to solve the governing equations ICAR numerically calculates the Fourier Transform of the topography $h(x,y)$ in the domain. In cases where $h(x,y)$ is not constant along the domain boundaries or where it exhibits discontinuities within the domain, this approach gives rise to numerical artifacts (see the Gibbs phenomenon, e.g., Arfken et al., 2013), introducing errors into the perturbed fields. Note that for a 2-D ridge as employed in this study $h(x,y) = h(x)$. Therefore, while $h(x_w) = h(x_e) = $ const, with $x_w$ and $x_e$ the $x$-coordinate of the western and eastern domain

boundary, respectively, $h(x) \neq$ const along the northern or southern domain boundary. This results in an average value of $v'$ of $0.02\,\mathrm{m\,s}^{-1}$ instead of the expected $0\,\mathrm{m\,s}^{-1}$ and therefore slightly altered values of $u'$ and $w'$ in comparison to the results from linear theory. These issues may be reduced by, for instance, filtering the topography accordingly or by adding a buffer around the domain (Florinsky, 2016). Additional research is necessary to determine which filtering methods or modifications to the topography are best suited to preprocess digital elevation models for ICAR. Secondly, ICAR does not solve for $w'$ directly but

only analytically calculates $u'$ and $v'$. The vertical perturbation is then determined by balancing the density-weighted horizontal winds from the continuity equation (Gutmann et al., 2016), starting at the lowest vertical level.

ICAR is intended as an computationally frugal alternative to full physics models, in principle allowing for very low model top elevations. While employing a low model top to take advantage of the associated computational cheapness is tempting,

increased efficiency should not come at the cost of the physical fidelity of the model. The results in this study clearly show that there is a lowest possible model top elevation $Z_{\min}$ that ensures that the physical processes within the domain are not influenced by the model top. Boundary conditions imposed at the upper boundary are found not to influence the value of $Z_{\min}$ for the investigated parameter space despite potentially mitigating errors in the potential temperature and water vapor fields. In particular, the cloud formation and precipitation processes within the domain are shown to almost exclusively depend on the

model top elevation $z_{\mathrm{top}}$ and not on the chosen set of boundary conditions, and only stabilize for $z_{\mathrm{top}} \geq Z_{\min}$. It seems unlikely that any boundary condition is able to accurately represent the effect of cloud and precipitation processes above the model domain and the resulting interaction with the corresponding processes in the model domain (e.g. the seeder-feeder mechanism). Therefore, in order to capture all relevant cloud and precipitation processes, the vertical extension of the domain should at the very least encompass the entire troposphere. Altogether these results highlight that model top elevations within the troposphere

as employed by past studies are to be avoided (e.g., Gutmann et al., 2016; Horak et al., 2019; Alonso-Gónzalez et al., 2020).

This study strongly suggests that no general value for $Z_{\min}$ is applicable to all possible scenarios with the results exhibiting large differences between the idealized simulations and the real case study. For the tested parameter space, including the real case, $Z_{\min}$ mainly depends on the background state and the height of the topography. The dependence on the background





state, characterized by the vertical wavelength $\lambda_z = 2\pi U/N_d$ of the hydrostatic mountain wave, shows that overall larger $\lambda_z$ result in smaller $Z_{\min}$ and, conversely, smaller $\lambda_z$ in larger $Z_{\min}$. The dependence of $Z_{\min}$ on the background state is explicable with the horizontal wind speed $U$ and the Brunt-Väisälä frequency $N$ affecting the location, amount and magnitude of the up- and downdrafts in the domain. Similarly, $Z_{\min}$ depends on ridge height due to the generally stronger up- and downdrafts

triggered by higher topographies (Eq. (5)). However, note that the dependence on the ridge height is weak compared to the dependence on the background state.

The determination of $Z_{\min}$ considers all MP species with respect to their time averaged spatial distribution and the time averaged total mass within the cross-section as well as the 12-hour ($P_{12\text{h}}$, idealized simulations) or 24-hour ($P_{24\text{h}}$, real case

simulations) precipitation sum along the cross-section. Note that potential temperature $\theta$ is indirectly included in determining $Z_{\min}$ since errors in the $\theta$ field influence the cloud formation and precipitation processes. However, this study shows that errors in the $\theta$ field introduced by the zero gradient boundary condition are mainly localized in the topmost vertical levels (Fig. 4), which correspond to approximately the uppermost $1$ to $2\,\text{km}$ of the domain, and result in only a negligible influence on cloud formation processes in the tested parameter space. While a constant gradient boundary condition reduces the errors in

the potential temperature field, the default zero gradient boundary condition is a suitable alternative for $\theta$ provided $z_{\text{top}}$ is high enough. This can be ensured by, for instance, employing the method to determine $Z_{\min}$ described in this study.

A comparison between ICAR-N and WRF simulations conducted for the same topography and sounding reveals substantial differences in the spatial distributions of $q_v$, $q_{\text{sus}}$ and $q_{\text{prc}}$ as well as the resulting $P_{12\text{h}}$. These differences are mainly attributable

to additional effects included in the WRF but not the ICAR-N wind field, such as non-linearities and the amplification of the perturbations due to the density decreasing with height. As a consequence both models predict distinctly different events to occur: A snow shower with the majority of snow falling upwind of the ridge in ICAR-N and a snow and graupel shower in WRF with the largest portion precipitating leeward of the ridge. While these results are obtained for one particular sounding they indicate that the linearisation of the wind field has the potential to significantly alter the distribution of precipitation in a

study domain. This could have drastic consequences for the results of studies relying on ICAR to provide precipitation fields for, i.e. applications in hydrology or glaciology.

For strongly stratified atmospheric conditions, a constant gradient BC was found to cause numerical stability issues in the idealized and real case simulations alike. Future studies could investigate further BC options that might allow a better approx-

imation of the potential temperature profile: Such approaches might, for instance, (i) analytically diagnose $\theta$ for the vertical level above the model top and then apply the corresponding values as a Dirichlet BC or (ii) prescribe the potential temperature from the corresponding height in the forcing data set as Dirichlet BC at the model top in ICAR.

The case study investigates the effect of the proposed modifications to ICAR on a real world application for the South Island

of New Zealand. It reveals that these modifications shift the distribution of precipitation upwind, leading to dryer conditions





in the alpine range but wetter coastal regions. The method for the determination of $Z_{min}$ presented in this study does not rely on tuning to measurements and may therefore be employed for every region in the world for which a suitable digital elevation model and atmospheric forcing data are available. Furthermore, the method ensures that for $z_{top} = Z_{min}$ the cloud formation processes within the domain are independent from influences of the model top and that only the absolutely necessary amount

of vertical levels is used in the simulations. This preserves as much of the computational efficiency of ICAR as possible without sacrificing additional physical fidelity. However, the extension of the method to determine $Z_{min}$ to longer study periods, compared to the 24 hours of the case study, and a larger variety of background states is not trivial and outside the scope of this study. If a substantial amount of simulations for different background states is required to determine $Z_{min}$ the associated computational cost may outweigh the gain of employing the lowest possible number of vertical levels for the entire study

period. Therefore, future research could investigate variations of the $Z_{min}$ determination employed in this study. For instance, a focus on the background states most frequent during each season, or on background states with shorter vertical wavelengths (resulting in higher values of $Z_{min}$) to find upper bounds for $Z_{min}$ may drastically reduce the required number of simulations.

With regards to the case study, the unmodified version of ICAR (ICAR-O) is found to produce enhanced precipitation in

the alpine range due to artifacts (heightened mixing ratios of hydrometeors) in the topmost vertical levels in the horizontal vicinity of topographical peaks. This additionally caused the very low model top elevation found with the method employed in Horak et al. (2019): At each alpine weather station on the South Island of New Zealand Horak et al. (2019) calculated a mean squared error (MSE) between the simulated and measured precipitation accumulated over $24\,h$ ($P_{24h}$) at alpine sites. The artifacts in the topmost vertical levels of ICAR-O (with $z_{top} = 4.4\,km$) lead to an increase in precipitation at these alpine

sites in comparison to ICAR-N or, as noted by Horak et al. (2019), to ICAR-O simulations with higher model top elevations. Since all ICAR-O simulations generally underestimate precipitation amounts at alpine weather stations on the South Island of New Zealand, and overshooting of measured values does mostly not occur, the higher amounts of $P_{24h}$ for the simulation with $z_{top} = 4.4\,km$ then lowered the calculated MSE. Even though the atmospheric processes in the ICAR-N simulation are more correctly represented in comparison to ICAR-O, the lower amount of $P_{24h}$ at the alpine sites would result in a higher MSE.

Therefore, even though the calculated MSEs were lowest for a model top setting at $4\,km$, the seemingly correct results were produced for the wrong reasons. This additionally exemplifies why a comparisons to measurements alone cannot determine whether the model results are correct for the correct reason, only a detailed consideration of the underlying processes can be the basis fur such a conclusion.

## 6  Conclusions

The key findings and recommendations based on the extensive process-based evaluation of ICAR are summarized in the follwing:





- There is a minimum possible model top elevation $Z_{\min}$ to produces physically meaningful results with ICAR. If the model top elevation is lower, cloud formation and precipitation processes within the domain are affected by the model top.

- Results show that, in order to avoid spurious influences of the upper boundary to the microphysical processes within the domain, $Z_{\min}$ should be at least as high as the tropopause but may be required even higher in other situations.

- Determining an exact value for $Z_{\min}$ from comparisons to precipitation measurements may yield results in closer agreement to these measurements but potentially for the wrong reasons (i.e., model artifacts).

- In a proof of concept, the method described in this study to determine $Z_{\min}$ is applied to idealized simulations and a real case alike.

- While most of the tested boundary conditions (in comparison to the default zero gradient boundary condition) are suitable to reduce the errors in the water vapor and potential temperature fields, no tested combination of these boundary conditions can achieve a lower value for $Z_{\min}$.

- Model skill, when inferred only from comparisons to surface observations, does not necessarily reflect the model skill in representing atmospheric processes.

- The representation of the wind field in ICAR is improved by ensuring that the Brunt-Väisälä frequency is calculated from the background state of the atmosphere provided by the forcing data. Note that the current version of ICAR employs the perturbed state of the domain.

This study highlights the importance of a process-based in-depth evaluation not only with respect to ICAR but for models in general. Particularly for regional climate models (RCMs) and numerical weather prediction (NWP) models, the results of
20 the case study demonstrate a potential pitfall when model parameters are inferred solely from comparisons to measurements, potentially leading to situations for which model results are more prone to be right but for the wrong reasons. With the increasing complexity of RCMs and NWPs, ICAR could provide a computationally frugal framework to study and better understand singular model components. This would allow for a process-based evaluation of, e.g., MP schemes or advection schemes, contributing to the development and improvement of RCMs and NWPs.

25 *Code and data availability.* The modified version ICAR v1.0.1 employed for the simulations (Gutmann et al., 2020) as well as the results obtained (Horak, 2020) are available as download from the respective zenodo repositories.

*Author contributions.* The investigation and its design, the simulations and their analysis as well as the visualization of the results and writing (original draft and editing) were carried out by JH. The conceptualization of the paper was a joint effort from all authors, as were the discussion and refinement of the methods presented. Additionally, funding acquisition and project administration were carried out by MH.



*Competing interests.* The authors declare no competing interests.

*Acknowledgements.* The research presented has been funded by the Austrian Science Fund (FWF) grant 28006-N32. The computational results have been achieved with the high-performance computing support from Cheyenne (doi:10.5065/D6RX99HX) provided by NCAR's Computational and Information Systems Laboratory, sponsored by the National Science Foundation and with the HPC infrastructure LEO
5 of the University of Innsbruck. The following open-source libraries were employed to perform the data processing and analysis presented in this study: `numpy` (van der Walt et al., 2011), `pandas` (McKinney, 2010), `xarray` (Hoyer and Hamman, 2017), `matplotlib` (Hunter, 2007). The authors thank Lukas Umek for his contribution to finding the best dampening layer configuration in WRF.





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
