# Peer review of "A process-based evaluation of the Intermediate Complexity Atmospheric Research Model (ICAR) 1.0.1"

_Geoscientific Model Development, 2020_

## Referee Comment (RC1) · Anonymous Referee #1 · 23 Nov 2020

Note: comments are structured "[P]age #:[L]ine #:"

General comments:

In this paper, the authors present a detailed investigation of the upper boundary within the ICAR model, as well as a well-supported correction to the calculation of the Brunt-Väisälä frequency for linear theory calculations. They perform rigorous comparisons of their improved model (ICAR-N) with the old model (ICAR-O), the analytic solution to the underlying equations of ICAR, and to a more complex atmospheric model, WRF. The questions posed, methodologies employed to answer them, and final conclusions reached, are of value to the development community of ICAR. In the test case, these

results support their conclusion that some model top height exists which reduces errors to the advected quantities and microphysics of the model which maximizing computational efficiency. However, after testing a number of boundary conditions (BC) for the upper boundary, they fail to provide a clear recommendation for which combination of upper BC is most favorable. Such a determination would surely strengthen their conclusions, and be of great value to the community using ICAR. The demonstrated lack of dependence between minimum model top height (Zmin) and combination of BCs seems to contradict the hypothesis that Zmin is chosen to avoid errors in the assumed downward fluxes. The authors should explain this discrepancy and, if possible, provide evidence in support of a combination of upper BCs to be used as default in ICAR going forward.

Specific comments:

P7:L10-11: Why use a constant dz spacing for ICAR, but not for WRF? ICAR v1 supports this.

P7:L23-24: How did you test Thompson MP code to see that ICAR and WRF produce the same results? This sounds like you have ICAR producing identical output to WRF. This would be exceptional, but unlikely.

Section 2.2.2 – Please also state explicitly that upward fluxes result in quantities being lost (no longer tracked) by ICAR. This would motivate the use of a downward flux BC that seeks to balance this, and may explain why the current downward flux BC in ICAR does not produce drastically unrealistic simulations on first pass.

Section 3.4 – I could use some discussion of the different BC's, what they try to represent, and why you chose them. Not too much, perhaps just a sentence or two for BC's 1-4. Especially prior to where you refer to them on P9:L26.

Section 4.1 – Do you think ICAR should be considering wave amplification as a result of decreased density with height? This seems to be the cause of a major difference

between ICAR-N, the analytical solution, and WRF. EQ7 seems to suggest that such a correction is not too difficult to implement, but this may just appear deceptively simple.

P18: L6-7: "If ztop is set high enough these deviations therefore do not affect the cloud processes below" Would you expect these deviations to be advected down into the model where cloud processes occur? Please provide some discussion for why the errors in potential temperature remain in the upper most layers and are not advected elsewhere in the domain.

P19:L5-6: If all of the BC combinations are similar, why do you decide on BC code 111 in the end? Is this the most physical? The simplest computationally? Provide some support for this choice. Do you think that this should be the upper BC by default for ICAR?

Section 4.5 I would restructure/rewrite this section. If WRF and ICAR are using the same MP schemes, then all differences in the hydrometeor, water vapor, and precipitation fields should be due to the wind field and advection code. Indeed, this seems to be the main conclusions of sections 4.5.1 and 4.5.2. So, I would just approach this section by way of differences in the ICAR and WRF wind fields, and then use those differences to explain the observed differences in ICAR and WRF hydrometeor, water vapor, and precipitation fields. This way it is organized cause→effect instead of effect→cause.

P 28:L5: This point should be reflected in your conclusions.

P29:L14-15: I do not agree with this statement. Horak et al. 2019 did not test model top heights up to the Zmin of 15.2 km. It has not been shown that comparing simulation output with measurements leads to an incorrect result – you would have to show me the comparison with measurements for ICAR-O with a model top of 15.2 km for me to believe that statement. Your background information on the MSE of ICAR-O with the mentioned measurements given on P35:L21-23 clarifies your statement though – perhaps you could move some of this to the earlier reference.

P29:L15: This difference in model top heights between the two simulations (ICAR-N and ICAR-O) seems unfair. ICAR-N represents an altered model. However, the model top height is chosen to minimize errors relative to an ideal model setup. This choice of the model top height is a user parameter given to ICAR, not a feature of ICAR-N itself. I can see why you do this though, to show the "best-case" setup following your procedure. Still, I would like to see an "ICAR-O/N" simulation in this section with ICAR-O run with a model top of 15.2 km. If you also wanted to compare these simulations to the measurement dataset used in Horak et al. 2019, it would make this section much stronger.

P 31:L3: To support this statement of the upper BC in ICAR-O causing the excess hydrometeor concentration, I would like to see the vertical wind field for the model top of ICAR-O, or some evidence of strong downward fluxes.

P34:L3: You suggest that the model top height has an effect on the model mainly by controlling if the model top cuts through up and down drafts. To me, this suggests that the presence of negative fluxes, as you discuss in Section 2.2.2, and the elimination of these fluxes, should be dictating where the model top is. Following this logic, different upper boundary conditions should then also have an effect on the model top height. However, in figure 5, you demonstrate that this is only the case for potential temperature using a CG vs a ZG. Can you explain this inconsistency? Isn't it strange that there was 0 effect on Zmin by changing the upper BCs? Especially given that you conclude in section 4.6 that the ICAR-O simulations are affected by the upper BC used.

Technical comments:

P5:21 – sentence needs to be fixed for clarity

P6:L30-31: This description of ICAR-N, "calculates N from the perturbed state of the atmosphere predicted by the ICAR-O" should be somewhere in section 2.2.1. It makes clear the differences between ICAR-N and ICAR-O, I could have used it earlier.

P9: L22: "to as [a] set of boundary conditions"

P12:L29: "This section", please give section number of case study results.

Figure 6, Figure 13 caption should read: "Reduction of error (RE)"

Figure 13: I feel that this should preceded figure 12. Figure 13 supports your model configuration as discussed in the third paragraph of section 4.6, while figure 12 provides results relevant to paragraph 4. I see that they are ordered this way since you refer to the South Island DEM first, but the order ends up illogical when the whole figures are taken into consideration.

P36:L1: should read " possible model top elevation Zmin to produce"

P36:L8-9: This is not a finding or a recommendation. It should be removed from the list.

---

## Referee Comment (RC2) · Anonymous Referee #2 · 25 Nov 2020

The paper deals with numerical modifications to the ICAR model made and tested by the authors. In general, it is a very extensive work with lots of statistical tests and well-selected metrics. As such, the paper certainly fits to the scope of GMD. It has, however, weaknesses that are shortly mentioned before more specific remarks are added. The general remarks are listed according to the three main modifications discussed in the paper:

(a) The usage of an undisturbed potential temperature to calculate the background thermal stability N is a very useful new option to ICAR. The respective sub-section 2.2.1 should be written in a clearer way. I would recommend to use equations specifying the calculation of N, e.g. $N^2 = g/\Theta_0 \, d\Theta/dz$ with $\Theta = \Theta_b + \Theta'$; here, $\Theta(x,y,z,t)$ is the full field consisting of a background $\Theta_b(z)$ and wave perturbation $\Theta'$ and $\Theta_0$ is the constant surface value defined on page 7 line 33. By specifying if only $\Theta_b$ or the full $\Theta$ is used to calculate the background N, the reader can easily comprehend what is done.

Writing this equation, another thought came to my mind. It seems to be that the numerical simulations were conducted with another equation for calculating $\Theta_b$ from a given N-value, namely, $N^2 = g \, d(\ln(\Theta_b))/dz$ which would explain the values of the isentropes in Figure 1. I hope, my guess is right as this is not specified in the paper. Here is my general concern: as the linear wave equations are derived under the Boussinesq approximation (assuming linear $\Theta_b$ profiles with constant N) how the exponential increase of $\Theta_b$ fits to the given assumptions?

In other words, does the Boussinesq approximation per se limits the vertical extent and depth of the numerical model simulations? Linear numerical ICAR simulations covering the whole troposphere and lower stratosphere should be made with another set of linear wave equations that is derived from the anelastic equations. So, parts of the observed deviations might be related to the violation of the applicability of the Boussinesq approximation. A discussion of this aspect is highly appreciated in the parts relating to the tests of the model top.

(b) There is an extensive testing of different boundary conditions applied at the models top. Generally, there are three types of boundary conditions for finite difference schemes: Dirichlet boundary conditions (prescribes the value), Neumann boundary conditions (describes the derivative), and mixed boundary conditions. It would be beneficial for the reader to obtain a structured and - this is the main point - physically-motivated description of the various boundary condition settings as listed in Tables 2 and 3 based on the knowledge of finite-difference schemes.

Furthermore, each boundary condition leads to a different numerical problem to be solved and differences in the presented solutions are not surprising and obvious. However, they were never discussed in the frame of physical arguments. Only, an "optimization" based on the extensive tests was presented that might be feasible but I learnt not much and I have doubts if it can lead to general conclusions.

The other main issue with the presentation of the boundary condition is the missing information about the boundary conditions for the velocity components. Obviously (again I have to guess as this information is not provided in the paper), the perturbations of the velocity components are calculated everywhere in the model domain and at the boundaries. This would explain, why finite values of w' appear at the uppermost model level. Here, I would recommend a very simple, well-approved solution that avoids all the extensive testing: set w'=0 at the model top and relax all PERTURBATION variables u', v', w', Theta', q', … to zero in a shallow sponge layer beneath the model top. In this way, you avoid any flux of material in and out of the model top and the wind, Theta, and moisture fields only consist of the background values.

(c) The following comments relate to the general aspects in the paper. I had a hard time reading the text. The text is full of many details that are hard to track, sometimes repetitions hinder the reading, and there is information missing that initiates thoughts if I as a reader have missed something in the previous text or if this information simply not given in the text. I really would appreciate a careful editorial revision of the whole text as the whole writing is in contrast to the high-quality figures prepared for this submission.

I just go through some sections to provide examples (only selected examples - I didn't scan the whole text):

page 1, line 2: "As a consequence, ..." Of what?? The sentence further says that a model may yield correct results for the wrong reasons. So, is this a consequence of the content written in the sentence before?

page 1, line 8: Is evaluation the same as the above-mentioned verification?

page 2: Parts of the Introduction are rather general that (in my view) only have a slight or limited relation to the content of the paper, especially, the parts belonging to the thoughts about verification. Either they should be deepened or omitted.

page 2, 1st para: It is also the imperfection of observing systems (especially, when you consider remote-sensing systems and combinations of them) that lead to a fragmentary and often unsatisfying verification.

page 2, line 10: A good reference is Stensrud, D. (2007). Parameterization Schemes: Keys to Understanding Numerical Weather Prediction Models. Cambridge: Cambridge University Press. doi:10.1017/CBO9780511812590. Furthermore, it is often the lack of knowledge about essential processes that limit predictability (e.g. gravity wave parametrizations).

page 2, line 12: I have problems with the saying "right, but for the wrong reason". Most often, only one selected diagnostics is picked. The Zhang paper makes it clear: if the authors would have looked at vertical winds instead they had realized that there is an essential mechanism not represented in the model, namely the convection. So, the story with the causal chain (see also page 1 line 3) can be misleading. It is too much linear thinking in it - at least for my taste.

page 2, line 28/29: I could imagine that a more educational verification would be the comparison with a linearized version of WRF or another NWP models. This would provide a real one-to-one comparison. I wonder why this option is not considered.

Section 2

I recommend to rewrite the whole section (especially, Section 2.1) totally and add all parts that appear later in the text regarding the model set up (essentially, 2nd and 3rd para from page 7, Section 3.3, and maybe more). Section 2 should provide the reader with all information to understand the numerical integration of ICAR. This is probably best done by presenting the applied linear wave solutions, the advection equation (for which variables?? It was not clear to me when I read the paper first), and by specifying the initial and boundary conditions for all quantities by means of equations. The authors might argue this is done in the Gutmann paper but the above example of the calculation of the Brunt-Väisälä frequency shows that clarification is necessary as much as is possible.

Regarding advection: Do you advect full Theta or Theta'? Do you advect specific variables? For example, is Psi in Equation (1) rho times, say Theta?

There are probably more issues but I stop here. Altogether, I'm not convinced by the stated advantage of using ICAR (less computer time – this was not documented) when one has to spend massive resources (time and man power) to optimize a model for applications (microphysical and moist processes) that are rarely linear. Also, the back-link to the verification theme in the Introduction could be added!

Andreas Dörnbrack

---

## Referee Comment (RC3) · Anonymous Referee #3 · 18 Dec 2020

**General comments**
This study performed a detailed process-based analysis of ICAR and its sensitivity to the calculation of the Brunt-Väisälä frequency, sounding, boundary conditions, mountain geometry, specifically focusing on cloud microphysics processes. Overall, I think the manuscript is organized well and most of the results are described clearly (except where noted). The method described to determine the minimum model top seems promising and the paper provides a useful example for researchers to test their own idealized simulations before moving into more costly 3D real cases with NWP models. I have provided numerous specific comments below, and overall, I think a revised manuscript would be a good contribution to GMD.

**Specific comments**
**[P]age#, [L]ine#**
- P1, L21 – "a large shift in" is vague. IS this a spatial/temporal shift? Is it a shift greater or less than the observations or simulations?
- P2, L5 – what is meant by "epistemological reasons"? It is vague and unclear to me what message is being conveyed. I read the abstract in Oreskes et al. and it helped to understand your "reasons", but I think it would be helpful for the reader to provide an example or paraphrase Oreskes and/or provide your own explanation.
- P2, L30 – "cannot be inferred from", why? What other information is needed? [Schlunzen 1997]
- P3, L14 – "distribution of precipitation", is this referring to spatial or temporal distribution?
- P3, L14 and throughout the text – "correct for the wrong/right reasons" is a catchy phrase, but I think there are places in the text where it would be good to state out right what you want to say. For example, you could replace the phrase with "results that compare well with observations, yet were produced by a different chain of processes than those found in the observations" or "model results that were produced by a chain of processes similar to those found in the observations".
- P5 – How frequent is the forcing timestep compared to the model time step?
- P7 – What is the model time step used for the ideal case configuration?
- P8, L2 – Do you mean that simulations had a constant RH with height at the extremes of no moisture (RH=0%) and a completely saturated vertical column (RH=100%)? You do test model tops from 4.4km to 14.4 km, and although saturated conditions are realistic for the lower troposphere, it's a bit unrealistic to have an orographic cloud (saturated conditions) be deeper than 10 km, especially going into heights of 14.4 km. Perhaps I'm missing something, but if RH=100% in the initial sounding, you would have a cloud moving over the mountain, as opposed to have cloud develop through orographic lift as the moisture encounters the barrier and reaches saturation. This needs to be clearer.
- P13, L31 – I don't see the "slight distortions" you speak of, that said, what is the physical importance of this distortions?
- P14, L7 – So the large deviations over a small spatial area are averaged out in the MAE calculation? If so, clearly state this point, don't allow for any misinterpretations. Tell the reader what you want them to understand.
- P14, L14-15 – What is meant by "an elevation dependence"? Explicitly describe these features and why they are relevant to note.

- P16, L8 – Potentially repetitive sentence starting with "Potential temperature…", this statement was essentially said on L3-4.
- P18, L10 – So the upper levels become more stable? How much do the upper levels "heat up"? Potential temperature increases on the order of 1K or 10K? Do you know why this is happening?
- P21 – In Figure 6 I noticed the spread in the RE in dependence of z_top has a large spread due to the scenario for q_sus, P_12h, Qv, and Qsus, while the other variables have a narrower spread, meaning that the dependence on scenario is much less. Could you discuss this result in the text and provide some insights on why the scenario sensitivity varies so much for some variables?
- P23, L12 – I wouldn't say "farther upwind", it's more like over the windward slope
- P25 – For Fig 9c, what do you think is happening very far downwind approaching the rightmost boundary, why does ICAR get drier with height?
- P26, L3 – Don't use "observations" here, I think it should probably be "both simulations"
- P26, L3-4 – This sentence was confusing to me. What is meant by "close to the surface"? upwind or downwind? Are you referring to the windward and leeward slope from the previous sentence?
- P26, L6 – Please reference Fig 11a after "upwind of the ridge"
- P27, L19-20 – So ICAR-N is making more cloud ice than cloud water, right? To trigger autoconversion you would need to reach a certain threshold of cloud water mixing ratio, then the scheme should convert water vapor to cloud water. This would make sense for WRF since the vertical velocities are faster over the windward slope relative to ICAR-N. Is there a significant change in the height of the freezing level upwind that could potentially impact the development of cloud ice in ICAR-N? What is the ice nucleation process in the scheme, i.e., what conditions must be met to convert water vapor to cloud ice? Do you think perhaps the Bergeron-Findeisen process (cloud ice grows at the expense of supercooled cloud water, leading to conversion from cloud ice to snow, and subsequent depositional growth of snow?) is more prominent in ICAR-N, thus leading to more cloud ice than cloud water in the suspended hydrometeors?
- P33, L18 – How computationally frugal is ICAR compared to WRF simulations in the case study explored here? Can you provide comparison of computational costs and wall-clock time?

**Technical corrections**
**[P]age#, [L]ine#**
- P4, L 7 – remove the word "eventually", removing it still keeps the same message and the word is unnecessary
- P5, L23 – change "is" to "are"
- P8, L3 – rewrite to "Since ICAR does not currently support…"
- P15 – the caption for Figure 1 seems incorrect… "Perturbations of the horizontal perturbation", should this be "Vertical cross-sections of the horizontal perturbation"?
- P20, L3 – it should be "Fig. 6a-g"
- P20, L23 – clarify that you mean spatial distribution of these quantities

- P21 – The blue contours in Figure 7a are difficult to distinguish for me, perhaps adding a different line type (dashed, dotted, etc) could help, or different colors that aren't so similar
- P24, L17 – should be Fig. 10d
- P24, L18 – should be Fig, 10b
- P26 – Can you have the isentropes be at the same interval and starting potential temperature as in Fig 9? This will facilitate comparison better.
- P30 – Caption for Fig 12, either the text is incorrect or panels c and d are mislabeled. In the text it says ICAR-O should be Fig 12c and ICAR-N is Fig 12d (P30, L1-2), but that's not how the panels are labeled. From the P24h results, it seems like panel d is ICAR-N, unless panel b is also mislabeled….please check these figures and make sure they're labeled correctly in the figure, in the caption, and in the manuscript text.
- P30, L5 – remove the "to" before "producing"
- P33, L18 – "an" should be "a" before "computationally"
- P35, L26 – delete "a" before "comparisons"
- P35, L28 – typo, should be "for"
- P35, L31 – typo, should be "following"
- P36, L1 – change to "produce"

---

## Author Comment (AC1) · 28 Jan 2021

**Response to Reviewer 1**

**Abbreviations and the use of colors in this response:**

> AR    Author Response (Johannes Horak)
>
> RC    Reviewer Comment
>
> Note that in this response orange bold text**, such as this,** or a red equation, indicates a change in the manuscript text (either an addition or some rephrasing). Blue or crossed out text  indicates the removal of text. Additionally, each modification in the manuscript is complemented by the respective page number and line where the modification occurred (e.g. **P01L01**). In the case where a table was altered, the respective modification is highlighted by an orange box around the modification. A modified figure is indicated by an orange border.

**RC:** In this paper, the authors present a detailed investigation of the upper boundary within the ICAR model, as well as a well-supported correction to the calculation of the BruntVäisälä frequency for linear theory calculations. They perform rigorous comparisons of their improved model (ICAR-N) with the old model (ICAR-O), the analytic solution to the underlying equations of ICAR, and to a more complex atmospheric model, WRF. The questions posed, methodologies employed to answer them, and final conclusions reached, are of value to the development community of ICAR. In the test case, these results support their conclusion that some model top height exists which reduces errors to the advected quantities and microphysics of the model which maximizing computational efficiency.

> **AR:**
>
> We thank the reviewer for the time and effort they took to evaluate the manuscript and provide constructive feedback. We thoroughly went through every comment and revised the manuscript accordingly where appropriate.
>
> Please find a detailed response to every comment below.

**RC:** However, after testing a number of boundary conditions (BC) for the upper boundary, they fail to provide a clear recommendation for which combination of upper BC is most favorable. Such a determination would surely strengthen their conclusions, and be of great value to the community using ICAR. The demonstrated lack of dependence between minimum model top height (Zmin) and combination of BCs seems to contradict the hypothesis that Zmin is chosen to avoid errors in the assumed downward fluxes. The authors should explain this discrepancy and, if possible, provide evidence in support of a combination of upper BCs to be used as default in ICAR going forward.

> **AR:** We agree that a recommendation for a set of BCs would be preferable, however, our study has shown that the main strategy to avoid errors in the $q_v$, $q_{sus}$ and $q_{prc}$ fields is to employ a high model top setting that covers at least the troposphere. The only quantity where a BC may yield significant improvements even at higher model top settings is potential temperature. However, our study has shown that a CG BC on Theta leads to potential problems with the numerical stability of simulations. While this did not affect the idealized studies it was an issue for the real case scenario. Therefore, we chose to avoid a general recommendation. We rephrased the relevant section in the discussion in order to better clarify that a higher model top is our recommendation.

Concerning the discrepancy – please refer to the answer to RC comment concerning P34:L3 (farther below).

**Adjustment to the manuscript**

> **P38L21:** It seems unlikely that any boundary condition is able to accurately represent the effect of cloud and precipitation processes above the model domain and the resulting interaction with the corresponding processes in the model domain (e.g. the seeder-feeder mechanism). Therefore, in order to capture all relevant cloud and precipitation processes**, it is recommended that** the vertical extension of the domain should at the very least encompass the entire troposphere."

**RC:** P7:L10-11: Why use a constant dz spacing for ICAR, but not for WRF? ICAR v1 supports this.

> **AR:** While ICAR v1 does indeed support variable vertical layer thicknesses, these are, nonetheless, horizontally constant. The WRF setting was, except for the amount of vertical levels, left at its default, resulting in variable dz spacings. For quantitative comparability WRF cross sections were linearly interpolated to the ICAR grid.

**RC:** P7:L23-24: How did you test Thompson MP code to see that ICAR and WRF produce the same results? This sounds like you have ICAR producing identical output to WRF. This would be exceptional, but unlikely.

> **AR:** We rephrased to better indicate the code review and WRF 3.4 simulations employed to rule out differences in the Thompson microphysics implementation.

> We went through the Thompson MP code and compared the definitions of the variables definable in the ICAR options and the values of the constants defined in the first 386 lines of code. The only difference found was for the value of C_sqrd where ICAR Thompson uses 0.3 and WRF Thompson 0.15. We then ran simulations with the C_sqrd value set to 0.15 but this only yielded negligible differences in the simulation results for the idealized default scenario. Additionally, we checked the ICAR Thompson MP code for differences made since it was forked from the WRF repository. Where we found differences we undid the changes and tested whether the idealized simulations were affected – we did not find any indication that the functionality of ICAR Thompson differed from WRF Thompson. As an additional check we simulated the idealized default scenario with the WRF version from which the ICAR Thompson code was forked from (WRF-3.4) and noticed only negligible differences to the results obtained with the WRF 4.1.1 version.

**Adjustment to the manuscript**

> **P09L23:** The Thompson microphysics scheme as described in Sect. 2 is employed in all models. The **ICAR implementation of the Thompson MP was forked from WRF version 3.4. Preliminary tests were conducted, showing that WRF 3.4 and WRF 4.1.1 yielded the same results for the default scenario, with only negligible differences. Additionally, the** code of the Thompson MP implementation in ICAR and WRF **4.1.1** was reviewed and tested to ensure that  **differences between the implementations did not affect the results**. All input files and model configurations are available for download (Horak, 2020).

**RC:** Section 2.2.2 – Please also state explicitly that upward fluxes result in quantities being lost (no longer tracked) by ICAR. This would motivate the use of a downward flux BC that seeks to balance this, and may explain why the current downward flux BC in ICAR does not produce drastically unrealistic simulations on first pass.

    **AR:** We revised Section 2.2.2. to explicitly state this potential issue.

    **Adjustment to the manuscript**

        **P8L01:** **While the effect described above is related to downdrafts at the model top, note that updrafts, on the other hand, may cause moisture to be transported out of the domain, leading to a mass loss. However, for $k=N_z$ and $w^t_{Nz+1/2} > 0$ and $w^t_{Nz+1/2} > 0$, the discretization of the vertical flux divergence in Eq. (9) yields**

$$\frac{\partial(w\psi)}{\partial z} \approx \frac{1}{\Delta z}\left(\psi^t_{N_z-1} w^t_{N_z-1/2} - \psi^t_{N_z} w^t_{N_z+1/2}\right)$$

        **Therefore, this issue cannot be addressed by applying different boundary conditions, since Eq. (17) does not depend on $\psi_{Nz+1}$.**

    **Adjustment to the manuscript**

        **P8L16:** Therefore, this study investigates whether the application of computationally cheaper alternative boundary conditions is able to reduce errors caused by, e.g., the unphysical mass influx **and loss** described above.
* * *
**RC:** Section 3.4 – I could use some discussion of the different BC's, what they try to represent, and why you chose them. Not too much, perhaps just a sentence or two for BC's1-4. Especially prior to where you refer to them on P9:L26.

    **AR:** For clarification we added the equations corresponding to the respective BCs to Table 3. Furthermore, as requested, the introductory paragraph in Section 3.4. now gives some brief additional information for each of the BCs.

    **Adjustment to the manuscript**

        **P11L24:** To this end several alternative BCs to the existing zero gradient boundary condition are added to the ICAR code, their abbreviations**, mathematical formulation** and their numerical implementation are summarized in Table 2. **All BCs constitute Neumann BCs except for the zero value Dirichlet BC.** Per default ICAR imposes a ZG BC at the model top to all quantities**, corresponding to the assumption that, e.g. the mixing ratio of hydrometeors $q_{hyd}$ above the domain is the same as in the topmost vertical level. A ZV BC imposed on, e.g., $q_{hyd}$ avoids any advection from outside of the domain into it. The CG, CF and CFG BCs assume that a either the gradient, flux or flux gradient of $\psi$, respectively, remains constant at the model top, representing different physical situations. The respective discretizations of the equations given in Table 2 then determine the value of $\psi_{Nz+1}$.**

Please note that in order to show the adjustment right below, we inserted a screenshot of the updated table and highlighted the changes to the caption and table itself with orange rectangles.

**Adjustment to the manuscript**

**P12:**

**Table 2.** Overview of all types of boundary conditions that were imposed at the model top of ICAR in the sensitivity study. The table lists the ID number, the abbreviation used in this study, the full name and equation of the BC evaluated at $z = z_{\text{top}}$, and the resulting equation for $\psi_{N_z+1}$ required to calculate the flux at the top boundary of the domain in equation (16). Note that the zero gradient BC is a special case of the constant gradient BC and that the constant $c$ is chosen as $\psi_{N_z} - \psi_{N_z-1}$ Due to the upwind advection scheme each BC is only applied if $w_{N_z} < 0$.

| ID | abbreviation | boundary condition | | $\psi_{N_z+1}$ |
|----|--------------|--------------------|--------------------------|------------------|
| 0 | ZG | zero gradient | $\frac{\partial \psi}{\partial z} = 0$ | $\psi_{N_z}$ |
| 1 | CG | constant gradient | $\frac{\partial \psi}{\partial z} = c$ | $\max(0, 2\psi_{N_z} - \psi_{N_z-1})$ |
| 2 | ZV | zero value | $\psi = 0$ | $0$ |
| 3 | CF | constant flux | $\frac{\partial(w\psi)}{\partial z} = 0$ | $\frac{w_{N_z-1}}{w_{N_z}}\psi_{N_z}$ |
| 4 | CFG | constant flux gradient | $\frac{\partial^2(w\psi)}{\partial z^2} = 0$ | $\frac{1}{w_{N_z}}(2\psi_{N_z-1}w_{N_z-1} - \psi_{N_z-2}w_{N_z-2})$ |

**RC:** Section 4.1 – Do you think ICAR should be considering wave amplification as a result of decreased density with height? This seems to be the cause of a major difference between ICAR-N, the analytical solution, and WRF. EQ7 seems to suggest that such a correction is not too difficult to implement, but this may just appear deceptively simple.

**AR:** While such a modification was not the focus of the presented manuscript, the authors agree that it could be beneficial for ICAR. We added this aspect to the corresponding paragraph in the discussion section.

**Adjustment to the manuscript**

**P39L21:** This could have drastic consequences for the results of studies relying on ICAR to provide precipitation fields for, i.e. applications in hydrology or glaciology. **Future work could implement and investigate whether the amplification of perturbations due to the vertical density gradient yields ICAR-N results closer to those of WRF.**

**RC:** P18: L6-7: "If ztop is set high enough these deviations therefore do not affect the cloud processes below" Would you expect these deviations to be advected down into the model where cloud processes occur? Please provide some discussion for why the errors in potential temperature remain in the upper most layers and are not advected elsewhere in the domain.

**AR:** We provided additional discussion to clarify this point.

**Adjustment to the manuscript**

**P21L05:** As shown in Fig. 4, for simulations with higher model tops these are mainly confined to the topmost kilometer of the model domain. If $z_{top}$ is set high enough these deviations therefore do not affect the cloud processes below. **A potential reason for this behavior is that air that is either too warm or cold, depending on the error introduced by the BC, is advected into the topmost vertical level. From there it is redistributed by vertical and horizontal advection until an equilibrium is reached, effectively confining the introduced errors to the topmost vertical levels of the domain.** While the results indicate that a CG BC effectively reduces errors in $\theta$, it is found to be problematic for atmospheres with stronger stratifications.

**RC:** P19:L5-6: If all of the BC combinations are similar, why do you decide on BC code 111 in the end? Is this the most physical? The simplest computationally? Provide some support for this choice. Do you think that this should be the upper BC by default for ICAR?

> **AR:** We rephrased the corresponding sentence to provide support to the choice, see below. However, it should not be the default boundary condition in ICAR due to the potential numerical instabilities caused by placing a constant gradient BC on Theta. Furthermore, a general statement about which BC is the most physical is difficult to make since it effectively depends on the specific scenario that is investigated.

> **Adjustment to the manuscript**

> > **P22L04:** To reduce the parameter space in the following analysis, and since the results for each BC combination are very similar, the idealized simulations from here on focus on CG BCs imposed at the model top (BC code 111). **This combination is chosen over the others for its computational simplicity, the larger REs observed for $\theta$ and $q_v$, as well as the potential to reduce $z_{min}(\theta,\text{BCs})$ in the idealized simulations.**

**RC:** Section 4.5 I would restructure/rewrite this section. If WRF and ICAR are using the same MP schemes, then all differences in the hydrometeor, water vapor, and precipitation fields should be due to the wind field and advection code. Indeed, this seems to be the main conclusions of sections 4.5.1 and 4.5.2. So, I would just approach this section by way of differences in the ICAR and WRF wind fields, and then use those differences to explain the observed differences in ICAR and WRF hydrometeor, water vapor, and precipitation fields. This way it is organized cause!effect instead of effect!cause.

> **AR:** The authors agree that this is a possible restructuring of Section 4.5. However, in this case the intent specifically was to emphasize and identify differences between the fields (effects) and trace them back to their causes. In the context of the manuscript the authors feel that this approach is more instructive since it, overall, focuses on the effects brought on by the proposed modifications and the limitations of linear theory.

**RC:** P 28:L5: This point should be reflected in your conclusions.

> **AR:** Please refer to our AR on page 1 as to why we chose not to recommend a specific boundary condition (AR starting with "*We agree that a recommendation for a set of BCs would be preferable, ...*"). Additionally, the authors think that their main finding with regards to the boundary conditions is reflected in the discussion and conclusions, see:

**P38L17:** *"Boundary conditions imposed on $q_v$ and the hydrometeors at the upper boundary are found not to influence the value of $Z_{min}$ for the investigated parameter space despite potentially mitigating errors in the potential temperature and water vapor fields. In particular, the cloud formation and precipitation processes within the domain are shown to almost exclusively depend on the model top elevation $z_{top}$ and not on the chosen set of boundary conditions, and only stabilize for $z_{top} \geq Z_{min}$. It seems unlikely that any boundary condition is able to accurately represent the effect of cloud and precipitation processes above the model domain and the resulting interaction with the corresponding processes in the model domain (e.g. the seeder-feeder mechanism). Therefore, in order to capture all relevant cloud and precipitation processes, it is recommended that the vertical extension of the domain should at the very least encompass the entire troposphere."*

and

**P41L09**: *"While most of the tested boundary conditions (in comparison to the default zero gradient boundary condition) are suitable to reduce the errors in the water vapor and potential temperature fields, no tested combination of these boundary conditions can achieve a lower value for $Z_{min}$."*.

**RC:** P29:L14-15: I do not agree with this statement. Horak et al. 2019 did not test model top heights up to the Zmin of 15.2 km. It has not been shown that comparing simulation output with measurements leads to an incorrect result – you would have to show me the comparison with measurements for ICAR-O with a model top of 15.2 km for me to believe that statement. Your background information on the MSE of ICAR-O with the mentioned measurements given on P35:L21-23 clarifies your statement though – perhaps you could move some of this to the earlier reference.

> **AR:** We agree that the statement may be considered misleading without the additional context. Since the discussion on P40:L23 addresses this issue and the statement is, overall, better located there, we removed the statement.
>
> **Adjustment to the manuscript**
>
> > **P33L15:** This indicates that  the cloud formation processes in the ICAR-O simulation with the low model top elevation are likely unphysical and strongly disturbed by the model top.

**RC:** P29:L15: This difference in model top heights between the two simulations (ICAR-N and ICAR-O) seems unfair. ICAR-N represents an altered model. However, the model top height is chosen to minimize errors relative to an ideal model setup. This choice of the model top height is a user parameter given to ICAR, not a feature of ICAR-N itself. I can see why you do this though, to show the "best-case" setup following your procedure. Still, I would like to see an "ICAR-O/N" simulation in this section with ICARO run with a model top of 15.2 km. If you also wanted to compare these simulations to the measurement dataset used in Horak et al. 2019, it would make this section much stronger.

> **AR:** We agree that strictly the comparison is not the same, and, as correctly stated, the intention was rather to compare the best-case setup to the setup chosen in Horak 2019 and to highlight the resulting differences. We included an adapted version of Figure 13 and Figure 14 that includes ICAR-O (BCs 000) with ztop = 15.2 km in the middle column and adjusted the paragraph discussing these results accordingly. Please refer to **Section 4.6** in the updated manuscript, additionally, please find the key results from that Section below.

**Adjustment to the manuscript**

**P36L17:**

**Note that for this case study the effect of raising the model top elevation is mainly the removal of artificial clouds in the topmost model levels (compare Fig. 14a and b) and a weakening of the updrafts upwind of the initial peak in the topography (not shown), yielding a lower concentration of $\overline{q}_{prc}$ (compare 14d and e). Calculating the Brunt-Väisälä frequency from the atmospheric background state instead of the perturbed state of the domain, on the other hand, results in stronger updrafts and increased amounts of $\overline{q}_{prc}$ and $P_{24h}$ (compare Fig. 14e and f, as well as Fig. 14h and i).**

**AR (continued):**

However, we consider a comparison to measurements, such as performed in Horak 2019, as outside of the scope of the manuscript since it would require additional multi-year simulations for the South Island of New Zealand with ICAR-O(ztop=15.2 km).

**RC:** P 31:L3: To support this statement of the upper BC in ICAR-O causing the excess hydrometeor concentration, I would like to see the vertical wind field for the model top of ICAR-O, or some evidence of strong downward fluxes.

**AR:** Please find below an additional panel for ICAR-O that shows the vertical wind velocity w in the topmost half level of ICAR-O. Furthermore, the additional panel c shows the regions where the mass influx is triggered due to the mechanism described in 2.2.2 (orange regions). This requires not only a downward flux (which does not necessarily have to be strong) but rather a strong negative vertical gradient in w and nonzero amounts of a quantity $\psi$. Overall the conditions

$\psi > 0$ and

$w_{Nz+1/2} < w_{Nz-1/2} < 0$

must be satisfied. We adjusted the respective paragraphs accordingly to clarify the two conditions.

[Figure]

**Adjustment to the manuscript**

> **P07L28:** In case of downdrafts, $\psi_{Nz} > 0$ and vertical convergence in the wind field across the topmost vertical mass level $w_{Nz+1/2} < w_{Nz-1/2}$, this results in a negative vertical flux-gradient and an associated increase in $\psi$ (see equation 16).

**Adjustment to the manuscript**

> **P19L11:** The choice of an alternative BC over the standard ZG BC has the largest potential for a reduction of error when **(i)** the grid cells of the uppermost vertical level coincide with  regions of vertical convergence where $w < 0$ and $dw/dz < 0$ and (ii)  the vertical flux gradients $\phi_z$ in these regions are negative (see Sect. 2.2.2). **Note that this particularly requires $\psi > 0$.**

**RC:** P34:L3: You suggest that the model top height has an effect on the model mainly by controlling if the model top cuts through up and down drafts. To me, this suggests that the presence of negative fluxes, as you discuss in Section 2.2.2, and the elimination of these fluxes, should be dictating where the model top is. Following this logic, different upper boundary conditions should then also have an effect on the model top height. However, in figure 5, you demonstrate that this is only the case for potential temperature using a CG vs a ZG. Can you explain this inconsistency? Isn't it strange that there was 0

effect on Zmin by changing the upper BCs? Especially given that you conclude in section 4.6 that the ICAR-O simulations are affected by the upper BC used.

**AR:**

Whether the model cuts through a "trigger-region" where $w_{Nz+1/2} < w_{N\_z-1/2} < 0$ is satisfied is one of the two necessary conditions. The other is that $\psi > 0$. In our scenarios and the real case, water vapor $q_v$, suspended hydrometeors $q_{sus}$ and precipitating hydrometeors $q_{prc}$ generally tend towards zero with increasing elevation while potential temperature increases with elevation.

Therefore, for $q_v$, $q_{sus}$ and $q_{prc}$ the location of the trigger-regions becomes less important once the model top is at a sufficiently high elevation. This in contrast to $\theta$ which, in trigger regions, is always affected by the mechanism described in Sect. 2.2.2. Consequently, alternating the BC for $\theta$ has an effect on $Z_{min}$ but less so for $q_v$, $q_{sus}$ and $q_{prc}$ (and, by extension, precipitation P).

Figure 5 essentially reflects this behavior. While the BCs may reduce some error in $q_v$, $q_{sus}$ and $q_{prc}$ and $P_{12h}$ at lower model tops (but not nearly enough as to closely approximate the fields in the reference simulation and lower $Z_{min}$) they do not or much less so once the model top is high enough since concentrations of $q_v$, $q_{sus}$ and $q_{prc}$ are very low. As described above for $\theta$ the situation is different and here $Z_{min}$ is affected by the choice of the BC.

Due to the very low model top used in ICAR-O the boundary conditions do play a bigger role and cause the influx of additional water into the domain. Just using a higher model top elevation for ICAR-O would mostly alleviate this issue (compare the updated Figure 14). However, in that case ICAR-O still calculates the Brunt-Väisälä frequency from the perturbed state of the domain.

We clarified the text in Sect. 2.2.2 to emphasize $\psi > 0$ and added a third condition in the analysis of why microphysics species and potential temperature respond differently to the choice of boundary condition.

**Adjustment to the manuscript**

> **P07L28:** In case of downdrafts, $\psi_{Nz} > 0$ and vertical convergence in the wind field across the topmost vertical mass level $w_{Nz+1/2} < w_{Nz-1/2}$, this results in a negative vertical flux-gradient and an associated increase in $\psi$ (see equation 16).

**Adjustment to the manuscript**

> **P19L11:** The choice of an alternative BC over the standard ZG BC has the largest potential for a reduction of error when **(i)** the grid cells of the uppermost vertical level coincide with  regions of vertical convergence where $w < 0$ and $dw/dz < 0$ and (ii)  the vertical flux gradients $\phi_z$ in these regions are negative (see Sect. 2.2.2). **Note that this particularly requires $\psi > 0$.** For potential temperature, in case of the specified sounding, **all** conditions are always satisfied in some region no matter at what elevation the model top is chosen,

**RC:** P5:21 – sentence needs to be fixed for clarity

> **AR:** We modified the sentence accordingly.

**Adjustment to the manuscript**

**P07L15:** In the following the mass levels are indexed from 1to $N_z$ and the half levels bounding the k-th mass level **are denoted as k-1/2 and k+1/2. Note that** the vertical wind components  **are calculated at half levels with Eq. (8) and that, in particular, no boundary condition is required to determine w at the model top.**

**RC:** P6:L30-31: This description of ICAR-N, "calculates N from the perturbed state of the atmosphere predicted by the ICAR-O" should be somewhere in section 2.2.1. It makes clear the differences between ICAR-N and ICAR-O, I could have used it earlier.

**AR:** We rephrased parts of Section 2.1 and 2.2.1 to better clarify and introduce the difference between ICAR-O and ICAR-N.

**Adjustment to the manuscript**

**P05L03: Note that ICAR employs quantities from the perturbed state of the domain to calculate N even though in linear mountain wave theory N is a property of the background state (e.g. Durran, 2015).**

**Adjustment to the manuscript**

**P6-P7:**

**2.2.1 Calculation of the Brunt-Väisäla frequency**

**From the initial state of θ and the microphysics species fields at $t_0$ (see Eq. 12), ICAR calculates the (moist or dry, Eq. 6 and 7) Brunt-Väisälä frequency N for all model times $t_m$ smaller than the first forcing time $t_{f1}$. During each model time step, the θ and microphysics species fields in the ICAR domain are modified by advection and microphysical processes. Therefore, for model times $t_m > t_0$, θ and all the microphysics species q represent the perturbed state of the respective fields, denoted as**

$$\theta \quad = \quad \Theta + \theta' \text{ and}$$

$$q \quad = \quad q_0 + q'.$$

**Note that in this notation, the perturbed water vapor field is denoted as $q_v$, the background state water vapor field as $q_{v0}$ and the perturbation field as $q_v'$. Consequently, during all intervals $t_{fn} \leq t\_m < t_{fn+1}$, where $t_{fi}$ are subsequent forcing time steps, N is based on the perturbed states of potential temperature and the microphysics species at $t_{fn}$. More specifically, all atmospheric variables ICAR uses for the calculation of N with Eqs. (6) and (7) are represented by the perturbed fields.**

**However, in linear mountain wave theory N is a property of the unperturbed background state (e.g. Durran, 2015), an assumption that is not satisfied by the calculation method employed by the standard version of ICAR. This study therefore employs a modified version of ICAR that, in accordance with linear mountain wave theory, calculates N from the state of the atmosphere given by the forcing data set if**

**the corresponding option is activated. In the following, the modification of ICAR basing the calculation of N on the background state is referred to as ICAR-N, while the unmodified version, that bases the calculation on the perturbed state of the atmosphere, is referred to as the original version (ICAR-O). If properties applying to both versions are discussed, the term ICAR is chosen.**

**RC:** P9: L22: "to as [a] set of boundary conditions"

> **AR:** We inserted the missing "a".

**RC:** P12:L29: "This section", please give section number of case study results.

> **AR:** We rephrased to "**Section 4.6** additionally investigates..."

**RC:** Figure 6, Figure 13 caption should read: "Reduction of error (RE)"

> **AR:** We modified the caption as suggested.

**RC:** Figure 13: I feel that this should preceded figure 12. Figure 13 supports your model configuration as discussed in the third paragraph of section 4.6, while figure 12 provides results relevant to paragraph 4. I see that they are ordered this way since you refer to the South Island DEM first, but the order ends up illogical when the whole figures are taken into consideration.

> **AR:** We agree and have changed the order of the Figures.

**RC:** P36:L1: should read " possible model top elevation Zmin to produce"

> **AR:** We corrected the sentence as suggested.

**RC:** P36:L8-9: This is not a finding or a recommendation. It should be removed from the list

> **AR:** We rephrased the corresponding item to fit the list.

> **Adjustment to the manuscript**

>> **P41L07:**  **The** method described in this study to determine $Z_{min}$  **may be** applied to idealized simulations and a real case alike. **This was demonstrated as proof of concept.**

---

## Author Comment (AC2) · 28 Jan 2021

**Response to Reviewer 2**

**Abbreviations:**

AR Author Response (Johannes Horak)

RC Reviewer Comment

Note that in this response orange bold text**, such as this,** or a red equation, indicates a change in the manuscript text (either an addition or some rephrasing). Blue or crossed out text  indicates the removal of text. Additionally, each modification in the manuscript is complemented by the respective page number and line where the modification occurred (e.g. **P01L01**). In the case where a table was altered, the respective modification is highlighted by an orange box around the modification.

**RC:** The paper deals with numerical modifications to the ICAR model made and tested by the authors. In general, it is a very extensive work with lots of statistical tests and well-selected metrics. As such, the paper certainly fits to the scope of GMD. It has, however, weaknesses that are shortly mentioned before more specific remarks are added. The general remarks are listed according to the three main modifications discussed in the paper:

**AR:**

We thank the reviewer for taking the time to read the manuscript and provide constructive criticism. Please find our detailed answers below!

**RC:** (a) The usage of an undisturbed potential temperature to calculate the background thermal stability N is a very useful new option to ICAR. The respective sub-section 2.2.1 should be written in a clearer way. I would recommend to use equations specifying the calculation of N, e.g. $N^2 = g/\Theta_0 \, d\Theta/dz$ with $\Theta = \Theta_b + \Theta'$; here, $\Theta(x,y,z,t)$ is the full field consisting of a background $\Theta_b(z)$ and wave perturbation $\Theta'$ and $\Theta_0$ is the constant surface value defined on page 7 line 33. By specifying if only $\Theta_b$ or the full $\Theta$ is used to calculate the background N, the reader can easily comprehend what is done.

**AR:** We have carefully revised Section 2.1 and Section 2.2.1 to better introduce and clarify the changes made to ICAR and which quantities (perturbed or background state) are employed for the calculations. Due to the larger revisions we did not copy all changes in our Author response, please refer to the revised Sections in the updated manuscript. However, please find below smaller changes that directly address specific criticisms.

In particular Section 2.1 now shows the equations used to calculate the Brunt-Väisälä frequency and mentions early on that ICAR employs the quantities from the perturbed state of the domain (in contrast to the assumptions of linear mountain wave theory). This is now additionally indicated by the symbols used in the equations, e.g. the variables mentioned in context with the background state (given by the forcing data) are now distinct from those describing the perturbed state of the domain.

Similarly, Section 2.2.1 was revised to better indicate the difference between ICAR-N and ICAR-O and more clearly state which equations are employed and what state (base state or perturbed state) is the input for their evaluation.

**Adjustment to the manuscript**

**P04L06:** The forcing data set represents the background state of the atmosphere and must comprise the horizontal wind components **(U, V)** pressure **p**, potential temperature **Θ** and water vapor mixing ratio **$q_{v0}$**.

**Adjustment to the manuscript**

**P04L26:**

Note that depending on whether a grid cell is saturated or not, either the moist, $N_m$ **(Emanuel, 1994)**, or dry Brunt-Väisälä frequency $N_d$ is  **employed in Eq. (4) and calculated as**

$$N_d^2 = g\frac{dln\theta}{dz} \quad \text{and}$$

$$N_m^2 = \frac{1}{1+q_w}\{\Gamma_m\frac{d}{dz}[(c_p+c_l q_w)ln\theta_e]-[c_{\dot{\iota}}lnT+g]\frac{dq_w}{dz}\} \quad ,$$

**with the acceleration due to gravity g, the temperature T, the potential temperature θ, the equivalent potential temperature $\theta_e$, the saturated adiabatic lapse rate $\Gamma_m$, the saturation mixing ratio $q_s$, the cloud water mixing ratio $q_c$ and the total water content $q_w=q_s+q_c$, and the specific heats at constant pressure of dry air and liquid water $c_p$ and $c_l$. Note that ICAR employs quantities from the perturbed state of the domain to calculate N even though in linear mountain wave theory N is a property of the background state (Durran, 2015). Statically unstable atmospheric conditions (i.e., $N^2 < 0$) in the forcing data are avoided by enforcing a minimum Brunt-Väisälä frequency of $N_{min} = 3.2 \ 10^{-4} \ s^{-1}$ throughout the domain.**

**Adjustment to the manuscript**

**P06-P07**

**2.2.1 Calculation of the Brunt-Väisäla frequency**

**From the initial state of θ and the microphysics species fields at $t_0$ (see Eq. 12), ICAR calculates the (moist or dry, Eq. 6 and 7) Brunt-Väisälä frequency N for all model times $t_m$ smaller than the first forcing time $t_{f1}$. During each model time step, the θ and microphysics species fields in the ICAR domain are modified by advection and microphysical processes. Therefore, for model times $t_m > t_0$, θ and all the microphysics species q represent the perturbed state of the respective fields, denoted as**

$\theta = \Theta + \theta'$ and

$q = q_0 + q'$.

Note that in this notation, the perturbed water vapor field is denoted as $q_v$, the background state water vapor field as $q_{v0}$ and the perturbation field as $q_v'$. Consequently, during all intervals $t_{fn} \leq t\_m < t_{fn+1}$, where $t_{fi}$ are subsequent forcing time steps, N is based on the perturbed states of potential temperature and the microphysics species at $t_{fn}$. More specifically, all atmospheric variables ICAR uses for the calculation of N with Eqs. (6) and (7) are represented by the perturbed fields.

However, in linear mountain wave theory N is a property of the unperturbed background state (e.g. Durran, 2015), an assumption that is not satisfied by the calculation method employed by the standard version of ICAR. This study therefore employs a modified version of ICAR that, in accordance with linear mountain wave theory, calculates N from the state of the atmosphere given by the forcing data set if the corresponding option is activated. In the following, the modification of ICAR basing the calculation of N on the background state is referred to as ICAR-N, while the unmodified version, that bases the calculation on the perturbed state of the atmosphere, is referred to as the original version (ICAR-O). If properties applying to both versions are discussed, the term ICAR is chosen.

**RC:** Here is my general concern: as the linear wave equations are derived under the Boussinesq approximation (assuming linear Theta_b profiles with constant N) how the exponential increase of Theta_b fits to the given assumptions?

In other words, does the Boussinesq approximation per se limits the vertical extent and depth of the numerical model simulations? Linear numerical ICAR simulations covering the whole troposphere and lower stratosphere should be made with another set of linear wave equations that is derived from the anelastic equations. So, parts of the observed deviations might be related to the violation of the applicability of the Boussinesq approximation. A discussion of this aspect is highly appreciated in the parts relating to the tests of the model top.

**AR:**

The authors believe that the previous version of the manuscript may have potentially caused a misunderstanding, in particular with regards to the governing equations underlying ICAR, the equation employed to calculate the $\Theta(z)$ profile of the background state employed for the idealized simulations, the symbol employed for the surface value of the potential temperature $\Theta_0$ and the reference to Durran (2015). In the following paragraphs we attempt to clarify this and indicate separately which changes we made to the manuscript.

The reference to Durran 2015 was to underline that in linear mountain wave theory, N is a property of the background state and not the perturbed state of the atmosphere. However, Durran employs a definition of N yielding a linear $\Theta(z)$ profile, which is not employed by us. In the

following we compare the equation for N used by Durran (2015) and that employed by us where we have explicitly stated when a variable depends on z:

Durran (2015) equation for N:

$$N^2 = \frac{g}{\Theta_0} \frac{d\bar{\Theta}(z)}{dz}$$

the equation for N employed by us:

$$N_d^2 = g \frac{dln\,\Theta(z)}{dz} = \frac{g}{\Theta(z)} \frac{d\,\Theta(z)}{dz}$$

In the denominator of the first fraction, Durran uses $\Theta_0$ to denote a constant reference potential temperature, which does not occur in the equation employed by us, in our case it is $\Theta(z)$ – the vertical potential temperature profile of the background state which is not necessarily equivalent to Durrans $\bar{\Theta}(z)$.

We additionally want to note that our definition of the surface value of potential temperature (previously Section 3.2, P7L32: "...is characterized by a potential temperature at the surface, $\Theta_0 = 270K$, …") was denoted with the symbol $\Theta_0$ which may have contributed to the misunderstanding. Note that Durrans $\Theta_0$ is not the same as the potential temperature at the surface, which is what our $\Theta_0$ denotes.

Furthermore in the previous version of the manuscript we did not state on which equation we based our calculation of $\Theta(z)$, which left more room for interpretation. These $\Theta(z)$ profiles are calculated by solving for $\theta$ in equation (6) (see the revised manuscript), resulting in $\Theta(z) = \Theta_0 \exp(N^2 z/g)$ where $\Theta_0$ is the surface value of potential temperature. The manuscript now states the equation on which the calculation of our $\Theta(z)$ profiles is based. We additionally removed the symbols for the surface values of potential temperature and pressure since they do not occur anywhere else in the manuscript and may have contributed to the misunderstanding.

**Adjustment to the manuscript**

> We revised all of **Section 2.1** and **Section 2.2.1** to better clarify the underlying equations and which input quantities were used for which equation. Since these are larger changes please refer to the updated manuscript to view the revised Sections.

**AR (continued):** Section 2.1 now explicitly states the equation used by ICAR to calculate the dry Brunt-Väisälä frequency:

**Adjustment to the manuscript**

> **P04L29:**
>
> $$N_d^2 = g \frac{dln\,\theta}{dz} \quad (6)$$

**AR (continued):** Section 3.2 (Topographies and initial soundings) now refers to equation (6) as the basis for the calculation of the $\Theta(z)$ profiles. Note that we also removed the symbols for the surface values of potential temperature and pressure as to avoid misunderstandings resulting from these.

**Adjustment to the manuscript**

**P10L01:**

The vertical potential temperature profile of the base state $\Theta(z)$ is characterized by a potential temperature at the surface,  **of** 270 K, a constant Brunt-Väisälä frequency, $N = 0.01$ s$^{-1}$ **and calculated by solving Eq. (6) for $\theta$**. The horizontal wind components of the base state are chosen as $U = 20$ ms$^{-1}$ and $V = 0$ ms$^{-1}$, and the surface pressure as  1013 hPa.

**AR (continued)**

Note that the linearized and Boussinesq approximated governing equations of ICAR (see Barstad, 2006 or equations 1-8 in the revised manuscript), do not explicitly depend on $\Theta(z)$. They only require N to be constant, which is satisfied by our $\Theta(z)$ profiles. Additionally, note that ICAR is able to handle $\Theta(z)$ profiles yielding variable values of N by splitting the domain into smaller subdomains (please refer to Gutmann (2016) for a description of how this is accomplished by ICAR).

In addition to Barstad (2006), Boussinesq approximated governing equations that do not require linear $\Theta(z)$ profiles have been documented in the literature. For instance, Markowski (2011, p. 166, equations 6.18-6.21) and Nappo (2013, p.31, equations 2.1 – 2.4). Markowski (2011) explicitly states $N^2 = g/\overline{\Theta}\, d\overline{\Theta}/dz$ (page 166, right column at the bottom) and in his notation $\overline{\Theta} = \Theta(z)$ (page 166, left column first line).

Nappo (2013) uses $N^2 = -g/\rho_0\, \partial\rho_0/\partial dz$ and states explicitly, that $\rho_0$ is a function of z on page 32 in the first line below equation 2.17. Note that $N^2$ in Nappo (2013) and Markowski (2011) are essentially equivalent due to equation 1.65 in Nappo (2013): $g/\Theta_0\, \partial\Theta_0/\partial z = -g/\rho_0\, \partial\rho_0/\partial z$ .

The authors expect that differences due to a version of ICAR based on a set of linear wave equations derived from the anelastic equations would manifest mainly above the troposphere (e.g. Doyle, 2021, particularly Fig. 3), thereby not affecting orographic precipitation in most cases. A modification to ICAR with larger effects would, in our opinion, be the inclusion of the amplification of the perturbations due to the decreasing density (see equation 21 on P11L18 in the updated manuscript and P11L19-21). Nonetheless, the authors agree that implementing and evaluating either of these modifications to ICAR are interesting and necessary avenues for future research. However, they consider it outside the scope of this manuscript. We added these aspects to the relevant paragraphs in the discussion.

**Adjustment to the manuscript**

**P39L22:**

**Future work could implement and investigate whether the amplification of perturbations (see Eq. 21) due to the vertical density gradient yields ICAR-N results closer to that of WRF. Another conceivable avenue for future investigations in that regard could be the implementation and evaluation of a set of linear wave equations derived from the anelastic equations into ICAR-N.**

Durran, D.: MOUNTAIN METEOROLOGY | Lee Waves and Mountain Waves, in: Encyclopedia of Atmospheric Sciences (Second Edition), edited by North, G. R., Pyle, J., and Zhang, F., pp. 95 – 102, Academic Press, Oxford, second edition edn., https://doi.org/10.1016/B978-0-12-382225-3.00202-4, 2015.

Barstad, I. and Grønås, S.: Dynamical structures for southwesterly airflow over southern Norway: the role of dissipation, Tellus A, 58, 2–18, https://doi.org/10.1111/j.1600-0870.2006.00152.x, 2006.

Gutmann, Ethan, et al. "The intermediate complexity atmospheric research model (ICAR)." *Journal of Hydrometeorology* 17.3 (2016): 957-973.

Doyle, J. D., Gaberšek, S., Jiang, Q., Bernardet, L., Brown, J. M., Dörnbrack, A., Filaus, E., Grubišić, V., Kirshbaum, D. J., Knoth, O., Koch, S., Schmidli, J., Stiperski, I., Vosper, S., & Zhong, S. (2011). An Intercomparison of T-REX Mountain-Wave Simulations and Implications for Mesoscale Predictability, *Monthly Weather Review*, *139*(9), 2811-2831. Retrieved Jan 23, 2021

Markowski, P., & Richardson, Y. (2011). *Mesoscale meteorology in midlatitudes* (Vol. 2). John Wiley & Sons.

Nappo, Carmen J. *An introduction to atmospheric gravity waves*. Academic press, 2013.

**RC:** (b) There is an extensive testing of different boundary conditions applied at the models top. Generally, there are three types of boundary conditions for finite difference schemes: Dirichlet boundary conditions (prescribes the value), Neumann boundary conditions (describes the derivative), and mixed boundary conditions. It would be beneficial for the reader to obtain a structured and – this is the main point - physically-motivated description of the various boundary condition settings as listed in Tables 2 and 3 based on the knowledge of finite-difference schemes.

**AR:** The first paragraph in Section 3.4. now gives additional information and context for the tested boundary conditions. Additionally, we added the equations corresponding to the respective BCs to Table 3.

**Adjustment to the manuscript**

**P11L24:** To this end several alternative BCs to the existing zero gradient boundary condition are added to the ICAR code, their abbreviations**, mathematical formulation** and their numerical implementation are summarized in Table 2. **All BCs constitute Neumann BCs except for the zero value Dirichlet BC.** Per default ICAR imposes a ZG BC at the model top to all quantities**, corresponding to the assumption that, e.g. the mixing ratio of hydrometeors $q_{hyd}$ above the domain is the same as in the topmost vertical level. A ZV BC imposed on, e.g., $q_{hyd}$ avoids any advection from outside of the domain into it. The CG, CF and CFG BCs assume that a either the gradient, flux or flux gradient of $\psi$, respectively, remains constant at the model top, representing different physical situations. The respective discretizations of the equations given in Table 2 then determine the value of $\psi_{Nz+1}$.**

Please note that in order to show the adjustment right below, we inserted a screenshot of the updated table and highlighted the changes to the caption and table itself with orange rectangles.

**Adjustment to the manuscript**

**P12:**

**Table 2.** Overview of all types of boundary conditions that were imposed at the model top of ICAR in the sensitivity study. The table lists the ID number, the abbreviation used in this study, the full name and equation of the BC evaluated at $z = z_{\text{top}}$, and the resulting equation for $\psi_{N_z+1}$ required to calculate the flux at the top boundary of the domain in equation (16). Note that the zero gradient BC is a special case of the constant gradient BC and that the constant $c$ is chosen as $\psi_{N_z} - \psi_{N_z-1}$. Due to the upwind advection scheme each BC is only applied if $w_{N_z} < 0$.

| ID | abbreviation | boundary condition | | $\psi_{N_z+1}$ |
|----|--------------|--------------------|---|----------------|
| 0 | ZG | zero gradient | $\frac{\partial \psi}{\partial z} = 0$ | $\psi_{N_z}$ |
| 1 | CG | constant gradient | $\frac{\partial \psi}{\partial z} = c$ | $\max(0, 2\psi_{N_z} - \psi_{N_z-1})$ |
| 2 | ZV | zero value | $\psi = 0$ | $0$ |
| 3 | CF | constant flux | $\frac{\partial(w\psi)}{\partial z} = 0$ | $\frac{w_{N_z-1}}{w_{N_z}}\psi_{N_z}$ |
| 4 | CFG | constant flux gradient | $\frac{\partial^2(w\psi)}{\partial z^2} = 0$ | $\frac{1}{w_{N_z}}(2\psi_{N_z-1}w_{N_z-1} - \psi_{N_z-2}w_{N_z-2})$ |

**RC:** Furthermore, each boundary condition leads to a different numerical problem to be solved and differences in the presented solutions are not surprising and obvious. However, they were never discussed in the frame of physical arguments. Only, an "optimization" based on the extensive tests was presented that might be feasible but I learnt not much and I have doubts if it can lead to general conclusions.

**AR:** The authors agree that differences in the solutions were to be expected. We think that we did not state the underlying motivation for our tests clear enough. As pointed out by the reviewer and our manuscript, the problem of optimizing boundary conditions could potentially be circumvented mostly by e.g. employing a relaxation layer or just setting the model top high enough. Both methods have the "disadvantage" of requiring high model top elevations. However, particularly for ICAR a low model top elevation would be desirable due to the associated lower computational cost. At the beginning of our investigation it seemed feasible that an optimal boundary condition could potentially be found that would, overall, lead to satisfactory results for low model top elevations. The tested BCs are numerically easy to implement and correspond to comparatively simple physical situations, such as keeping the flux or the flux gradient at the model top constant, all of which could assumed to be physically reasonable assumptions at the model top under certain (but not all) circumstances. While we were able to reduce errors (depending on the boundary condition combination), for low model top elevation simulations these reductions weren't enough to more closely reproduce the fields in the respective reference simulations. Therefore we consider this less of an optimization: There simply is no optimal combination of the tested BCs that achieves the goal of lowering the minimum possible model top elevation $Z_{\text{min}}$ (apart from the discussed exception occurring for θ). Therefore, from our work it follows that future research should either explore more sophisticated options for BCs or entirely different strategies, such as the relaxation layer suggested by the reviewer.

We updated Section 2.2.2 and the discussion accordingly to better clarify our motivation and consequences for future studies.

**Adjustment to the manuscript**

> **P08L08:** **A solution to address both issues would potentially be to include a relaxation layer directly beneath the model top (see, e.g. Skamarock et al., 2019). Within this relaxation layer vertical wind speeds would tend towards zero with decreasing distance to the model top and perturbed quantities would be relaxed towards their value in the background state. Another potential solution is employed by** full physics models such as the Integrated Forecasting System (IFS) of the European Center for Medium-Range Weather Forecasts (ECMWF, 2018), the COSMO model (Doms and Baldauf, 2018) or the Weather Research and Forecasting (WRF) model (Skamarock et al., 2019). **These models** place the location of the upper boundary at elevations high enough where moisture fluxes across the boundary are negligible. While applying **either** treatment to ICAR is, in general, an option, it is undesirable since **both necessarily result in higher model tops and therefore** would severely increase the computational cost of ICAR simulations. **Hence**, this study investigates whether the application of **computationally cheap** alternative boundary conditions is able to reduce errors caused by, e.g., the unphysical mass influx **and loss** described above.

**Adjustment to the manuscript**

> **P39L31:** **Another possible venue for future research that aims to mitigate the influence of the upper boundary could be the implementation of a relaxation layer directly underneath the model top. In this layer perturbed quantities could, as they approach the model top, gradually be relaxed towards their background state values, while w is relaxed towards zero.**

**RC:** The other main issue with the presentation of the boundary condition is the missing information about the boundary conditions for the velocity components. Obviously (again I have to guess as this information is not provided in the paper), the perturbations of the velocity components are calculated everywhere in the model domain and at the boundaries. This would explain, why finite values of w' appear at the uppermost model level. Here, I would recommend a very simple, well approved solution that avoids all the extensive testing: set w'=0 at the model top and relax all PERTURBATION variables u', v', w', Theta', q', … to zero in a shallow sponge layer beneath the model top. In this way, you avoid any flux of material in and out of the model top and the wind, Theta, and moisture fields only consist of the background values.

**AR:** We have emphasized the information that w' is calculated at half levels (e.g. also at the top of the topmost vertical level) at the relevant locations in the manuscript.

The authors agree that setting w' = 0 at the model top and adding a shallow relaxation sponge layer beneath the model top is potentially viable solution that warrants further exploration. We added the sponge layer as a potential solution to a relevant paragraph in Section 2.2.2 and added it to the relevant part in the discussion **(see our previous AR and the associated adjustments to the manuscript directly above)**.

However, considering the state of the art with regards to ICAR before our investigation, boundary conditions seemed the more likely approach to allow ICAR simulations to run without compromising physical fidelity AND retain at least some computational advantage of the low model tops. Our results now show that BCs, while correcting some errors in the potential temperature and microphysics fields, mostly play a negligible role in the required model top elevation. In light of these results a sponge layer could – depending on its depth – potentially allow the choice of lower model tops than obtainable with the BC approach. Nonetheless it would still require substantial testing and development efforts to include and evaluate the functionality in ICAR. We considers this a very interesting and promising approach but outside of the scope of the presented manuscript.

**Adjustment to the manuscript**

> **P04L10:** **In particular mass based quantities such as water vapor are stored at the grid cell center while the horizontal wind components u and v are stored at the centers of the west/east or south/north faces of the grid cells and the vertical wind component w at the center of the top/bottom faces of each grid cell.**

**Adjustment to the manuscript**

> **P07L16:** **Note that the vertical wind components are calculated at half levels with Eq. (8) and that, in particular, no boundary condition is required to determine w at the model top.**

**RC:** (c) The following comments relate to the general aspects in the paper. I had a hard time reading the text. The text is full of many details that are hard to track, sometimes repetitions hinder the reading, and there is information missing that initiates thoughts if I as a reader have missed something in the previous text or if this information simply not given in the text. I really would appreciate a careful editorial revision of the whole text as the whole writing is in contrast to the high-quality figures prepared for this submission.

> **AR:** **We have carefully revised the manuscript, in particular Section 2** with a view on the reviewer's criticism. Particularly the introduction of all relevant equations and a better differentiation between background state variables and perturbed state variables is aimed to address the issues raised.

**RC:** page 1, line 2: "As a consequence, ..." Of what?? The sentence further says that a model may yield correct results for the wrong reasons. So, is this a consequence of the content written in the sentence before?

> **AR: P01L01-02:** We rephrased to "The **evaluation** of models in general is a non-trivial task and can, due to epistemological and practical reasons, never be considered as complete. **Due to this incompleteness,** a model may yield correct results for the wrong reasons, i.e. by a different chain of processes than found in observations."

**RC:** page 1, line 8: Is evaluation the same as the above-mentioned verification?

> **AR:** Thank you for pointing this out, indeed it is not and we have made modifications in the manuscript to clarify the difference. The sentence now reads:

**Adjustment to the manuscript**

**P01L01:** The **evaluation** of models in general is a non-trivial task and can, due to epistemological and practical reasons, never be considered as complete.

**Adjustment to the manuscript**

**P02L22:** In acknowledgment of the fundamental limitation of verification, **models are evaluated rather than verified, and** ...

**RC:** page 2: Parts of the Introduction are rather general that (in my view) only have a slight or limited relation to the content of the paper, especially, the parts belonging to the thoughts about verification. Either they should be deepened or omitted.

**AR:** We deepened and rephrased the first two paragraphs of the introduction to better establish the connection of our specific research question to the general topic of model evaluation. The authors believe that the wider view (including the epistemological and practical problems of model verification that translate to model evaluation) is particularly useful to emphasize that the comparison of model output to isolated measurements alone is not enough to sufficiently evaluate the reliability of a model.

**Adjustment to the manuscript**

**P02L03:** All numerical models of natural systems are approximations to reality. They generate predictions that may further the understanding of natural processes and allow the model to be tested against measurements. However, the complete verification **or demonstration of the truth of such** a model is impossible for epistemological **and practical** reasons (Popper, 1935; Oreskes et al., 1994). **While the correct prediction of an observation increases trust in a model it does not verify the model, e.g. correct predictions for one situation do not imply that the model works in other situations or even that the model arrived at the prediction through what would be considered the correct chain of events according to scientific consensus. In contrast, a model prediction that disagrees with a measurement falsifies the model, thereby indicating, for instance, issues with the underlying assumptions. From a practical point of view, the incompleteness and scarcity of data, as well as the imperfections of observing systems place further limits on the verifiability of models. The same limitations apply to model evaluation as well, however, evaluation focuses on establishing the reliability of a model rather than its truth.**

**RC:** page 2, 1st para: It is also the imperfection of observing systems (especially, when you consider remote-sensing systems and combinations of them) that lead to a fragmentary and often unsatisfying verification.

**AR:** The authors agree that this should be addressed explicitly as well. When extending the paragraph we included the reviewers suggestion.

**Adjustment to the manuscript**

**P02L10:** From a practical point of view, the incompleteness and scarcity of data, as well as the imperfections of observing systems place further limits on the verifiability of models.

**RC:** page 2, line 10: A good reference is Stensrud, D. (2007). Parameterization Schemes: Keys to Understanding Numerical Weather Prediction Models. Cambridge: Cambridge University Press. doi:10.1017/CBO9780511812590. Furthermore, it is often the lack of knowledge about essential processes that limit predictability (e.g. gravity wave parametrizations).

**AR:** We agree and included this as an additional reference.

**Adjustment to the manuscript**

**P02L15:** Those models approximate and simplify the world and processes in it by discretizing the governing equations in time and space and by modeling subgrid-scale processes with adequate parametrizations (e.g. Stensrud, 2009).

**RC:** page 2, line 12: I have problems with the saying "right, but for the wrong reason". Most often, only one selected diagnostics is picked. The Zhang paper makes it clear: if the authors would have looked at vertical winds instead they had realized that there is an essential mechanism not represented in the model, namely the convection. So, the story with the causal chain (see also page 1 line 3) can be misleading. It is too much linear thinking in it - at least for my taste.

**AR:** The authors agree that this is a rather obvious example, however, it serves to illustrate the problem. Nonetheless, human error, such as picking the wrong diagnostic, is arguably also a limitation to model evaluation and the fundamental source of error in epistemology. It may be reduced by measures introduced to, among other things, deal with this source of error such as peer review, careful study design or guidelines for model setups (e.g. see reference Warner, 2011 in the manuscript), but never entirely mitigated, and result in the model yielding predictions matching observations despite arriving at them through a possibly overlooked different process. Of course once the mistakes are identified it is usually easier to see how it could have been avoided in the first place.

However, it is also possible, for instance, that observations revealing that some current model arrives at a certain result through a different casual chain than what actually takes place, have not been performed or cannot be performed with high enough accuracy. It is also possible that the required observational techniques simply have not been developed or even envisaged yet.

Overall we do acknowledge the criticism and modified the manuscript in cases where it contributed to better clarity.

**Adjustment to the manuscript**

**P03L21:** This study aims to improve the understanding of the ICAR model and develop recommendations that maximize the probability that the results of ICAR simulations, such as the spatial distribution of precipitation, are correct and caused by the physical processes modelled by ICAR and not by numerical artifacts or any influence of the model top (correct for the right reasons).

**Adjustment to the manuscript**

**P03L25:** For a given initial state, a correct representation of the fields of wind, temperature and moisture as well as of the microphysical processes are a necessity to obtain the correct distribution of precipitation .

**Adjustment to the manuscript**

**P15L18:** **Section 4.6** additionally investigates whether this seemingly optimal result, as suggested by the lowest mean squared errors, was achieved  **due to the low model top potentially influencing the microphysical processes within the domain and the calculation of N being based on the perturbed fields.**

**Adjustment to the manuscript**

**P36L27:** Hence, it seems that the underestimation in precipitation near the crest and to its lee of an ICAR simulation with reasonably high model top compared to WRF (Fig. 9) is partly compensated in an ICAR simulation with a too low model top (ICAR-O$_{4 \text{ km}}$ in Fig. 14) by spurious effects introduced by the upper boundary conditions.  **Note that this seeming improvement is not due to a more realistic representation of cloud formation processes.**

**RC:** page 2, line 28/29: I could imagine that a more educational verification would be the comparison with a linearized version of WRF or another NWP models. This would provide a real one-to-one comparison. I wonder why this option is not considered.

**AR:** We agree that this approach would be of educational value, however, our aim was not to provide a one-to-one comparison. The intent behind using the standard version of WRF was to also infer differences due to non-linearities, wave amplification and the density decrease with height. This is stated in the introduction on **P03L31** and the methods Section on **P08L25-L26**. Note that the capability of ICAR to approximate the exact linear solution is inferred from comparing it to the analytical solution.

**RC:** Section 2 I recommend to rewrite the whole section (especially, Section 2.1) totally and add all parts that appear later in the text regarding the model set up (essentially, 2nd and 3rd para from page 7, Section 3.3, and maybe more). Section 2 should provide the reader with all information to understand the numerical integration of ICAR. This is probably best done by presenting the applied linear wave solutions, the advection equation (for which variables?? It was not clear to me when I read the paper first), and by specifying the initial and boundary conditions for all quantities by means of equations. The authors might argue this is done in the Gutmann paper but the above example of the calculation of the Brunt-Väisälä frequency shows that clarification is necessary as much as is possible.

**AR:** The authors agree that as much clarification as possible is needed, even if some equations from the cited references are repeated. Therefore, we followed most of the recommendations by the reviewer and revised Section 2, with a particular focus on a rewrite of Section 2.1. We included all the relevant equations from Gutmann 2016 to provide better context for the presented modifications, specified equations for the initial and boundary conditions and indicated which quantities are advected with the advection equation. **Please refer to the revised Section 2 and partial revisions in Section 3.4 in the new manuscript.**

However, we retained the basic structure since it follows, in our opinion, a logical and intuitive pattern: Providing a description of ICAR as it is in Sect. 2.1, describing the motivation and the modifications to ICAR in Sect. 2.2. and then detailing the specific model setups in the method subsection 3.1. This additionally allows readers to specifically jump to a given section to reread details of, for instance, the setup instead of having to search through one large section for these details. While we agree that other ways to structure the manuscript are possible, for the presented study this approach appears suitable to the authors.

**RC:** Regarding advection: Do you advect full Theta or Theta'? Do you advect specific variables? Forexample, is Psi in Equation (1) rho times, say Theta?

**AR:** ICAR advects Theta' (or $\theta$ in the notation employed in our manuscript). In the former equation (1) (now equation 9), $\psi = \theta$ or $\psi = q_v$, and so on – density is not included. We clarified this in the rewrite of Section 2.1.

**Adjustment to the manuscript**

**P05L21:**

**The microphysics species, $n_i$, $n_r$ and $\theta$ are advected with the calculated wind field according to the advection equation (Gutmann et al., 2016):**

$$\frac{\partial \psi}{\partial t} = -\left(\frac{\partial(u\psi)}{\partial x} + \frac{\partial(v\psi)}{\partial y} + \frac{\partial(w\psi)}{\partial z}\right),$$

**where $\psi$ denotes any of the advected quantities.**

**RC:** There are probably more issues but I stop here. Altogether, I'm not convinced by the stated advantage of using ICAR (less computer time – this was not documented) when one has to spend massive resources (time and man power) to optimize a model for applications (microphysical and moist processes) that are rarely linear. Also, the back-link to the verification theme in the Introduction could be added!

**AR:** The authors once again thank the reviewer for his constructive criticism and for sharing his opinion. Please note that the computational advantages of ICAR are documented in Gutmann 2016.

Regarding the back-link to the verification theme, please note the final paragraph in the conclusions:

**P41L17:** *This study highlights the importance of a process-based in-depth evaluation not only with respect to ICAR but for models in general. Particularly for regional climate models (RCMs) and numerical weather prediction (NWP) models, the results of the case study demonstrate a potential pitfall when model parameters are inferred solely from comparisons to measurements, potentially leading to situations for which model results are more prone to be right but for the wrong reasons.*

---

## Author Comment (AC3) · 28 Jan 2021

**Response to Reviewer 3**

**Abbreviations:**

AR Author Response (Johannes Horak)

RC Reviewer Comment

Note that in this response orange bold text, such as this, or a red equation, indicates a change in the manuscript text (either an addition or some rephrasing). Blue or crossed out text (example) indicates the removal of text. Additionally, each modification in the manuscript is complemented by the respective page number and line where the modification occurred (e.g. **P01L01**). In the case where a table was altered, the respective modification is highlighted by an orange box around the modification. A modified figure is indicated by an orange border.

**RC:** This study performed a detailed process-based analysis of ICAR and its sensitivity to the calculation of the Brunt-Väisälä frequency, sounding, boundary conditions, mountain geometry, specifically focusing on cloud microphysics processes. Overall, I think the manuscript is organized well and most of the results are described clearly (except where noted). The method described to determine the minimum model top seems promising and the paper provides a useful example for researchers to test their own idealized simulations before moving into more costly 3D real cases with NWP models. I have provided numerous specific comments below, and overall, I think a revised manuscript would be a good contribution to GMD.

**AR:**

We want to thank the reviewer for their time invested in reading through our manuscript, providing constructive criticism and feedback, as well as putting forward additional questions. We went through all reviewer comments and addressed the issues that were raised.

**RC:** P1, L21 – "a large shift in" is vague. IS this a spatial/temporal shift? Is it a shift greater or less than the observations or simulations?

**AR:** We clarified to "spatial shift" and rephrased the sentence to clarify that the shift is in relation to a simulation performed with an unmodified version of ICAR.

**Adjustment to the manuscript**

**P01L22:** The case study indicates a large shift in that the precipitation maximum for calculated by the ICAR simulation employing the developed recommendations in contrast is spatially shifted upwind in comparison to an unmodified version of ICAR.

**RC:** P2, L5 – what is meant by "epistemological reasons"? It is vague and unclear to me what message is being conveyed. I read the abstract in Oreskes et al. and it helped to understand your "reasons", but I think it would be helpful for the reader to provide an example or paraphrase Oreskes and/or provide your own explanation.

**AR:** We extended the corresponding paragraph to clarify.

**Adjustment to the manuscript**

**P02L03:** All numerical models of natural systems are approximations to reality. They generate predictions that may further the understanding of natural processes and allow the model to be tested against measurements. However, the complete verification or **demonstration of the truth of such** a model is impossible for epistemological **and** practical reasons (Popper, 1935; Oreskes et al., 1994). While the correct prediction of an observation increases trust in a model it does not verify the model, e.g. correct predictions for one situation do not imply that the model works in other situations or even that the model arrived at the prediction through what would be considered the correct chain of events according to scientific consensus. In contrast, a model prediction that disagrees with a measurement falsifies the model, thereby indicating, for instance, issues with the underlying assumptions. From a practical point of view, the incompleteness and scarcity of data, as well as the imperfections of observing systems place further limits on the verifiability of models. The same limitations apply to model evaluation as well, however, evaluation focuses on establishing the reliability of a model rather than its truth.

RC: P2, L30 – "cannot be inferred from", why? What other information is needed?

**AR:** We rephrased the corresponding part of the paragraph since, without more context, this statement may be considered misleading.

**Adjustment to the manuscript**

**P03L03:** Therefore, a direct comparison to a full physics-based model is generally not sufficient for an evaluation of ICAR. Note that since ICAR is not intended to provide a full representation of atmospheric physics. Furthermore, whether the results obtained from ICAR simulations are correct for the right reasons cannot be inferred from comparisons to measurements alone (Schlünzen, 1997), for instance, precipitation measurements, alone. Similar spatial distributions of precipitation may result from a variety of different atmospheric states. Therefore, the modelled processes yielding the investigated results need to be considered as well.

RC: P3, L14 – "distribution of precipitation", is this referring to spatial or temporal distribution?

AR: We rephrased the sentence to clarify to "spatial distribution of precipitation".

**Adjustment to the manuscript**

**P03L22:** This study aims to improve the understanding of the ICAR model and develop recommendations that maximize the probability that the results of ICAR simulations, such as the **spatial** distribution of precipitation, ...

**RC:** P3, L14 and throughout the text – "correct for the wrong/right reasons" is a catchy phrase, but I think there are places in the text where it would be good to state out right what you want to say. For example, you could replace the phrase with "results that compare well with observations, yet were produced by a different chain of processes than those found in the observations" or "model results that were produced by a chain of processes similar to those found in the observations".

**AR:** We went through all eight instances of this phrase used in the manuscript and added additional clarification in one case and removed the phrase in three cases. Note that the phrase is

used once in the abstract, five times in the introduction (not counting the direct quote from Zhang et al., 2013), once Section 4.6 and once in the conclusions. A variation of it was used in Section 3.6 and has been rephrased.

**Adjustment to the manuscript**

**P03L21:**This study aims to improve the understanding of the ICAR model and develop recommendations that maximize the probability that the results of ICAR simulations, such as the **spatial** distribution of precipitation, are correct **and caused by the physical processes modelled by ICAR and not by numerical artifacts or any influence of the model top (correct** for the right reasons).

**Adjustment to the manuscript**

**P03L25:** For a given initial state, a correct representation of the fields of wind, temperature and moisture as well as of the microphysical processes are a necessity to obtain the correct distribution of precipitation for the right reasons.

**Adjustment to the manuscript**

**P15L18: Section 4.6** additionally investigates whether this seemingly optimal result, as suggested by the lowest mean squared errors, was achieved for the wrong reasons. due to the low model top potentially influencing the microphysical processes within the domain and the calculation of N being based on the perturbed fields.

**Adjustment to the manuscript**

**P36L27:** Hence, it seems that the underestimation in precipitation near the crest and to its lee of an ICAR simulation with reasonably high model top compared to WRF (Fig. 9) is partly compensated in an ICAR simulation with a too low model top (ICAR-O4 km in Fig. 14) by spurious effects introduced by the upper boundary conditions. It follows that the seeming improvement in the latter case is right but for the wrong reasons. Note that this seeming improvement is not due to a more realistic representation of cloud formation processes.

RC: P5 – How frequent is the forcing timestep compared to the model time step?

**AR:** For the presented idealized simulations, where a forcing time step was 6 hours and the model time step of ICAR approximately 40 s, the ratio is roughly 540 model timesteps per forcing time step. The ratio was similar for the real case scenario. We added the model timestep durations to the section "Simulation Setup".

**Adjustment to the manuscript**

**P09L14:** The model time step of ICAR is automatically calculated by ICAR to satisfy the Courant-Friedrichs-Lewy criterion (Courant et al., 1928; Gutmann et al., 2016) and is approximately 40 s while for WRF it is set to 2 s.

RC: P7 – What is the model time step used for the ideal case configuration?

AR: We added another sentence to indicate both, the ICAR and the WRF model time steps.

**Adjustment to the manuscript**

**P09L14:** The model time step of ICAR is automatically calculated by ICAR to satisfy the Courant-Friedrichs-Lewy criterion (Courant et al., 1928; Gutmann et al., 2016) and is approximately 40 s while for WRF it is set to 2 s.**

**RC:** P8, L2 – Do you mean that simulations had a constant RH with height at the extremes of no moisture (RH=0%) and a completely saturated vertical column (RH=100%)? You do test model tops from 4.4km to 14.4 km, and although saturated conditions are realistic for the lower troposphere, it's a bit unrealistic to have an orographic cloud (saturated conditions) be deeper than 10 km, especially going into heights of 14.4 km. Perhaps I'm missing something, but if RH=100% in the initial sounding, you would have a cloud moving over the mountain, as opposed to have cloud develop through orographic lift as the moisture encounters the barrier and reaches saturation. This needs to be clearer.

**AR:** The moisture is indeed set to RH=100% across the entire vertical column. The main reason behind this choice was to maximize hydrometeor concentrations and the precipitation within the domain. We rephrased the sentence to put more emphasize on the fact that RH=100% across all vertical levels. Due to the slight orographic lifting throughout the domain upwind of the topographical ridge, the saturation drops below 100% almost immediately after onset of the simulation and stabilizes during the 18 hour spinup such that the cloud formation then only occurs when the flow encounters the barrier.

**Adjustment to the manuscript**

**P10L04:** For the comparison of the ICAR and WRF wind fields to an analytical solution, dry conditions with RH = 0 % are employed while otherwise saturated conditions with RH = 100 % are prescribed **throughout the vertical column** at all heights.

**RC:** P13, L31 – I don't see the "slight distortions" you speak of, that said, what is the physical importance of this distortions?

**AR:** We rephrased the respective sentence to clarify that the description referred to the slight differences of the ICAR-N u' field in comparison to the analytical u' field and that this results in correspondingly higher wind speeds in the referred region.

**Adjustment to the manuscript**

**P16L21:** In comparison to the analytical fields (Fig. 1a) the u' field in ICAR-N exhibits slightly lower values of u', particularly visible in the region where u' < 0 m s-1 from approximately 8 km upward, resulting in higher horizontal wind speeds in this region (Fig. 1b).

**RC:** P14, L7 – So the large deviations over a small spatial area are averaged out in the MAE calculation? If so, clearly state this point, don't allow for any misinterpretations. Tell the reader what you want them to understand.

**AR:** We rephrased accordingly to clarify.

**Adjustment to the manuscript**

**P16L28:** The reason for the relatively small difference to the MAEs of ICAR-N is that the MAE is calculated calculation across the entire cross section while the largest deviations are localized in a comparatively small region averages out the large deviations in the small spatial area around the topographical ridge at the center.

**RC:** P14, L14-15 – What is meant by "an elevation dependence"? Explicitly describe these features and why they are relevant to note.

AR: We rephrased according to the reviewers suggestions.

**Adjustment to the manuscript**

**P17L04:** The amplitudes in the perturbation fields in WRF are larger and exhibit an elevation dependence. For u' the elevation dependence indicated by Eq. (21). For w', for instance, the amplitude increases by 0.7 ms-1 from 4 km to 10 km, resulting in an increased orographic lift compared to ICAR. The range of observed values for u' ...

**RC:** P16, L8 – Potentially repetitive sentence starting with "Potential temperature...", this statement was essentially said on L3-4.

AR: We removed the sentence as it was indeed a repetition of what was said above.

**Adjustment to the manuscript**

**P19 First Paragraph:** Potential temperature fields are improved the most when a CG BC is imposed on 0 (Fig. 2a).

**RC:** P18, L10 – So the upper levels become more stable? How much do the upper levels "heat up"? Potential temperature increases on the order of 1K or 10K? Do you know why this is happening?

**AR:** The heating of the upper levels was at least ~300K and could potentially be more. However, ICAR enforces a maximum potential temperature value and outputs warnings to inform the user of exceedingly high or low (thresholds are user definable per quantity) values of quantities and the associated problems with a particular simulation. We hypothesize that under highly stable conditions the constant gradient boundary condition drastically overestimates the potential temperature in the level above the model domain, thereby advecting hot air into the uppermost level(s). While this might not be problematic for just a couple of time steps over multiple model timesteps the Theta gradient between the topmost two levels drastically steepens, and facilitates the calculated influx of hot air, eventually causing a numerical instability.

**RC:** P21 – In Figure 6 I noticed the spread in the RE in dependence of  $z_{top}$  has a large spread due to the scenario for q\_sus, P\_12h, Qv, and Qsus, while the other variables have a narrower spread, meaning that the dependence on scenario is much less. Could you discuss this result in the text and provide some insights on why the scenario sensitivity varies so much for some variables?

**AR:** The differences mainly stem from scenarios that generate clouds with a large vertical extension. We added an additional paragraph to discuss this observation.

**Adjustment to the manuscript**

P24L03:

Note that the spread of RE in dependence of  $z_{top}$  (Fig. 6) for  $\overline{q}_{sus}$ ,  $\overline{Q}_v$ ,  $\overline{Q}_{sus}$  and  $P_{12h}$  is mainly caused by scenarios that generate clouds with large vertical extensions. To better approximate the microphysical processes in the scenarios, and the resulting distribution of precipitation, higher model tops are required, leading to the observed spread. This affects, in particular,  $\overline{Q}_v$ ,  $\overline{Q}_{sus}$  and  $\overline{Q}_{prc}$  since missing vertical levels may significantly impact the total masses. In addition, note that while total masses are always compared to the respective mass found in the reference simulations,  $\overline{q}_v$ ,  $\overline{q}_{sus}$  and  $\overline{q}_{prc}$  can only be compared within the vertical extent simulated by the simulation with the lower model top.

RC: P23, L12 – I wouldn't say "farther upwind", it's more like over the windward slope

AR: We rephrased the paragraph to better clarify and avoid misunderstandings.

**Adjustment to the manuscript**

**P27L11:** With respect to water vapor ICAR-N is drier upwind of the topographical ridge and wetter downwind in comparison to WRF (see Fig. 9a-c). The regions with this dry and wet bias extend up to an elevation of approximately 6 km in which, farther up to 200 km upwind of the ridge, WRF-ICAR-N exhibits slightly stronger updrafts than WRF. This stronger orographic lift in ICAR-N-(Fig. 10e and d). Similarly yields a higher conversion rate of water vapor to hydrometeors. On the other hand, above the ridge the downdrafts calculated by WRF are of a higher magnitude than those predicted by ICAR-N, see Fig. 10c and d. Therefore, upwind of the ridge WRF transports more moist air from close to the surface to higher elevations. Above the ridge, on the other hand, Here, WRF advects drier air from higher elevations to lower levels. Hence, the two large regions in ICAR-N exhibiting a dry and wet bias in qv respectively are likely caused by the differences in the wind field. However, a Additionally, a small region with a wet bias close to the mountain ridge slope on the windward side is presumably caused by microphysical conversion processes (Fig. 10c).

**RC:** P25 – For Fig 9c, what do you think is happening very far downwind approaching the rightmost boundary, why does ICAR get drier with height?

**AR:** We rephrased the sentence addressing this issue to clarify.

**Adjustment to the manuscript**

**P28L04:** This low level dry bias is likely increased by ICAR-N, overall, extracting more precipitation from the moist atmosphere than WRF (see 4.5.2).**

RC: P26, L3 – Don't use "observations" here, I think it should probably be "both simulations"

**AR:** We rephrased the corresponding sentence and, as a result, had to slightly modify the preceding part of the paragraph as well.

**Adjustment to the manuscript**

**P30L02:** Figure 11a illustrates that P12h on the windward slope is substantially higher in ICAR-N than in WRF<del>. Conversely</del> and, conversely, ICAR-N is drier along the leeward slope. Both observations correspond This corresponds well to ...

**RC:** P26, L3-4 – This sentence was confusing to me. What is meant by "close to the surface"? upwind or downwind? Are you referring to the windward and leeward slope from the previous sentence?

**AR:** We rephrased the corresponding sentence for clarity.

**Adjustment to the manuscript**

**P30L03:** Both observations correspond This corresponds well to the distribution and shape of the precipitating hydrometeors close to the surface above the windward and leeward slope (see Fig. 9g and h) and the differences of  $q_{pre}$  between ICAR-N and WRF (see Fig. 9i).

RC: P26, L6 – Please reference Fig 11a after "upwind of the ridge"

**AR:** We added the corresponding reference.

**Adjustment to the manuscript**

**P30L04:** The precipitation maximum predicted by ICAR-N is approximately 25 mm and lies 6 km upwind of the ridge peak in comparison to the 32 mm maximum in WRF, which lies 4 km upwind of the ridge (**Fig. 11a**).

**RC** P27, L19-20: So ICAR-N is making more cloud ice than cloud water, right? To trigger autoconversion you would need to reach a certain threshold of cloud water mixing ratio, then the scheme should convert water vapor to cloud water. This would make sense for WRF since the vertical velocities are faster over the windward slope relative to ICAR-N. Is there a significant change in the height of the freezing level upwind that could potentially impact the development of cloud ice in ICAR-N?

**AR:** Yes, ICAR produces more cloud ice than cloud water. We assume that the reviewer was referring to ice nucleation or growth of cloud ice due to water vapor deposition, since autoconversion, to our knowledge, usually refers to conversion of cloud water to rain. Indeed WRF simulations exhibit a stronger updraft in comparison to ICAR above the upwind ridge (Fig. 10d), triggering the conversion of water vapor to super cooled cloud water. However, no, significant difference in the freezing level between ICAR and WRF is found upwind of the ridge.

**RC** P27, L19-20 – (continued): What is the ice nucleation process in the scheme, i.e., what conditions must be met to convert water vapor to cloud ice? Do you think perhaps the Bergeron-Findeisen process (cloud ice grows at the expense of supercooled cloud water, leading to conversion from cloud ice to snow, and subsequent depositional growth of snow?) is more prominent in ICAR-N, thus leading to more cloud ice than cloud water in the suspended hydrometeors?

**AR:** The conditions for the onset of ice nucleation (Thompson, 2008) directly from water vapor are either that (i) RH exceeds saturation by 25% with respect to ice or (ii) saturated conditions with respect to water exist (RH >= 100%) and T <  $-12^{\circ}$ C. Upwind of the ridge ICAR exhibits RH values between 100% and 125% with respect to ice and RH < 100% with respect to water. This is similarly found in the WRF simulation. Overall these atmospheric condition make it unlikely that the microphysics scheme directly converts water vapor to cloud ice and the similar atmospheric conditions would rather lead to the conclusion that ICAR-N and WRF should yield the same results.

Therefore, a potential alternative explanation is that in ICAR cloud water droplets heterogeneously freeze to cloud ice according to the mechanism described by Bigg (1953) with larger ice particles (>= 200 micrometers) being directly converted to snow, and Snow growing by depositional growth according to Srivastava (1992). However, in Bigg (1953) the probability of a droplet of cloud water freezing is related to its diameter and the air temperature. Droplets with higher diameters are more likely to freeze. In the Thompson MP scheme, the droplet size distribution is determined by the cloud water mixing ratio (e.g. Jones, 2014): Larger cloud water mixing ratios correspond to the median droplet diameter of the distribution. Note that WRF exhibits higher cloud water mixing ratios than ICAR-N and should therefore be more likely to convert cloud water to ice or snow. Overall, the issue remains inconclusive.

We have excluded, to the best of our knowledge, that differences in the implementation of the microphysics code in ICAR-N and WRF are the cause for the differences in the simulations:

We went through the Thompson MP code and compared the definitions of the variables definable in the ICAR options and the values of the constants defined in the first 386 lines of code. The only difference found was for the value of C\_sqrd where ICAR Thompson uses 0.3 and WRF Thompson 0.15. We then ran simulations with the C\_sqrd value set to 0.15 but this only yielded negligible differences in the simulation results for the idealized default scenario. Additionally we checked the code for differences in the modifications made since it was forked from the WRF repository. Where we found differences we undid the changes and tested whether the idealized simulations were affected – we did not find any indication that the functionality of ICAR Thompson differed from WRF Thompson. As an additional check we simulated the idealized default scenario with the WRF version from which the ICAR Thompson code was forked from (WRF-3.4) and noticed only negligible differences to the results obtained with the WRF 4.1.1 version.

We rephrased to better indicate the code review and WRF 3.4 simulations employed to rule out differences in the Thompson microphysics implementation.

**Adjustment to the manuscript**

**P09L23:** The Thompson microphysics scheme as described in Sect. 2 is employed in all models. The **ICAR implementation of the Thompson MP was forked from WRF version 3.4. Preliminary tests were conducted, showing that WRF 3.4 and WRF 4.1.1 yielded the same results for the default scenario, with only negligible differences. Additionally, the** code of the Thompson MP implementation in ICAR and WRF 4.1.1 was reviewed and tested to ensure that both implementations produce the same results for the same input differences between the implementations did not affect the results. All input files and model configurations are available for download (Horak, 2020).

**AR** (continued):**

We therefore attribute the different results of ICAR-N and WRF to subtle differences in the wind fields. Nonetheless, despite our many attempts to identify the process responsible for this difference, the results are not conclusive. Overall we believe that a more detailed analysis that would get to the bottom of this is outside the scope of this paper and that the focus of our

manuscript was not the detailed investigation of the Thompson microphysics scheme. To better clarify that this is an open issue we added a sentence to the corresponding paragraph in the discussion.

**Adjustment to the manuscript**

**P39L16:** However, not all reasons for the differences could be identified, results remain inconclusive as to why ICAR-N mainly produces cloud ice while it is cloud water in WRF.

Thompson, G., Field, P. R., Rasmussen, R. M., & Hall, W. D. (2008). Explicit forecasts of winter precipitation using an improved bulk microphysics scheme. Part II: Implementation of a new snow parameterization. *Monthly Weather Review*, *136*(12), 5095-5115.

Bigg, E. K. (1953). The supercooling of water. *Proceedings of the Physical Society. Section B*, 66(8), 688.

Srivastava, R. C., & Coen, J. L. (1992). New Explicit Equations for the Accurate Calculation of the Growth and Evaporation of Hydrometeors by the Diffusion of Water Vapor, *Journal of Atmospheric Sciences*, *49*(17), 1643-1651. Retrieved Jan 3, 2021, from <a href="https://journals.ametsoc.org/view/journals/atsc/49">https://journals/atsc/49</a>

Jones, K. F., Thompson, G., Claffey, K. J., & Kelsey, E. P. (2014). Gamma Distribution Parameters for Cloud Drop Distributions from Multicylinder Measurements, *Journal of Applied Meteorology and Climatology*, *53*(6), 1606-1617.

**RC:** P33, L18 – How computationally frugal is ICAR compared to WRF simulations in the case study explored here? Can you provide comparison of computational costs and wallclock time?

**AR:** While one of the main advantages of ICAR is computational frugality we considerate not to be the focus of the manuscript. Our intent was to develop recommendations that improve the reliability of the model and the representation of cloud formation processes in it. Therefore, we chose not to include this aspect in the, already long, manuscript since this would additionally require a discussion of the influence of other parameters in the setup (such as model time step for instance). Note that Gutmann (2016) state in their Section 5 that ICAR conducted simulations for their central US domain were 140-400 times faster than WRF. In our RH=100% case, the WRF simulation with a model top at 26 km that simulated a 30 hour period required approximately 300 core hours to complete (model time step 2 s). In contrast to that, the ICAR-N simulation with a model top at 20.4 km required 19 core hours (model time step approximately 40 s), and the ICAR-N simulation with  $z_{top}$ =10.4 km required 16 core hours (model time step approximately 40 s). Note that longer model time steps for WRF would have led to numerical instabilities.

**RC:** P4, L 7 – remove the word "eventually", removing it still keeps the same message and the word is unnecessary

AR: Due to a larger revision of Section 2.1 the corresponding sentence was removed.

RC: P5, L23 – change "is" to "are"

**AR:** Due to a larger revision of Section 2.2.2 the corresponding sentence was altered and now reads: **P07L16:** "Note that the vertical wind components is defined as k-1/2 and k+1/2 are

calculated at half levels with Eq. (8) and that, in particular, no boundary condition is required to determine w at the model top."

P8, L3 – rewrite to "Since ICAR does not currently support..."

**AR:** Rephrased accordingly and now reads: "Since ICAR **does not** currently **does not** support periodic boundary conditions,"

**RC:** P15 – the caption for Figure 1 seems incorrect... "Perturbations of the horizontal perturbation", should this be "Vertical cross-sections of the horizontal perturbation"?

**AR:** We corrected the caption accordingly, the caption now reads: "Perturbations Vertical crosssections of the horizontal perturbation wind component u' (top row) and vertical perturbation wind component w'"

**RC:** P20, L3 – it should be "Fig. 6a-g"

**AR:** We corrected the reference accordingly, the sentence now reads: "As shown in Fig. 6a-**g**, for most investigated quantities"

RC: P20, L23 – clarify that you mean spatial distribution of these quantities

**AR:** We added the word spatial, the sentence now reads: "In other words, the **spatial** distribution of these quantities needs to be taken into account as well."

**RC:** P21 – The blue contours in Figure 7a are difficult to distinguish for me, perhaps adding a different line type (dashed, dotted, etc) could help, or different colors that aren't so similar

**AR:** We modified the linestyles in the figure to emphasize differences in the contours.

**Adjustment to the manuscript**

P25:

**RC:** P24, L17 – should be Fig. 10d**

**AR:** We corrected the reference, see the next AR below for the adjustment to the manuscript. **RC:** P24, L18 – should be Fig, 10b

**AR:** We corrected the reference accordingly.

**Adjustment to the manuscript**

**P28L16:**

This is caused by a combination of two factors: (i) The higher vertical wind speeds above the windward slope of the topographical ridge predicted by WRF, lead to lower effective falls speeds of the hydrometeors (see Fig.10d). (ii) Higher horizontal wind speeds additionally contribute to a larger horizontal drift of  $q_{prc}$  and precipitation spill-over in WRF (see Fig. 10b and, for a basic estimation of the drift distances, Sect. 4.5.2).

**RC:** P26 – Can you have the isentropes be at the same interval and starting potential temperature as in Fig 9? This will facilitate comparison better.

**AR:** We plotted the isentropes in Fig. 10 to the same interval as in Fig. 9.

**Adjustment to the manuscript**

---

## Author Response (AR2)

**Technical Corrections**

Please find below a list of the technical corrections made to the revised manuscript.

Johannes Horak

**P01L03:**
Correction of the phrasing in a sentence.

> different chain of processes than found in observations. While  guidelines and strategies exist in the atmospheric sciences to maximize the chances that models are correct for the right reasons, these are mostly applicable
>
> 5   to full-physics models, such as numerical weather prediction models. The Intermediate Complexity Atmospheric Research

**P01L25:**
We added a missing "an" and rephrased so that "such" doesn't occur twice in the same sentence.

> 25   when model skill is evaluated from statistical metrics based on comparisons to surface observations only, such an analysis may
>
> not reflect the skill of the model in capturing atmospheric processes  like gravity waves and cloud formation.

**P02L11:**
We split a sentence in two.

> systems place further limits on the verifiability of models. The same limitations apply to model evaluation as well.
>
> However, evaluation focuses on establishing the reliability of a model rather than its truth.

**P03L09:**
We added the word "precipitation" to be more specific. This is a change resulting from a revision that clarified better that problems do not stem from comparing models to measurements, but only if those are isolated measurements without consideration of the underlying process. The resulting changes were not carried over to this sentence.

> However, in the literature the evaluation efforts for ICAR so far focused mainly on comparisons to precipitation measurements or WRF output. Gutmann et al. (2016) compared monthly precipitation fields for Colorado, USA, obtained from ICAR

**P04L04:**
While iterating the manuscript the word "time-dependent" was dropped from this sentence. At this point 3-D should have changed to 4-D, we have corrected this now.

> ICAR are a digital elevation model supplying the high-resolution topography $h(x,y)$ and forcing data, i.e., a set of  4-D
>
> 5   atmospheric variables as supplied by atmospheric reanalysis such as ERA5 or coupled atmosphere-ocean general circulation

**P05L07:**
We added a part to clarify since w' and w are used interchangeably throughout the manuscript but it is never clearly stated (since w = W+w' and, in linear theory, W = 0).

> The vertical wind speed perturbation  $w = w'$ is calculated from the divergence of the horizontal winds $u$ and $v$, where $u = U + u'$ and $v = V + v'$, as

**P05L27:**

Equation 10 erroneously stated (x,y) ∈ B instead of (x,y) ∈ L.

data set and specified by a Dirichlet boundary condition as

$$\psi(x,y,z,t)\big|_{\cancel{(x,y)\in B}\,(x,y)\in L} = \psi_F(x,y,z,t), \tag{10}$$

**P05L29:**
We corrected for better phrasing.

At the upper boundary $T$ where $n_z = N_z$ and $N_z$  as the grid points along the $z$ direction, a zero gradient Neumann

30    boundary condition is imposed:

**P06L04:**
We removed remains from an iteration of the manuscript between submission and revision.

Note that capital $\Psi$ denotes not only the advected quantities $\psi$ but also $p$, .

**P11L01:**
We rephrased to add references to the equations in the manuscript and removed the reference to Gutmann et al. (2016) since the equation is now introduced in our revised manuscript.

ICAR calculates the perturbations to the horizontal background wind with  Eq. (1) and Eq. (2) while the vertical wind speed is calculated  according to Eq. (8). Perturbations to the potential temperature and microphysics species fields, on the other hand, result from

5    advection and microphysical processes calculated with numerical methods. In ICAR-O this introduces a time dependency for

**P13L19:**
We corrected term "group" to "species" since this is the term introduced in context with the microphysics scheme in Section 2.1.

ratio of the respective hydrometeor  species and $\rho_{ij}(t)$ the density of dry air within the grid cell. Note that in contrast to

**P15L32:**
We added local time since the abbreviation LT is never introduced.

The case study focuses on the 6 May 2015 LT (local time), a day with stably stratified large-scale northwesterly flow through-

**P16L14:**
We corrected "south-north" to the correct specification "north-south".

$-8.4\,\mathrm{m\,s^{-1}}$ to $8.2\,\mathrm{m\,s^{-1}}$ compared to the $-10.0\,\mathrm{m\,s^{-1}}$ to $10.0\,\mathrm{m\,s^{-1}}$ derived from the analytical expression. While, for the  north-south perturbations, the analytical solution yields $v' = 0\,\mathrm{m\,s^{-1}}$, ICAR-N calculates an average magnitude

**P19L21:**
We corrected a non subscripted v to a subscript v.

 $\tilde{\phi}_z(q_v)$ still exhibits minima and maxima at higher elevations due to the periodicity of the vertical velocity field (see

**P21 Caption of Figure 3:**

We corrected the indicated contour spacings, they erroneously listed the intervals of the filled contours that are already shown in the colorbars. However, they should state the intervals of the violet to teal contour lines, which they now do.

teal contour lines weaker vertical convergence in the range  between $-4.5 \cdot 10^{-4}\,\mathrm{s}^{-1}$ and $0\,\mathrm{s}^{-1}$ spaced in increments of  $0.9 \cdot 10^{-4}\,\mathrm{s}^{-1}$. The red contour line indicates where $w = 0\,\mathrm{m\,s}^{-1}$. In panel (b) grey and black lines additionally indicate the

**P22 Figure 4:**
We corrected the y-axis label. This is in response to a previous reviewer comment that was addressed in the text. However, the y-axis label in the Figure was not adjusted, it previously read "horizontally averaged MAE (K), now it is "MAE in dependence of elevation (K)"

[Figure]

**P30 Figure 10:**
We slightly adjusted some label positions in Figure 10 and made adjustments so that the isentrope labels are now not crossed out by the isentropes anymore. The changes are highlighted with orange circles in the "Previous Figure 10" farther below.

Updated Figure 10:

[Figure]

Previous Figure 10:

[Figure]

**P30 Figure 10:**
We adjusted the title of Figure 13f. It was previously erroneously "f) ICAR-N – ICAR-O$_{4km}$" and now correctly states " f) ICAR-N – ICAR-O$_{15.2km}$" (orange highlight in text indicates the change).

**P38L35:**
We adjusted the phrasing to avoid double brackets.

> triggered by higher topographies(, see Eq. (19)). However, note that the dependence on the ridge height is weak compared to the dependence on the background state.

**P39L07:**
We added a missing unit

> which correspond to approximately the uppermost  1 km to 2 km of the domain, and result in only a negligible influence on

**P41L07:**
Corrected a spelling error

> – The method described in this study to determine $Z_{min}$ may be applied to idealized simulations and  real cases alike. This was demonstrated as proof of concept.

**P41L11:**
We rephrased for clarity.

> – While most of the tested boundary conditions (in comparison to the default zero gradient boundary condition) are suit-
> able to reduce the errors in the water vapor and potential temperature fields, no tested combination of these boundary
> conditions  results in a lower value for $Z_{min}$.

10